# On the Training Convergence of Transformers for In-Context Classification of Gaussian Mixtures

**Wei Shen** [* 1]  **Ruida Zhou** [* 2]  **Jing Yang** [1]  **Cong Shen** [1]

## Abstract

Although transformers have demonstrated impressive capabilities for in-context learning (ICL) in practice, theoretical understanding of the underlying mechanism that allows transformers to perform ICL is still in its infancy. This work aims to theoretically study the training dynamics of transformers for in-context classification tasks. We demonstrate that, for in-context classification of Gaussian mixtures under certain assumptions, a single-layer transformer trained via gradient descent converges to a globally optimal model at a linear rate. We further quantify the impact of the training and testing prompt lengths on the ICL inference error of the trained transformer. We show that when the lengths of training and testing prompts are sufficiently large, the prediction of the trained transformer approaches the ground truth distribution of the labels. Experimental results corroborate the theoretical findings.

## 1. Introduction

Large language models (LLMs) based on the transformer architecture (Vaswani et al., 2017) have demonstrated remarkable in-context learning (ICL) abilities (Brown et al., 2020). When given a prompt consisting of examples of a learning task, these models can learn to solve this task for new test examples without any parameter updating. This behavior has been empirically demonstrated in state-of-the-art models on real-world tasks (OpenAI, 2023; Touvron et al., 2023).

This impressive capacity of transformer-based models has inspired many recent works aiming to understand the ICL abilities of transformers. A more comprehensive literature review can be found in Appendix A. Garg et al. (2022) was the first to study the ICL abilities of transformers for various function classes. They empirically showed that transformers can learn linear regression models in context. Later on, a line of research was developed to theoretically explain how transformers perform in-context linear regression. For example, Akyürek et al. (2022); Von Oswald et al. (2023); Bai et al. (2024); Fu et al. (2023); Giannou et al. (2024) showed *by construction* that, some specially-designed transformers can perform linear regression in context. Moreover, some recent works such as Zhang et al. (2023a); Huang et al. (2023); Chen et al. (2024) studied the training dynamics of a single-layer transformer for in-context linear regression. They proved the convergence of certain single-layer transformers during training and showed that the trained transformers are able to perform linear regression in context.

Building on the earlier works that largely focus on linear regression problems, several recent papers have started to investigate the ICL capabilities of transformers for classification problems. For instance, Bai et al. (2024) showed that, by construction, multi-layer transformers can be approximately viewed as multiple steps of gradient descents for logistic regression. Giannou et al. (2024) further showcased that the constructed transformers can approximately perform Newton's method for logistic regression. Lin & Lee (2024) studied the dual operating modes for in-context classification of Gaussian mixtures. However, their analyses were based on the idealized Bayes-optimal next-token predictor for ICL tasks and did not consider the training dynamics of transformer models. Some recent works have started to study the training dynamics of transformers for certain classification problems. For example, Li et al. (2024b) proved that a single-layer transformer can be trained to learn one-nearest neighbor in context. Li et al. (2024a) studied the training dynamics of a single-layer transformer for some binary classification tasks with finite, pairwise orthogonal patterns. Frei & Vardi (2024) analyzed a linear transformer model for in-context classification of Gaussian mixtures, assuming additional conditions such as a sufficiently large signal-to-noise ratio. However, all these works (Li et al., 2024a;b; Frei & Vardi, 2024) only considered the binary classification problems. The training dynamics

---

[*]Equal contribution  [1]Department of Electrical and Computer Engineering, University of Virginia, Charlottesville, USA [2]Department of Electrical Engineering, University of California, Los Angeles, USA. Correspondence to: Cong Shen <cong@virginia.edu>.

*Proceedings of the 42$^{nd}$ International Conference on Machine Learning*, Vancouver, Canada. PMLR 267, 2025. Copyright 2025 by the author(s).

of transformers for more general in-context classification problems beyond the specific settings and assumptions in Li et al. (2024a;b); Frei & Vardi (2024) remain largely under-explored.

In this work, we study the training dynamics of a single-layer transformer for both binary and multi-class classification of Gaussian mixtures during ICL, a fundamental problem in machine learning. Our main contributions can be summarized as follows:

- We prove that with appropriately distributed training data (Assumptions 3.2, 4.2), a single-layer transformer trained via gradient descent will converge to its global minimizer at a linear rate (Theorems 3.3, 4.3) for both in-context binary or multi-class classification problems. To the best of our knowledge, we are the first to prove the training convergence of transformers for in-context multi-class classification. Moreover, our analysis reveals that the trained single-layer transformer can be viewed as approximately implementing linear discriminant analysis (LDA).
- Due to the non-linearity of our loss function, we cannot directly find the closed-form expression of the global minimizer. Instead, we prove an important property that the global minimizer consists of a constant plus an error term that is induced by the finite training prompt length ($N$). We further show that the max norm of this error term is bounded, and converges to zero at a rate of $O(1/N)$.
- With properly distributed testing prompts (Assumptions 3.5, 4.4), we establish an upper bound of the inference error (defined in Equation (3)) of the trained transformer and quantify the impact of the training and testing prompt lengths on this error. We further prove that when the lengths of training prompts ($N$) and testing prompts ($M$) approach infinity, this error converges to zero at a rate of $O(1/N + 1/\sqrt{M})$, and the prediction of the trained transformer has an identical distribution to that of the ground-truth label (Theorems 3.6, 4.5).

## 2. Preliminaries

**Notations.** We denote $[n] = \{1, 2, \ldots, n\}$. For a matrix $A \in \mathbb{R}^{m \times n}$, we denote its Frobenius norm as $\|A\|_F$, and its max norm as $\|A\|_{\max} = \max_{i \in [m], j \in [n]} |A_{ij}|$. We use $A_{a,b}$ (or $A_{ab}$) to represent the element of matrix A at the $a$-th row and $b$-th column, and use $A_{a:c,b}$ to represent a vector of dimension $c - a + 1$ whose $i$-th element is $A_{(a+i-1),b}$. We denote the $l_2$ norm of a vector as $\| \cdot \|_2$. We denote the all-zero vector of size $n$ as $0_n$ and the all-zero matrix of size $m \times n$ as $0_{m \times n}$. We use $\sigma(x) := 1/(1 + \exp(-x))$ to denote the sigmoid function. We define $\mathrm{softmax}(\cdot) : \mathbb{R}^k \to (0,1)^k$, and its $i$-th element as $\mathrm{softmax}(\cdot)_i$, where $\mathrm{softmax}(x)_i = \exp(x_i)/(\sum_{j=1}^k \exp(x_j))$.

### 2.1. Single-layer transformer

Given an input embedding matrix $E \in \mathbb{R}^{d_e \times d_n}$, a single head self-attention module $F_{SA}$ with width $d_e$ will output

$$
\begin{aligned}
&F_{SA}(E; W^V, W^K, W^Q) \\
&= E + W^V E \cdot f_{\mathrm{attn}}\left( \frac{(W^K E)^\top W^Q E}{\rho} \right),
\end{aligned} \quad (1)
$$

where $W^V, W^K, W^Q \in \mathbb{R}^{d_e \times d_e}$ are the value, key, and query weight matrices, respectively, $\rho > 0$ is a normalization factor, and $f_{\mathrm{attn}}$ is an activation function for attention. There are different choices of $f_{\mathrm{attn}}$; for example Vaswani et al. (2017) adopts $\mathrm{softmax}$.

In this work, similar to Zhang et al. (2023a); Wu et al. (2023), we set $f_{\mathrm{attn}}(x) = x$ and define $W^{KQ} = (W^K)^\top W^Q \in \mathbb{R}^{d_e \times d_e}$. We use $F$ to denote this simplified model. Then, the output of $F$ with an input embedding matrix $E \in \mathbb{R}^{d_e \times d_n}$ can be expressed as

$$
F(E; W^V, W^{KQ}) = E + W^V E \cdot \frac{E^\top W^{KQ} E}{\rho}. \quad (2)
$$

In the following theoretical study and the subsequent experiments (Section 5.2), we show that this simplified transformer model has sufficient capability to approach the optimal classifier for the in-context classification of Gaussian mixtures.

### 2.2. In-context learning framework

We adopt a framework for in-context learning similar to that used in Bai et al. (2024). Under this framework, the model receives a prompt $P = (\mathcal{D}, x_{\mathsf{query}})$ comprising a set of demonstrations $\mathcal{D} = \{(x_i, y_i)\}_{i \in [N]} \overset{\text{i.i.d.}}{\sim} \mathcal{P}$ and a query $x_{\mathsf{query}} \sim \mathcal{P}_x$, where $\mathcal{P}$ is the joint distribution of $(x, y)$ and $\mathcal{P}_x$ is the marginal distribution of $x$. Here, $x_i \in \mathbb{R}^d$ is an in-context example, and $y_i$ is the corresponding label for $x_i$. For instance, in regression tasks, $y_i \in \mathbb{R}$ is a scalar. In this paper, we focus on classification tasks. Thus, the range of $y_i$ can be any set containing $c$ different elements, such as $\{1, \ldots, c\}$, for classification problems involving $c$ classes. The objective is to generate an output $\widehat{y}_{\mathsf{query}}$ that approximates the target $y_{\mathsf{query}} \sim \mathcal{P}_{y|x_{\mathsf{query}}}$.

Since $y_{\mathsf{query}}$ is a discrete random variable, we use the total variation distance to measure the difference between $\widehat{y}_{\mathsf{query}}$ and $y_{\mathsf{query}}$:

$$
\begin{aligned}
&\Delta(y_{\mathsf{query}}, \widehat{y}_{\mathsf{query}}) \\
&= \sup_{z \in R(y_{\mathsf{query}})} |\mathbb{P}(y_{\mathsf{query}} = z) - \mathbb{P}(\widehat{y}_{\mathsf{query}} = z)|, \quad (3)
\end{aligned}
$$

where $R(y_{\mathsf{query}})$ is the range of $y_{\mathsf{query}}$. When $\Delta(y_{\mathsf{query}}, \widehat{y}_{\mathsf{query}}) = 0$, $\widehat{y}_{\mathsf{query}}$ has the same distribution as $y_{\mathsf{query}}$, which means the output of the model perfectly approximates $y_{\mathsf{query}}$.

Unlike standard supervised learning, each prompt $P_\tau$ can be sampled from a different distribution $\mathcal{P}_\tau$ in ICL. We say that a model has the *ICL capability* if it can approximate $y_{\tau,\text{query}}$ for a broad range of $\mathcal{P}_\tau$'s with fixed parameters.

## 3. In-context binary classification

In this section, we study the learning dynamics of a single-layer transformer for in-context binary classification. It is a special case of the general multi-class classification. As a result, the analysis is more concise. The general in-context multi-class classification problem is studied in Section 4.

We first introduce the prompt and the transformer structure we will use for in-context binary classification. The prompt for in-context binary classification is denoted as $P = (x_1, y_1, \ldots, x_N, y_N, x_{\text{query}})$, where $x_i \in \mathbb{R}^d$ and $y_i \in \{-1, 1\}$. We can convert this prompt $P$ into its corresponding embedding matrix $E(P)$ in the following form:

$$E = E(P) = \begin{pmatrix} x_1 & x_2 & \cdots & x_N & x_{\text{query}} \\ y_1 & y_2 & \cdots & y_N & 0 \end{pmatrix}. \quad (4)$$

Similar to Huang et al. (2023); Wu et al. (2023); Ahn et al. (2024), we set some of the parameters in our model to 0 or 1 to simplify the optimization problem, and consider the parameters of our model $(W^V, W^{KQ})$ in the following sparse form:

$$W^V = \begin{pmatrix} 0_{d\times d} & 0_d \\ 0_d^\top & 1 \end{pmatrix}, \qquad W^{KQ} = \begin{pmatrix} W & 0_d \\ 0_d^\top & 0 \end{pmatrix}, \quad (5)$$

where $W \in \mathbb{R}^{d\times d}$. We set the normalization factor $\rho$ equal to the length of the prompt $N$. Let $F(E(P); W)$ be the output matrix of the transformer. We then read out the bottom-right entry of the output matrix through a sigmoid function, and denote this output as $\widehat{y}_{\text{out}}$. The output $\widehat{y}_{\text{out}}$ of the transformer with prompt $P$ and parameters $W$ can be expressed as

$$\widehat{y}_{\text{out}} = \sigma\left([F(E(P); W)]_{(d+1),(N+1)}\right)$$
$$= \sigma\left(\left(\frac{1}{N}\sum_{i=1}^N y_i x_i^\top\right) W x_{\text{query}}\right).$$

We denote the prediction of our model for $x_{\text{query}}$ as $\widehat{y}_{\text{query}}$, which is a random variable depending on $\widehat{y}_{\text{out}}$. Consider generating a random variable $u$ uniformly on $[0, 1]$. If $u \leq \widehat{y}_{\text{out}}$, we output $\widehat{y}_{\text{query}} = 1$; if $u > \widehat{y}_{\text{out}}$, we output $\widehat{y}_{\text{query}} = -1$. Then, we have $\mathbb{P}(\widehat{y}_{\text{query}} = 1) = \widehat{y}_{\text{out}}$, $\mathbb{P}(\widehat{y}_{\text{query}} = -1) = 1 - \widehat{y}_{\text{out}}$.

### 3.1. Training procedure

We study the binary classification of two Gaussian mixtures and use the following definition.

**Definition 3.1.** We say a data pair $(x, y) \sim \mathcal{P}^b(\mu_0, \mu_1, \Lambda)$ if $y$ follows a Bernoulli distribution with $\mathbb{P}(y = -1) = \mathbb{P}(y = 1) = 1/2$ and $f(x|y = -1) = \mathsf{N}(\mu_0, \Lambda)$, $f(x|y = 1) = \mathsf{N}(\mu_1, \Lambda)$, where $\mu_0, \mu_1 \in \mathbb{R}^d$ and $\Lambda \in \mathbb{R}^{d\times d}$ is a positive definite matrix.

We consider the case of $B$ training tasks indexed by $\tau \in [B]$. Each training task $\tau$ is associated with a prompt $P_\tau = (x_{\tau,1}, y_{\tau,1}, \ldots, x_{\tau,N}, y_{\tau,N}, x_{\tau,\text{query}})$ and a corresponding label $y_{\tau,\text{query}}$. We make the following assumption in this section.

**Assumption 3.2.** For each learning task $\tau \in [B]$, we assume

(1) $\{x_{\tau,i}, y_{\tau,i}\}_{i=1}^N$ and $\{x_{\tau,\text{query}}, y_{\tau,\text{query}}\} \overset{\text{i.i.d.}}{\sim} \mathcal{P}^b(\mu_{\tau,0}, \mu_{\tau,1}, \Lambda)$.

(2) $\mu_{\tau,0}$ is randomly sampled from $\mathsf{N}(0, I_d)$, $\mu_{\tau,1} = U_{\tau,\Lambda}\mu_{\tau,0}$ where $U_{\tau,\Lambda} = \Lambda^{1/2}U_\tau\Lambda^{-1/2}$, and $U_\tau$ is uniformly distributed over the closed set of real unitary matrices such that $U_\tau U_\tau^\top = I_d$.

We denote the distribution of $(\mu_{\tau,0}, \mu_{\tau,1})$ as $\mathcal{P}^b_\Omega(\Lambda)$. Note that $U_{\tau,\Lambda} = \Lambda^{1/2}U_\tau\Lambda^{-1/2}$ can be viewed as a linear transformation that preserves the inner product of vectors in $\Lambda^{-1}$-weighted norm, and we have $\mu_{\tau,0}^\top\Lambda^{-1}\mu_{\tau,0} - \mu_{\tau,1}^\top\Lambda^{-1}\mu_{\tau,1} = 0$.

Let $\widehat{y}_{\tau,\text{out}} = \sigma([F(E(P_\tau); W)]_{(d+1),(N+1)})$ be the output of our transformer for task $\tau$. We define the empirical risk over $B$ independent tasks as

$$\widehat{L}(W) = \frac{1}{2B}\sum_{\tau=1}^B -(1 + y_{\tau,\text{query}})\log(\widehat{y}_{\tau,\text{out}})$$
$$- (1 - y_{\tau,\text{query}})\log(1 - \widehat{y}_{\tau,\text{out}}). \quad (6)$$

Taking the limit of infinite training tasks $B \to \infty$, the expected training loss can be defined as

$$L(W) = \lim_{B\to\infty} \widehat{L}(W) = -\frac{1}{2}\mathbb{E}[(1 + y_{\tau,\text{query}})\log(\widehat{y}_{\tau,\text{out}})$$
$$+ (1 - y_{\tau,\text{query}})\log(1 - \widehat{y}_{\tau,\text{out}})], \quad (7)$$

where the expectation is taken over $(\mu_{\tau,0}, \mu_{\tau,1}) \sim \mathcal{P}^b_\Omega(\Lambda)$, $\{x_{\tau,i}, y_{\tau,i}\}_{i=1}^N, \{x_{\tau,\text{query}}, y_{\tau,\text{query}}\} \overset{\text{i.i.d.}}{\sim} \mathcal{P}^b(\mu_{\tau,0}, \mu_{\tau,1}, \Lambda)$.

Applying gradient descent over the expected training loss (7), we have the following theorem.

**Theorem 3.3.** *Under Assumption 3.2, the following statements hold.*

*(1) Optimizing the training loss $L(W)$ in Equation (7) with training prompt length $N$ via gradient descent $W^{t+1} = W^t - \eta\nabla L(W^t)$, we have that for any $t \geq 1$,*

$$\|W^t - W^*\|_F^2 \leq \exp(-t/\kappa)\|W^0 - W^*\|_F^2, \quad (8)$$

*where $W^0$ is the initial parameter and $W^*$ is the global minimizer of $L(W)$, and $\kappa = l/\alpha$. Here $\alpha, l$ are constants satisfying*

$$\alpha \le \lambda_{\min}(\nabla^2 L(W)) \le \lambda_{\max}(\nabla^2 L(W)) \le l, \quad (9)$$

*where $\alpha > 0$, $l < \infty$, $W \in R_W$, $R_W = \{W \in \mathbb{R}^{d \times d} \mid \|W - W^*\|_F \le \|W^0 - W^*\|_F\}$.*

*(2) Define $G = \frac{1}{2} W^* - \Lambda^{-1}$, $q = x_{\tau,\text{query}}$, $\mu = \mu_{\tau,1} - \mu_{\tau,0}$, $u = 2(\mu_{\tau,1} + \mu_{\tau,0})$, and $a = \mu^\top \Lambda^{-1} q$ for simplicity. Then we have*

$$
\begin{aligned}
&\|G\|_{\max} \\
\le &\frac{1}{N} \|S^{-1}(\mathbb{E}[\sigma'(a)(4qq^\top + \frac{1}{4} u u^\top \Lambda^{-1} q q^\top) \\
&+ \sigma''(a)(\frac{1}{8}(u^\top \Lambda^{-1} q)^2 \mu q^\top + 2 q^\top \Lambda^{-1} q \mu q^\top)])\|_{\max} \\
&+ o(1/N),
\end{aligned}
\tag{10}
$$

*where $S = 4\nabla^2 \widetilde{L}(2\Lambda^{-1})$, $\widetilde{L}(2\Lambda^{-1}) = \lim_{N \to \infty} L(2\Lambda^{-1})$, $\sigma'(\cdot)$ and $\sigma''(\cdot)$ are the first- and second-order derivatives of $\sigma(\cdot)$, respectively, and the expectation is taken over $(\mu_{\tau,0}, \mu_{\tau,1}) \sim \mathcal{P}_\Omega^b(\Lambda)$, $x_{\tau,\text{query}} \sim \mathcal{P}_x^b(\mu_{\tau,0}, \mu_{\tau,1}, \Lambda)$.*

The detailed proof of Theorem 3.3 can be found in Appendix D. In the following, we provide a brief proof sketch to highlight the key ideas.

**Proof sketch for Theorem 3.3.** As a first step, we prove in Lemma D.2 that the expected loss function $L(W)$ in Equation (7) is strictly convex with respect to (w.r.t.) $W$ and is strongly convex in any compact set of $\mathbb{R}^{d \times d}$. Moreover, we prove $L(W)$ has one unique global minimizer $W^*$. Since the loss function $L(W)$ we consider is highly non-linear, we cannot directly find the closed-form expression of $W^*$, as is often done in the prior literature. This poses a significant challenge to our analysis.

We address this technical challenge via the following method. First, in Lemma D.3, by analyzing the Taylor expansion of $L(W)$, we prove that as $N \to \infty$, our loss function $L(W)$ converges to $\widetilde{L}(W)$ pointwisely (defined in Equation (25)), and the global minimizer $W^*$ converges to $2\Lambda^{-1}$. Thus, we denote $W^* = 2(\Lambda^{-1} + G)$, and prove $\|G\|_{\max}$ is bounded and scales as $\|G\|_{\max} = O(N^{-1/2})$. Next, in Lemma D.4, by further analyzing the Taylor expansion of the equation $\nabla L(W^*) = 0$ at the point $2\Lambda^{-1}$, we establish a tighter bound $\|G\|_{\max} = O(N^{-1})$. In Lemma D.5, we prove that our loss function is $l$-smooth and provide an upper bound for $l$. Thus, in a compact set $R_W$, our loss function is $\alpha$-strongly convex and $l$-smooth. Finally, leveraging the standard results from the convex optimization, we prove Theorem 3.3.

According to Theorem 3.3, we have $W^t = W^* + H^t$ where $\|H^t\|_{\max} \le \exp(-t/(2\kappa))\|W^0 - W^*\|_F$. If we set

$T \ge 2\kappa \log(N \cdot \|W^0 - W^*\|_F)$, we have $\|H^T\|_{\max} \le 1/N$. Denoting $\widehat{W} = W^T$, we have $\widehat{W} = 2(\Lambda^{-1} + G + H^T/2) = 2(\Lambda^{-1} + \widehat{G})$, where $\widehat{G} = G + H^T/2$, $\|\widehat{G}\|_{\max} \le \|G\|_{\max} + \|H^T\|_{\max} = O(1/N)$. Thus, we have the following corollary.

**Corollary 3.4.** *If we optimize the expected loss $L(W)$ in Equation (7) via gradient descent with training prompt length $N$, initial parameters $W^0$, and learning rate $\eta = 1/l$, then, under Assumption 3.2, after $T \ge 2\kappa \log(N \|W^0 - W^*\|_F)$ steps, the updated model $\widehat{W}$ satisfies*

$$\widehat{W} = 2(\Lambda^{-1} + \widehat{G}), \tag{11}$$

*where $\|\widehat{G}\|_{\max} = O(1/N)$, $\kappa = l/\alpha$, and $\alpha, l$ are constants defined in (9).*

Theorem 3.3 and Corollary 3.4 show that training a single-layer transformer with properly distributed data (Assumption 3.2) for binary classification via gradient descent can *linearly* converge to its global minimum $W^* = 2(\Lambda^{-1} + G)$. Furthermore, when the prompt length $N$ grows, this global minimum $W^*$ will converge to $2\Lambda^{-1}$ at a rate of $O(1/N)$.

### 3.2. In-context inference

Next, we analyze the performance of the trained transformer in Equation (11) for in-context binary classification tasks. We make the following assumption.

**Assumption 3.5.** *For an in-context test prompt $P_{\text{test}} = (x_1, y_1, \ldots, x_M, y_M, x_{\text{query}})$, we assume*

*(1) $\{x_i, y_i\}_{i=1}^M \overset{\text{i.i.d.}}{\sim} \mathcal{P}^b(\mu_0, \mu_1, \Lambda)$, $x_{\text{query}} \in \mathbb{R}^d$.*
*(2) $\mu_0^\top \Lambda^{-1} \mu_0 = \mu_1^\top \Lambda^{-1} \mu_1$.*

With this assumption, for $y_{\text{query}} \sim \mathcal{P}_{y|x_{\text{query}}}^b(\mu_0, \mu_1, \Lambda)$, according to the Bayes' theorem, we have

$$
\begin{aligned}
&\mathbb{P}(y_{\text{query}} = 1 | x_{\text{query}}) \\
=&\frac{f(x_{\text{query}} | y_{\text{query}} = 1) \mathbb{P}(y_{\text{query}} = 1)}{\sum_{z \in \{\pm 1\}} f(x_{\text{query}} | y_{\text{query}} = z) \mathbb{P}(y_{\text{query}} = z)} \\
=&\sigma((\mu_1 - \mu_0)^\top \Lambda^{-1} x_{\text{query}}).
\end{aligned}
$$

If we test the trained transformer with parameters $\widehat{W}$ in Equation (11) and $P_{\text{test}}$, by a simple calculation, we have

$$\widehat{y}_{\text{out}} = \sigma\left(\left(\frac{2}{M} \sum_{i=1}^M y_i x_i^\top\right)(\Lambda^{-1} + \widehat{G}) x_{\text{query}}\right). \tag{12}$$

Intuitively, when the training prompt length $N \to \infty$, we have $\widehat{G} \to 0$, and when the test prompt length $M \to \infty$, we have $\frac{2}{M} \sum_{i=1}^M y_i x_i^\top \to (\mu_1 - \mu_0)^\top$. Thus, when $N, M \to \infty$, $\mathbb{P}(\widehat{y}_{\text{query}} = 1) = \widehat{y}_{\text{out}} \to \sigma((\mu_1 - \mu_0)^\top \Lambda^{-1} x_{\text{query}}) =$

$\mathbb{P}\left(y_{\text{query}} = 1 | x_{\text{query}}\right)$, and the prediction of the trained transformer $\widehat{y}_{\text{query}}$ perfectly matches with the distribution of the ground truth label $y_{\text{query}}$.

By analyzing the Taylor expansion of $\widehat{y}_{\text{out}}$ at point $\sigma((\mu_1 - \mu_0)^\top \Lambda^{-1} x_{\text{query}})$, we formally present the aforementioned intuition in the following theorem, which establishes an upper bound of the total variation distance between $y_{\text{query}}$ and $\widehat{y}_{\text{query}}$.

**Theorem 3.6.** *Consider a test prompt $P_{\text{test}}$ satisfying Assumption 3.5, and let $y_{\text{query}} \sim \mathcal{P}^b_{y|x_{\text{query}}}(\mu_0, \mu_1, \Lambda)$. Let $\widehat{y}_{\text{query}}$ be the prediction of the trained transformer with parameters $\widehat{W}$ in Equation (11). Then, for the inference error defined in Equation (3), we have*

$$\mathbb{E}[\Delta(y_{\text{query}}, \widehat{y}_{\text{query}})]$$
$$\leq \sigma'(\mu^\top \Lambda^{-1} q)\Bigg[\|\widehat{G}\|_{\max} \sum_{i,j \in [d]} |\mu_i q_j|$$
$$+ \frac{1}{\sqrt{M}}\left(\frac{1}{2}|u^\top \Lambda^{-1} q| + \frac{2\sqrt{2}}{\sqrt{\pi}} \sum_{i,j \in [d]} |\Lambda_{ij}^{-1/2} q_j|\right)\Bigg]$$
$$+ o\left(\frac{1}{N} + \frac{1}{\sqrt{M}}\right),$$

*where $\mu = \mu_1 - \mu_0$, $u = 2(\mu_1 + \mu_0)$, $q = x_{\text{query}}$, and the expectation is taken over $\{x_i, y_i\}_{i=1}^M \overset{i.i.d.}{\sim} \mathcal{P}^b(\mu_0, \mu_1, \Lambda)$.*

The proof of Theorem 3.6 can be found in Appendix E. Since $\|\widehat{G}\|_{\max} = O(1/N)$, Theorem 3.6 suggests that if we ignore the constants regarding $\mu_0, \mu_1, \Lambda, x_{\text{query}}$, the expected total variation distance between $y_{\text{query}}$ and $\widehat{y}_{\text{query}}$ is at most $O(1/N + 1/\sqrt{M})$. On the other hand, for data pair $(x, y) \sim \mathcal{P}^b(\mu_0, \mu_1, \Lambda)$, the distribution of $y$, $\mathbb{P}(y = 1|x) = \sigma((\mu_1 - \mu_0)^\top \Lambda^{-1} x)$, can be characterized by a logistic regression model $\sigma(w^\top x + b)$ with parameters $w = \Lambda^{-1}(\mu_1 - \mu_0)$ and $b = 0$. Therefore, when $N, M \to \infty$, the prediction of the trained transformer is equivalent to the optimal logistic regressor for binary classification problems with distribution $\mathcal{P}^b(\mu_0, \mu_1, \Lambda)$.

Note that different from Assumption 3.2 which states that $\mu_{\tau,0}, \mu_{\tau,1}, x_{\tau,\text{query}}$ are sampled according to some specific distributions during training, Assumption 3.5 does not impose strong distributional constraints on $\mu_0, \mu_1$ and $x_{\text{query}}$, which shows the strong generalization ability of the trained transformer. Moreover, even if $M \to \infty$, the distribution variation between $y_{\text{query}}$ and $\widehat{y}_{\text{query}}$ does not disappear unless $N \to \infty$. Thus, the ICL ability of trained transformers for binary classification is limited by the finite length of training prompts. Similar behaviors have also been observed in Zhang et al. (2023a) for in-context linear regression.

*Remark* 3.7. Theorem 3.6 requires Assumption 3.5 to hold. For example, we need the covariance matrix $\Lambda$ in training and testing to be the same. A similar consistency requirement of $\Lambda$ in training and testing had also been observed for in-context linear regression in Zhang et al. (2023a). Here, we discuss the consequences when Assumption 3.5 does not hold. For example, suppose the labels of our data in test prompts are not balanced where $\mathbb{P}(y = 1) = p_1, \mathbb{P}(y = -1) = p_0$. Besides, $\mu_0, \mu_1$ do not have the same $\Lambda^{-1}$ weighted norm, and the covariance matrix of test data satisfies $\Gamma \neq \Lambda$. Then, as $N, M \to \infty$, we have

$$\frac{2}{M} \sum_{i=1}^M y_i x_i^\top \to 2(p_1 \mu_1 - p_0 \mu_0)^\top,$$

and

$$\mathbb{P}(\widehat{y}_{\text{query}} = 1) \to \sigma(2(p_1 \mu_1 - p_0 \mu_0)^\top \Lambda^{-1} x_{\text{query}}).$$

On the other hand, the distribution of the ground truth label is $\mathbb{P}(y_{\text{query}} = 1) = \sigma((\mu_1 - \mu_0)^\top \Gamma^{-1} x_{\text{query}} + (\mu_0^\top \Lambda^{-1} \mu_0 - \mu_1^\top \Lambda^{-1} \mu_1)/2 + \log(p_1/p_0))$. Define $z \triangleq (\mu_1 - \mu_0)^\top \Gamma^{-1} x_{\text{query}} + (\mu_1^\top \Lambda^{-1} \mu_1 - \mu_0^\top \Lambda^{-1} \mu_0)/2 + \log(p_1/p_0)$ and $\hat{z} \triangleq 2(p_1 \mu_1 - p_0 \mu_0)^\top \Lambda^{-1} x_{\text{query}}$. Then, we can see that unless $\hat{z} = z$ or $|\sigma(\hat{z}) - \sigma(z)|$ is sufficiently small, the transformer cannot correctly perform the in-context binary classification.

*Remark* 3.8. Another important insight of our analysis is that the pre-trained single-layer transformer can be viewed as approximately implementing linear discriminant analysis (LDA). For example, suppose we are given $\{x_i, y_i\}_{i=1}^M$ and $x_{\text{query}}$, and we need to predict the label $y_{\text{query}}$ for $x_{\text{query}}$. LDA assumes that $\{x_i, y_i\}_{i=1}^M$ and $\{x_{\text{query}}, y_{\text{query}}\}$ are i.i.d. samples, with $\mathbb{P}(y_i = 1) = \mathbb{P}(y_i = -1)$, and the conditional probability density functions $f(x_i|y_i = 1)$ and $f(x_i|y_i = -1)$ are Gaussian with means $\mu_1, \mu_{-1}$ and same covariance $\Sigma$. Under these assumptions, it can be derived that the optimal decision criterion for $x_{\text{query}}$ is to predict $y_{\text{query}} = 1$ if $(\mu_1 - \mu_{-1})^\top \Sigma^{-1} x_{\text{query}} + \frac{1}{2}(\mu_{-1}^\top \Sigma^{-1} \mu_{-1} - \mu_1^\top \Sigma^{-1} \mu_1) > 0$ and $y_{\text{query}} = -1$, otherwise. LDA can estimate $\hat{\mu}_1$ as the average of $x_i$ with $y_i = 1$, estimate $\hat{\mu}_{-1}$ as the average of $x_i$ with $y_i = -1$, and estimate the covariance $\hat{\Sigma}$ from the within-class variances. For the single-layer transformer, it can compute the in-context estimate $\hat{\mu}_1 - \hat{\mu}_{-1} = \frac{1}{M} \sum_{i=1}^M y_i x_i$, however, it is hard for the single-layer transformer to estimate $\hat{\Sigma}$ in context. Thus, in our paper, we make the following assumptions (Assumptions 3.2 and 3.5). We assume the pre-train data and test data have the same covariance matrix $\Lambda$ so that the transformer can learn an approximation of $\Lambda$ during pre-training. Moreover, we assume the two class means $\mu_0, \mu_1$ have the same $\Lambda$-weighted norm so that $\mu_0^\top \Sigma^{-1} \mu_0 - \mu_1^\top \Sigma^{-1} \mu_1 = 0$. Under these assumptions, the quadratic term cancels out, and the estimated decision criterion simplifies to $(\frac{1}{M} \sum_{i=1}^M y_i x_i)^\top \Lambda^{-1} x_{\text{query}}$, which is very close to Equation (12) in our paper. When we use $\widehat{W}$ to approximate $2\Lambda^{-1}$, the estimated decision

criterion becomes exactly Equation (12). Therefore, when $\widehat{W} = 2\Lambda^{-1}$ and the in-context examples are balanced across classes, the transformer's decision criterion is the same as that of the LDA with exact knowledge of $\Lambda$. Our experiments also corroborate this theoretical findings. For example, in Figure 3, since the pre-trained transformer has already learned a relatively good approximation of $\Lambda^{-1}$, while LDA must estimate $\Lambda^{-1}$ in context, the trained transformer significantly outperforms LDA when the number of in-context examples is small. As the context length increases, LDA's performance approaches that of the trained transformer.

# 4. In-context multi-class classification

We now extend the study of the learning dynamics of a single-layer transformer to in-context multi-class classification, generalizing the results of the previous section. We will present the detailed formulation and then focus on the main differences to binary classification.

We first introduce the prompt and the transformer structure that will be used for in-context multi-class classification. The prompt for in-context multi-class classification involving $c \geq 2$ classes can be expressed as $P = (x_1, y_1, \ldots, x_N, y_N, x_{\text{query}})$, where $x_i \in \mathbb{R}^d$, $y_i \in \{\mathbf{e}_1, \mathbf{e}_2, \ldots, \mathbf{e}_c\}$, and $\mathbf{e}_i$ is the $i$-th standard unit vector of $\mathbb{R}^c$. Its embedding matrix can be formulated as

$$E = E(P) = \begin{pmatrix} x_1 & x_2 & \cdots & x_N & x_{\text{query}} \\ y_1 & y_2 & \cdots & y_N & 0_c \end{pmatrix}. \quad (13)$$

Similar to the binary case, we set some of the parameters in our model as 0 and 1 to simplify the optimization problem and consider the parameters of our model $(W^V, W^{KQ})$ in the following sparse form:

$$W^V = \begin{pmatrix} 0_{d \times d} & 0_{d \times c} \\ 0_{c \times d} & I_c \end{pmatrix}, \qquad W^{KQ} = \begin{pmatrix} W & 0_{d \times c} \\ 0_{c \times d} & 0_{c \times c} \end{pmatrix}, \quad (14)$$

where $W \in \mathbb{R}^{d \times d}$. We set the normalization factor $\rho$ equal to the length of the prompt $N$. We read out the bottom-right $c$-dimensional column vector from the output matrix with a softmax function as the output, denoted as $\widehat{y}_{\text{out}}$. With parameters $W$ and a prompt $P = (x_1, y_1, \ldots, x_N, y_N, x_{\text{query}})$, the output can be expressed as

$$\widehat{y}_{\text{out}} = \text{softmax}\left([F(E(P); W)]_{(d+1):(d+c),(N+1)}\right)$$
$$= \text{softmax}\left(\left(\frac{1}{N} \sum_{i=1}^N y_i x_i^\top\right) W x_{\text{query}}\right).$$

We denote the prediction of the model for $x_{\text{query}}$ as $\widehat{y}_{\text{query}}$, which is a random variable depending on $\widehat{y}_{\text{out}}$. Randomly sample a random variable $u$ that is uniformly distributed on

$[0, 1]$. If $u \in \left[\sum_{j=1}^{i-1}(\widehat{y}_{\text{out}})_j, \sum_{j=1}^i(\widehat{y}_{\text{out}})_j\right)$, where $(\widehat{y}_{\text{out}})_j$ is the $j$-th element of $\widehat{y}_{\text{out}}$, we let $\widehat{y}_{\text{query}} = \mathbf{e}_i$. Thus, $\mathbb{P}(\widehat{y}_{\text{query}} = \mathbf{e}_i) = (\widehat{y}_{\text{out}})_i$.

## 4.1. Training procedure

We focus on the multi-class classification of Gaussian mixtures and use the following definition.

**Definition 4.1.** We say a data pair $(x, y) \sim \mathcal{P}^m(\mu, \Lambda)$ if $\mathbb{P}(y = \mathbf{e}_i) = 1/c$ and $f(x|y = \mathbf{e}_i) = \mathsf{N}(\mu_i, \Lambda)$ for $i \in [c]$, where $\mu = (\mu_1, \ldots, \mu_c) \in \mathbb{R}^{d \times c}$ and $\Lambda \in \mathbb{R}^{d \times d}$ is a positive definite matrix.

We consider the case of $B$ training tasks indexed by $\tau \in [B]$. Each training task $\tau$ is associated with a prompt $P_\tau = (x_{\tau,1}, y_{\tau,1}, \ldots, x_{\tau,N}, y_{\tau,N}, x_{\tau,\text{query}})$ and a corresponding label $y_{\tau,\text{query}}$. We make the following assumption in this section.

**Assumption 4.2.** For each learning task $\tau \in [B]$, we assume:

(1) $\{x_{\tau,i}, y_{\tau,i}\}_{i=1}^N$ and $\{x_{\tau,\text{query}}, y_{\tau,\text{query}}\} \overset{\text{i.i.d.}}{\sim} \mathcal{P}^m(\mu_\tau = (\mu_{\tau,1}, \ldots, \mu_{\tau,c}), \Lambda)$.

(2) $\mu_{\tau,1}$ is sampled from $\mathsf{N}(0, I_d)$, $\mu_{\tau,k} = U_{\tau,k,\Lambda}\mu_{\tau,1}$, $k = 2, 3, \ldots, c$, where $U_{\tau,k,\Lambda} = \Lambda^{1/2}U_{\tau,k}\Lambda^{-1/2}$, and $U_{\tau,k}$ are uniformly distributed over the closed set of real unitary matrices such that $U_{\tau,k}U_{\tau,k}^\top = I_d$.

We denote the distribution of $\mu_\tau$ as $\mathcal{P}_\Omega^m(\Lambda)$. Note that $U_{\tau,k,\Lambda} = \Lambda^{1/2}U_{\tau,k}\Lambda^{-1/2}$ can be viewed as linear transformation that preserves the inner product of vectors in the $\Lambda^{-1}$ weighted norm, and we have $\mu_{\tau,i}^\top \Lambda^{-1} \mu_{\tau,i} = \mu_{\tau,j}^\top \Lambda^{-1} \mu_{\tau,j}$, for $i, j \in [c]$. Let $\widehat{y}_{\tau,\text{out}} = \text{softmax}\left([F(E(P_\tau); W)]_{(d+1):(d+c),(N+1)}\right)$ be the output of the transformer for task $\tau$. We define the empirical risk over $B$ independent tasks as

$$\widehat{L}(W) = \frac{1}{B} \sum_{\tau=1}^B \sum_{k=1}^c -(y_{\tau,\text{query}})_k \log((\widehat{y}_{\tau,\text{out}})_k). \quad (15)$$

Taking the limit of infinite training tasks $B \to \infty$, the expected training loss can be defined as

$$L(W) = \lim_{B \to \infty} \widehat{L}(W)$$
$$= -\mathbb{E}\left[\sum_{k=1}^c (y_{\tau,\text{query}})_k \log((\widehat{y}_{\tau,\text{out}})_k)\right], \quad (16)$$

where the expectation is taken over $\mu_\tau \sim \mathcal{P}_\Omega^m(\Lambda)$, $\{x_{\tau,i}, y_{\tau,i}\}_{i=1}^N, \{x_{\tau,\text{query}}, y_{\tau,\text{query}}\} \overset{\text{i.i.d.}}{\sim} \mathcal{P}^m(\mu_\tau, \Lambda)$.

Applying gradient descent over the expected training loss in Equation (16), we have the following theorem.

**Theorem 4.3.** *(Informal) Under Assumption 4.2, the following statements hold.*

(1) *Optimizing training loss $L(W)$ in Equation* (16) *with training prompt length $N$ via gradient descent $W^{t+1} = W^t - \eta \nabla L(W^t)$, for any $t \geq 1$, we have*

$$\|W^t - W^*\|_F^2 \leq \exp(-t/\kappa)\|W^0 - W^*\|_F^2, \quad (17)$$

*where $W^0$ is the initial parameter and $W^*$ is the global minimizer of $L(W)$, $\kappa = l/\alpha$. Here, $\alpha, l$ are constants such that*

$$\alpha \leq \lambda_{\min}(\nabla^2 L(W)) \leq \lambda_{\max}(\nabla^2 L(W)) \leq l, \quad (18)$$

*where $\alpha > 0$, $l < \infty$, $W \in R_W, R_W = \{W \in \mathbb{R}^{d \times d} \mid \|W - W^*\|_F \leq \|W^0 - W^*\|_F\}$.*

(2) *Defining $G = W^*/c - \Lambda^{-1}$, we have $W^* = c(\Lambda^{-1} + G)$ and $\|G\|_{\max} = O(c/N)$.*

(3) *After $T \geq 2\kappa \log(N \cdot \|W^0 - W^*\|_F)$ steps, denoting the updated model $\widehat{W}$ satisfies*

$$\widehat{W} = c(\Lambda^{-1} + \widehat{G}), \quad (19)$$

*where $\|\widehat{G}\|_{\max} = O(c/N)$.*

The formal statement and complete proof of Theorem 4.3 can be found in Appendix F. Technically, the proof of Theorem 4.3 builds on that of Theorem 3.3, but the more complicated cross terms in the Taylor expansions of the softmax functions, which are due to the nature of *multi-class* classification, bring new challenges to the analysis. To address these issues, we derive new bounds on the expected errors of the cross terms in Lemma F.1, F.2, which may be of independent interest to other similar problems.

Theorem 4.3 shows that training a single-layer transformer with properly distributed data (Assumption 4.2) for in-context multi-class classification via gradient descent can linearly converge to its global minimum $W^* = c(\Lambda^{-1} + G)$. When the prompt length $N$ grows, this global minimum $W^*$ will converge to $c\Lambda^{-1}$ at a rate of $O(c/N)$. Compared to the binary case, the new results establish the scaling behavior w.r.t. the number of classes $c$.

## 4.2. In-context inference

**Assumption 4.4.** For an in-context test prompt $P_{\text{test}} = (x_1, y_1, \ldots, x_M, y_M, x_{\text{query}})$, we assume

(1) $\{x_i, y_i\}_{i=1}^M \overset{\text{i.i.d.}}{\sim} \mathcal{P}^m(\mu, \Lambda)$, $\mu = (\mu_1, \ldots, \mu_c) \in \mathbb{R}^{d \times c}$, $x_{\text{query}} \in \mathbb{R}^d$.
(2) $\mu_i^\top \Lambda^{-1} \mu_i = \mu_j^\top \Lambda^{-1} \mu_j$, for $i, j \in [c]$.

With this assumption, for $y_{\text{query}} \sim \mathcal{P}^m_{y|x_{\text{query}}}(\mu, \Lambda)$, according to the Bayes' theorem, we have

$$\mathbb{P}(y_{\text{query}} = \mathbf{e}_k | x_{\text{query}})$$
$$= \frac{f(x_{\text{query}} | y_{\text{query}} = \mathbf{e}_k)\mathbb{P}(y_{\text{query}} = \mathbf{e}_k)}{\sum_{j=1}^c f(x_{\text{query}} | y_{\text{query}} = \mathbf{e}_j)\mathbb{P}(y_{\text{query}} = \mathbf{e}_j)}$$
$$= \text{softmax}(\mu^\top \Lambda^{-1} x_{\text{query}})_k.$$

If we test the trained transformer with parameters $\widehat{W}$ in Equation (19) and prompt $P_{\text{test}}$, by a simple calculation, we have

$$\widehat{y}_{\text{out}} = \text{softmax}\left(\left(\left(\frac{c}{M}\sum_{i=1}^M y_i x_i^\top\right)(\Lambda^{-1} + \widehat{G})x_{\text{query}}\right)\right). \quad (20)$$

Note that, when the training prompt length $N \to \infty$, we have $\widehat{G} \to 0$, and when the test prompt length $M \to \infty$, we have $\frac{c}{M}\sum_{i=1}^M y_i x_i^\top \to \mu^\top$. Thus, when $N, M \to \infty$, $\mathbb{P}(\widehat{y}_{\text{query}} = \mathbf{e}_k) = (\widehat{y}_{\text{out}})_k \to \text{softmax}(\mu^\top \Lambda^{-1} x_{\text{query}})_k = \mathbb{P}(y_{\text{query}} = \mathbf{e}_k | x_{\text{query}})$, i.e., the prediction of the trained transformer $\widehat{y}_{\text{query}}$ matches the ground truth label $y_{\text{query}}$.

By analyzing the Taylor expansion of $\widehat{y}_{\text{out}}$ at point $\text{softmax}(\mu^\top \Lambda^{-1} x_{\text{query}})$, we crystallize the aforementioned intuition in the following theorem, which establishes an upper bound of the total variation distance between $y_{\text{query}}$ and $\widehat{y}_{\text{query}}$.

**Theorem 4.5.** *(Informal) Let $P_{\text{test}}$ satisfy Assumption 4.4 and $y_{\text{query}} \sim \mathcal{P}^m_{y|x_{\text{query}}}(\mu, \Lambda)$. Denote $\widehat{y}_{\text{query}}$ as the prediction of the trained transformer with parameter $\widehat{W}$ in Equation* (19). *Then, for the inference error defined in Equation* (3), *we have*

$$\mathbb{E}[\Delta(y_{\text{query}}, \widehat{y}_{\text{query}})] = O(c^2 N^{-1} + c^{3/2} M^{-1/2}),$$

*where the expectation is taken over $\{x_i, y_i\}_{i=1}^M \overset{\text{i.i.d.}}{\sim} \mathcal{P}^m(\mu, \Lambda)$.*

The formal statement and proof of Theorem 4.5 can be found in Appendix G. We can see that the convergence rate of the inference error in multi-class classification w.r.t. $N$ and $M$ is similar to that in the binary classification, except for the constant coefficient $c$. This suggests that classification tasks with more classes may have higher errors than those with fewer classes. On the other hand, for data pair $(x, y) \sim \mathcal{P}^m(\mu, \Lambda)$, the distribution of $y$, $\mathbb{P}(y = \mathbf{e}_k | x) = \text{softmax}(\mu^\top \Lambda^{-1} x)_k$, can be characterized by a softmax regression model $\text{softmax}(Wx + b)$ with parameters $W = \mu^\top \Lambda^{-1}$ and $b = 0$. When $N, M \to \infty$, the prediction of the trained transformer is equivalent to the optimal softmax regressor for multi-class classification problems with distribution $\mathcal{P}^m(\mu, \Lambda)$. Note that different from Assumption 4.2 which states that $\mu_\tau, x_{\tau,\text{query}}$ are sampled according to some specific distributions during training, Assumption 4.4 does not impose strong distributional constraints on $\mu$ or $x_{\text{query}}$, which shows the strong generalization ability of the trained transformer. We also discuss the consequences when Assumption 4.4 does not hold in Remark G.2, which highlights the necessity of Assumption 4.4. Moreover, even if $M \to \infty$, the distribution variation between $y_{\text{query}}$ and $\widehat{y}_{\text{query}}$ does not disappear unless

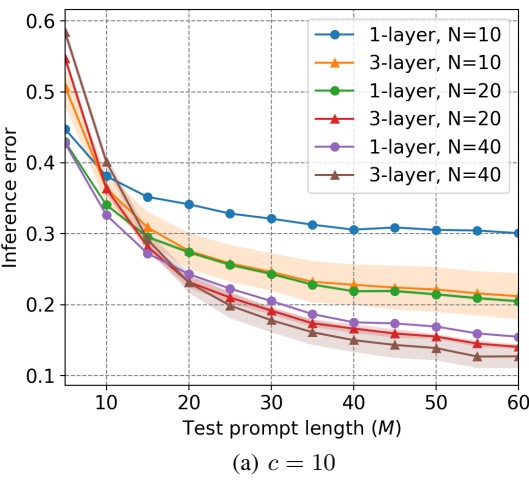
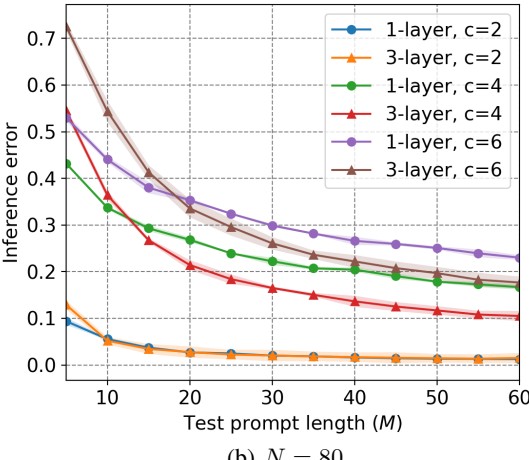

(a) $c = 10$            (b) $N = 80$

Figure 1. '1-layer': single-layer transformer defined in Section 4, '3-layer': 3-layer transformers with softmax attention. $N$: training prompt length. $c$: number of Gaussian mixtures.

$N \to \infty$. Thus, the ICL ability of the trained transformers for multi-class classification is limited by the finite length of training prompts. Similar behavior has also been observed in Zhang et al. (2023a) for in-context linear regression and in Section 3.2 for in-context binary classification.

## 5. Experiments

In this section, we report the experiment results on multi-layer, nonlinear transformers to investigate their similarities and differences to the single-layer, linear transformer we theoretically analyzed in the pervious sections. Detailed experimental settings and additional results can be found in Appendix H.

We train single-layer and multi-layer transformers for in-context classification of Gaussian mixtures with different numbers of Gaussian mixtures $c$, different lengths of training prompts $N$, and test them with different test prompt lengths $M$. The results are reported in Figure 1. We can see that for both single-layer and multi-layer transformers, the inference errors decrease as $N$ and $M$ increase, and they increase as $c$ increases, which not only verifies our theoretical claims but also shows that, the simplified model we have studied indeed exhibits behavioral similarities to the more complex multi-layer, nonlinear transformers, and some of our observations for this simplified model also hold for more complex transformers.

### 5.1. Varying covariances and norms

Note that in Assumption 3.2, 4.2, 3.5, 4.4, we assume that the covariance $\Lambda$ during pre-training and during inference are the same, and the means of all Gaussian components

$\{\mu_{\tau,i}, i \in [c]\}$ have the same $\Lambda^{-1}$ weighted norm. In Remark 3.7, G.2, we also discuss the situation when Assumptions 3.5, 4.4 do not hold and show the necessities of them. In this subsection, we consider training transformers with data of varying covariances $\Lambda$ and with Gaussian component means of unequal $\Lambda^{-1}$ weighted norms, and examine how these factors affect the ICL abilities of transformers. Results are shown in Figure 2.

From Figure 2 (a), we can see that both models perform better when their $\mu_{\tau,i}$ have the same $\Lambda^{-1}$ weighted norm ('same norm'), however, in the 'different norms' setting, the performance of '1-layer' deteriorates more significantly, while transformers with a more complex structure ('3-layer') show better robustness under this distribution shift. Similar situations also happen in Figure 2 (b), where '3-layer' also shows better tolerance to the covariance shifts than '1-layer'.

Experimental results in Figure 2 show the necessities of Assumptions 3.2, 4.2, 3.5, 4.4 for the single-layer transformers considered in this paper, and also demonstrate the better robustness of multi-layer, nonlinear transformers. Developing a better understanding of the robustness of more complex transformers is an intriguing direction for future research.

### 5.2. Comparison of transformers with other ML algorithms

Additionally, we conduct experiments comparing the ICL performances of the transformers with other machine learning algorithms for the classification of three Gaussian mixtures. From Figure 3, we can see that all three transformer models significantly outperform the classical methods (softmax regression, linear discriminant analysis), demonstrating the strong ICL capacities of transformers.

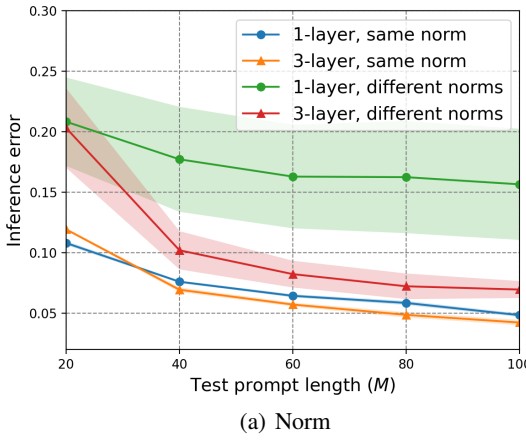

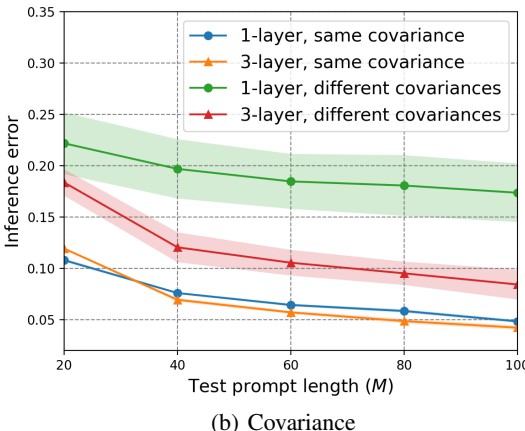

(a) Norm                               (b) Covariance

*Figure 2.* We generate $\Lambda_i = \mathrm{diag}(\lambda_{i1}, \ldots, \lambda_{id})$, $i \in \{0, 1, 2, 3\}$, where $\lambda_{ij} = |\hat{\lambda}_{ij}|$ and $\hat{\lambda}_{ij} \overset{\text{i.i.d.}}{\sim} \mathsf{N}(3, 1)$. All models are trained with prompt length $N = 100$, tested with prompts satisfying Assumption 4.4 with $\Lambda_0$. $c = 3$. (a) 'same norm': pre-training data are sampled according to Assumption 4.2 with $\Lambda_0$. 'different norms': For each $\tau$, with probability $\mathbb{P}(k = j) = 1/10, \mu_{\tau,i} \sim \mathsf{N}(k, I_d), j = 0, 1, \ldots, 9$. (b) 'same covariance': pre-training data are sampled according to Assumption 4.2 for the fixed $\Lambda_0$. 'different covariances': Sample additional $\Lambda_1, \Lambda_2, \Lambda_3$. Then, generate pre-training data according to Assumption 4.2 with $\Lambda_0, \Lambda_1, \Lambda_2, \Lambda_3$.

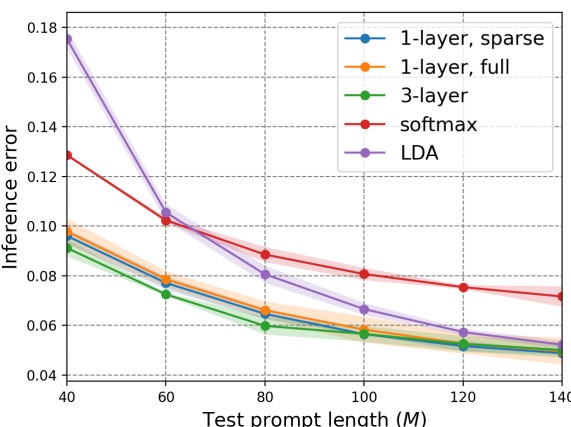

*Figure 3.* '1-layer, sparse': single-layer transformer defined in Section 4, '1-layer, full': single-layer transformer with full parameters (59), '3-layer': a 3-layer transformer with softmax attention, 'softmax': softmax regression, 'LDA': linear discriminant analysis. All three transformers are trained with prompt length $N = 100$.

## 6. Conclusion

We studied the learning dynamics of transformers for in-context classification of Gaussian mixtures, and showed that with properly distributed data, a single-layer transformer trained via gradient descent converges to its global minimum. Moreover, we established the upper bounds of the inference errors of the trained transformers and discussed how the training and test prompt lengths influence the performance of the model. Experimental results also corroborated the theoretical claims. There are some directions worth further exploring. One potential avenue is to investigate whether the assumptions regarding the training and test prompts can be relaxed. Additionally, we have only ex-

amined single-layer transformers with linear attention and sparse parameters. The learning dynamics of multi-layer transformers with nonlinear attention (e.g., softmax) for in-context classification problems remain an interesting area for future investigation.

## Acknowledgements

The work of Wei Shen and Cong Shen was supported in part by the US National Science Foundation (NSF) under awards ECCS-2033671, ECCS-2143559, CPS-2313110, and CNS-2002902. The work of Ruida Zhou was supported in part by NSF grants 2139304, 2146838 and the Army Research Laboratory grant under Cooperative Agreement W911NF-17-2-0196. The work of Jing Yang was supported in part by NSF award ECCS-2531023.

## Impact Statement

This paper presents work whose goal is to advance the field of Machine Learning. There are many potential societal consequences of our work, none of which we feel must be specifically highlighted here.

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

# Appendix

The Appendix is organized as follows. In Section A, we provide a literature review of the related works that studied the ICL abilities of transformers. In Section B, we introduce the additional notations for the proofs in the Appendix. In Section C, we introduce some useful Lemmas we adopt from previous literature. In Sections D, E, F, G, we present the proofs of Theorem 3.3, 3.6, 4.3, 4.5 respectively. In Section H, we provide additional results and details of our experiments.

## A. Related work

It has been observed that transformer-based models have impressive ICL abilities in natural language processing (Brown et al., 2020; Nye et al., 2021; Wei et al., 2022; Dasgupta et al., 2022; Zhang et al., 2022). Garg et al. (2022) first initiated the study of the ICL abilities of transformers in a mathematical framework and they empirically showed that transformers can in-context learn linear regression, two-layer ReLU networks, and decision trees. Subsequently, numerous works have been developed to explain the ICL capacities of transformers in solving in-context mathematical problems. These works mainly use two approaches: constructing specific transformers capable of performing certain in-context learning tasks, and studying the training dynamics of transformers for such tasks.

**Constructions of transformers.** Akyürek et al. (2022); Von Oswald et al. (2023) showed by construction that multi-layer transformers can be viewed as multiple steps of gradient descent for linear regression. Akyürek et al. (2022) also showed that constructed transformers can implement closed-form ridge regression. Guo et al. (2023) showed that constructed transformers can perform in-context learning with representations. Bai et al. (2024) proved that constructed transformers can perform various statistical machine learning algorithms through in-context gradient descent and showed that constructed transformers can perform in-context model selection. Lin et al. (2023) demonstrated that constructed transformers can approximate several in-context reinforcement learning algorithms. Fu et al. (2023); Giannou et al. (2024) further proved that constructed transformers can perform higher-order optimization algorithms like Newton's method. Pathak et al. (2023) showed that transformers can learn mixtures of linear regressions. Giannou et al. (2023) proved that looped transformers that can emulate various in-context learning algorithms. Cheng et al. (2023) showed that transformers can perform functional gradient descent for learning non-linear functions in context. Zhang et al. (2024) showed that a linear attention layer followed by a linear layer can learn and encode a mean signal vector for in-context linear regression.

**Training dynamics of transformers.** Mahankali et al. (2023); Ahn et al. (2024) proved that the global minimizer of the in-context learning loss of linear transformer can be equivalently viewed as one-step preconditioned gradient descent for linear regression. Zhang et al. (2023a) proved the convergence of gradient flow on a single-layer linear transformer and discussed how training and test prompt length will influence the prediction error of transformers for linear regression. Huang et al. (2023) proved the convergence of gradient descent on a single-layer transformer with softmax attention with certain orthogonality assumptions on the data features. Li et al. (2023c) showed that trained transformers can learn topic structure. Wu et al. (2023) analyzed the task complexity bound for pretraining single-layer linear transformers on in-context linear regression tasks. Tarzanagh et al. (2023) built the connections between single-layer transformers and support vector machines (SVMs). Nichani et al. (2024) showed that transformers trained via gradient descent can learn causal structure. Chen et al. (2024) proved the convergence of gradient flow on a multi-head softmax attention model for in-context multi-task linear regression. Kim & Suzuki (2024); Yang et al. (2024) proved that trained transformers can learn nonlinear features in context. Bu et al. (2024) studied the training convergence of transformer for tasks encoded with multiple cross-concept semantics. Li et al. (2024b) proved that a one-layer transformer can be trained to learn one-nearest neighbor for binary classification in context. Li et al. (2024a) studied the training dynamics of a single layer transformer for in-context classification problems. However, they only studied the binary classification tasks with *finite* patterns. They generated their data as $x = \mu_j + \kappa v_k$, where $\{\mu_j\}_{j=1}^{M_1}$ are in-domain-relevant patterns and $\{\nu_k\}_{k=1}^{M_2}$ are in-domain-irrelevant patterns, $M_1 \geq M_2$ and these patterns are all pairwise orthogonal. Thus, the possible distribution of their data is finite and highly limited. In contrast, our work explores the ICL capabilities of transformers for both binary and multi-class classification of Gaussian mixtures. Specifically, our data is drawn according to $\mathcal{P}^b(\mu_0, \mu_1, \Lambda)$ or $\mathcal{P}^m(\mu, \Lambda)$, and the range and possible distributions of our data are *infinite*. Recently, (Frei & Vardi, 2024) also studied the the ICL of a linear transformer model for classifying Gaussian mixtures and showed that trained transformers can exhibit benign overfitting in-context. However, their analysis relies on additional assumptions, such as a sufficiently large signal-to-noise ratio. In contrast, our work does not rely on such assumptions. Additionally, compared to (Frei & Vardi, 2024), we consider a more general multi-class setting. We believe these distinctions highlight the independent contributions of our work.

Some works also studied the ICL from other perspectives. To name a few, Xie et al. (2021) explained the ICL as implicit

Bayesian inference; Wang et al. (2023) explained the LLMs as latent variable models; Zhang et al. (2023b) explained the ICL abilities of transformers as implicitly implementing a Bayesian model averaging algorithm; and Li et al. (2023b) studied the generalization and stability of the ICL abilities of transformers. Hahn & Goyal (2023) showed that ICL can arise through recombination of compositional structure found in linguistic data. Lin & Lee (2024) studied the dual operating modes for in-context classification of Gaussian mixtures. However, the analyses of both Hahn & Goyal (2023) and Lin & Lee (2024) are based on the idealized optimal next-token predictor for ICL tasks and did not discuss the training dynamics of transformer models.

Apart from studies focusing on the training convergence properties of transformers in in-context learning (ICL), many other works have investigated the convergence behavior of transformers when learning different tasks. For example, (Jelassi et al., 2022) and (Li et al., 2023a) theoretically analyzed how Vision Transformers learn and converge. (Wang et al., 2024) showed that transformers can provably learn sparse token selection, whereas fully-connected networks cannot. (Zhang et al., 2025) demonstrated that transformers can learn optimal variable selection in group-sparse classification. (Gao et al., 2024) studied the global convergence of large-scale transformers under gradient flow dynamics.

## B. Additional notations

We denote $X \sim \mathsf{Bin}(n, p)$ if a random variable $X$ follows the binomial distribution with parameters $n \in \mathbb{N}$ and $p \in [0, 1]$, which means $\mathbb{P}(X = k) = \frac{n!}{k!(n-k)!} p^k (1-p)^{n-k}$. We denote $X \sim \mathsf{Multin}(n, p)$ if random variables $X = (X_1, X_2, \ldots, X_k)$ follow the Multinomial distribution with parameters $n \in \mathbb{N}$ and $p_1 = p_2 = \cdots = p_k = 1/k$, which means $\mathbb{P}(X = (x_1, x_2, \ldots, x_k)) = \frac{n!}{\prod_{i=1}^{k} x_k!} k^{-n}$. We denote $\zeta_i(x) = \mathrm{softmax}(x)_i = \exp(x_i)/(\sum_{j=1}^{k} \exp(x_j))$ for simplicity. We define $\delta_{ii} = 1, \delta_{ij} = 0, i \neq j$. For $x \in \mathbb{N}$, we define $t_1(x) = \lfloor (x-1)/d \rfloor + 1, t_2(x) = ((x-1) \bmod d) + 1$.

## C. Useful lemmas

**Lemma C.1** ((Karimi et al., 2016)). *If $f : \mathbb{R}^d \to \mathbb{R}$ is $\mu$-strongly convex, then*

$$f(x) - \min_x f(x) \geq \frac{\mu}{2} \|x^* - x\|_2^2$$

*where $x^* = \mathrm{argmin}_x f(x)$.*

**Lemma C.2** ((Bubeck, 2015)). *Suppose $f : \mathbb{R}^d \to \mathbb{R}$ is $\alpha$-strongly convex and $\beta$-smooth for some $0 < \alpha \leq \beta$. Then, the gradient descent iterating $w^{t+1} = w^t - \eta \nabla f(w^t)$ with learning rate $\eta = 1/\beta$ and initialization $w^0 \in \mathbb{R}^d$ satisfies that for any $t \geq 1$,*

$$\|w^t - w^*\|_2^2 \leq \exp(-t/\kappa) \|w^0 - w^*\|_2^2$$

*where $\kappa = \beta/\alpha$ is the condition number of $f$, and $w^* = \mathrm{argmin}_{w \in \mathbb{R}^d} f(w)$ is the minimizer of $f$.*

## D. Training procedure for in-context binary classification

In this section, we present the proof of Theorem 3.3.

### D.1. Proof sketch

First, we prove in Lemma D.2 that the expected loss function $L(W)$ in (7) is strictly convex w.r.t. $W$ and is strongly convex in a compact set of $\mathbb{R}^{d \times d}$. Moreover, we prove $L(W)$ has one unique global minimizer $W^*$. Then, in Lemma D.3, by analyzing the Taylor expansion of $L(W)$, we prove that as $N \to \infty$, our loss function $L(W)$ point wisely converges to $\widetilde{L}(W)$ (defined in (25)), and the global minimizer $W^*$ converge to $2\Lambda^{-1}$. We denote $W^* = 2(\Lambda^{-1} + G)$, and prove $\|G\|_{\max} = O(N^{-1/2})$. Next, in Lemma D.4, by further analyzing the Taylor expansion of the equation $\nabla L(W^*) = 0$ at the point $2\Lambda^{-1}$, we establish a tighter bound $\|G\|_{\max} = O(N^{-1})$. In Lemma D.5, we prove that our loss function is $l$-smooth and provide an upper bound for $l$. Thus, in a compact set $R_W$, our loss function is $\alpha$-strongly convex and $l$-smooth. Finally, leveraging the standard results from convex optimization, we prove Theorem 3.3 in subsection D.4.

In this section, we use the following notations.

### D.2. Notations

Recall the expected loss function (7) is

$$L(W) = -\frac{1}{2}\mathbb{E}\left[(1 + y_{\tau,\text{query}})\log(\widehat{y}_{\tau,\text{out}}) + (1 - y_{\tau,\text{query}})\log(1 - \widehat{y}_{\tau,\text{out}})\right], \tag{21}$$

where

$$\widehat{y}_{\tau,\text{out}} = \sigma\left(\left(\frac{2}{N}\sum_{i=1}^{N} y_{\tau,i}x_{\tau,i}^\top\right)\frac{W}{2}x_{\tau,\text{query}}\right)$$

is the output of the transformer, and the label of the data follows the distribution

$$\mathbb{P}\left(y_{\tau,\text{query}} = 1|x_{\tau,\text{query}}\right) = \sigma((\mu_{\tau,1} - \mu_{\tau,0})^\top \Lambda^{-1}x_{\tau,\text{query}})).$$

In this section, we introduce the following notations to analyze (7). We denote $\mu = \mu_{\tau,1} - \mu_{\tau,0}$, $\mu_1 = \mu_{\tau,1}$, $\mu_0 = \mu_{\tau,0}$ and $q = x_{\tau,\text{query}}$. Then with probability $\mathbb{P}\left(y_{\tau,\text{query}} = 1\right) = 1/2$ we have $q = \mu_1 + v$, and with probability $\mathbb{P}\left(y_{\tau,\text{query}} = 0\right) = 1/2$ we have $q = \mu_0 + v$, where $v \sim \mathsf{N}(0, \Lambda)$. We define $p = \frac{2}{N}\sum_{i=1}^{N} y_{\tau,i}x_{\tau,i}$. Since with probability $\mathbb{P}\left(y_{\tau,i} = 1\right) = 1/2$ we have $x_{\tau,i} = \mu_1 + v_i$, and with probability $\mathbb{P}\left(y_{\tau,i} = 0\right) = 1/2$ we have $x_{\tau,i} = \mu_0 + v_i$, where $v_i \sim \mathsf{N}(0, \Lambda)$, we known $p = 2N_1\mu_1/N - 2N_0\mu_0/N + g$, where $g = \frac{2}{N}\sum_{i=1}^{N} v_i$, $g \sim \mathsf{N}(0, 4\Lambda/N)$, $N_1 \sim \mathsf{Bin}(N, 1/2)$. Defining $h = N_1/N - 1/2$, $u = 2(\mu_1 + \mu_0)$, we have $N_0/N = 1/2 - h$ and

$$p = \mu + hu + g. \tag{22}$$

Then, the expected loss function (7) can be expressed as

$$L(W) = \mathbb{E}[-\sigma(\mu^\top \Lambda^{-1}q)\log(\sigma(p^\top Wq/2)) - (1 - \sigma(\mu^\top \Lambda^{-1}q))\log(1 - \sigma(p^\top Wq/2))]. \tag{23}$$

The gradient of the loss function (7) can be expressed as

$$\nabla L(W) = \frac{1}{2}\mathbb{E}[(\sigma(p^\top Wq/2) - \sigma(\mu^\top \Lambda^{-1}q))pq^\top]. \tag{24}$$

Moreover, we define a function $\widetilde{L}(W)$ as

$$\widetilde{L}(W) = \mathbb{E}[-\sigma(\mu^\top \Lambda^{-1}q)\log(\sigma(\mu^\top Wq/2)) - (1 - \sigma(\mu^\top \Lambda^{-1}q))\log(1 - \sigma(\mu^\top Wq/2))]. \tag{25}$$

In Lemma D.3, we show that as $N \to \infty$, $L(W)$ will point wisely converge to $\widetilde{L}(W)$.

### D.3. Lemmas

**Lemma D.1.** *Suppose $N_1 \sim \mathsf{Bin}(N, 1/2)$. Defining $h = N_1/N - 1/2$, we have*

$$\mathbb{E}[h] = 0$$
$$\mathbb{E}[h^2] = \frac{1}{4N}$$
$$\mathbb{E}[h^3] = 0$$
$$\mathbb{E}[h^n] = O(N^{-2}), \text{ for } n \geq 4$$
$$\mathbb{E}[|h|] \leq \frac{1}{2N^{1/2}}$$
$$\mathbb{E}[|h^3|] = O(N^{-3/2}).$$

*Proof.* Since $N_1 \sim \mathsf{Bin}(N, 1/2)$, the moment-generating function of $N_1$ is

$$M_{N_1}(t) = \left(\frac{1}{2} + \frac{1}{2}\exp(t)\right)^N.$$

We can compute the moment-generating function of $h$ as follows:

$$M_h(t) = \exp\left(-\frac{t}{2}\right) M_{N_1}\left(\frac{t}{N}\right) = \left(\frac{\exp\frac{-t}{2N} + \exp\frac{t}{2N}}{2}\right)^N = \left(\cosh\left(\frac{t}{2N}\right)\right)^N$$

$$= \left(1 + \frac{t^2}{8N^2} + \sum_{i=2}^{\infty}\frac{t^{2i}}{(2i)!(2N)^{2i}}\right)^N.$$

Thus, we know the coefficients of $t$, $t^2$, $t^3$ are $0, 1/(8N), 0$ respectively, and the coefficients of $t^n, n \geq 4$ are $O(1/N^2)$. We have

$$\mathbb{E}[h] = 0$$
$$\mathbb{E}[h^2] = \frac{1}{4N}$$
$$\mathbb{E}[h^3] = 0$$
$$\mathbb{E}[h^n] = O(1/N^2), \text{ for } n \geq 4.$$

Moreover, according to the Jensen's inequality, we have

$$\mathbb{E}[|h|] \leq \left(\mathbb{E}[h^2]\right)^{1/2} = \frac{1}{2N^{1/2}}$$
$$\mathbb{E}[|h^3|] \leq \left(\mathbb{E}[h^4]\right)^{3/4} = O(N^{-3/2}).$$

$\square$

**Lemma D.2.** *For the loss function $L(W)$ (7), we have $\nabla^2 L(W) \succ 0$. For any compact set $R_W$ of $\mathbb{R}^{d \times d}$, when $W \in R_W$, we have $\nabla^2 L(W) \succ \gamma I_d$ for some $\gamma > 0$. Additionally, $L(W)$ has one unique global minimizer on $\mathbb{R}^{d \times d}$.*

*For $\widetilde{L}(W)$ defined in (25), we also have $\nabla^2\widetilde{L}(W) \succ 0$. For any compact set $R_W$ of $\mathbb{R}^{d \times d}$, when $W \in R_W$, we have $\nabla^2\widehat{L}(W) \succ \gamma I_d$ for some $\gamma > 0$. Additionally, $\widetilde{L}(W)$ has one unique global minimizer on $\mathbb{R}^{d \times d}$.*

*Proof.* We vectorize $W$ as $\mathrm{Vec}(W) \in \mathbb{R}^{d^2}$, where $\mathrm{Vec}(W)_i = W_{t_1(i),t_2(i)}$, $t_1(x) = \lfloor(x-1)/d\rfloor + 1, t_2(x) = ((x-1) \bmod d) + 1$. Then, we have

$$(\nabla L(W))_i = \mathbb{E}_{p,q}\left[\frac{1}{2}(\sigma(p^\top W q/2) - \sigma(\mu^\top \Lambda^{-1} q))p_{t_1(i)}q_{t_2(i)}\right]. \tag{26}$$

The Hessian matrix of the loss function (7) is

$$(\nabla^2 L(W))_{ij} = \mathbb{E}_{p,q}\left[\frac{1}{4}\sigma(p^\top W q/2)(1 - \sigma(p^\top W q/2))p_{t_1(i)}q_{t_2(i)}p_{t_1(j)}q_{t_2(j)}\right].$$

Considering $z \in \mathbb{R}^{d^2}$ such that $z \neq 0$, we have

$$z^\top \nabla^2 L(W)z = \mathbb{E}_{q,p}\left[\frac{1}{4}\sigma(p^\top W q/2)(1 - \sigma(p^\top W q/2))\sum_{ab}z_a z_b p_{t_1(a)}q_{t_2(a)}p_{t_1(b)}q_{t_2(b)}\right]$$

$$= \int \frac{1}{4}\sigma(p^\top W q/2)(1 - \sigma(p^\top W q/2))\left(\sum_{a \in [d^2]}z_a p_{t_1(a)}q_{t_2(a)}\right)^2 f_{pq}(p,q)dpdq,$$

where $f_{pq}(p,q)$ are the probability density function (PDF) function of $p, q$. Since for any $p, q$, $\sigma(p^\top W q/2)(1 - \sigma(p^\top W q/2)) > 0$, we have $z^\top \nabla^2 L(W)z \geq 0$. Thus, $\nabla^2 L(W) \succeq 0$ and $L(W)$ is convex.

Moreover, for any $z \neq 0$, we denote $z_{ij} = z_{((i-1)d+j)}, i, j \in [d]$. Suppose $a, b \in \arg\max_{i,j} |z_{ij}|$, we consider a set of constants $\{c_{1pi}, c_{2pi}\}, \{c_{1qi}, c_{2qi}\}, i, j \in [d]$, where $c_{1pa} = d, c_{2pa} = d + 1, c_{1qb} = d, c_{2qb} = d + 1$, and $c_{1pi} = 1/16, c_{2pi} = 1/8, i \neq a, c_{1qj} = 1/16, c_{2qj} = 1/8, j \neq b$. Then, for any $c_{pi} \in [c_{1pi}, c_{2pi}], c_{qj} \in [c_{1qj}, c_{2qj}]$. We have

$$\left| \sum_{i,j\in[d]} z_{ij} c_{pi} c_{qj} \right| \geq \left[ d^2 - 2(d+1)(d-1)/8 - (d-1)^2/64 \right] \max_{ij} |z_{ij}| \geq d^2 \max_{ij} |z_{ij}|/2.$$

Then, we define region $\Omega(a, b) \triangleq \{p = \sum_i c_{pi} \mathbf{e}_i, q = \sum_j c_{qj} \mathbf{e}_j, c_{pi} \in [c_{1pi}, c_{2pi}], c_{qj} \in [c_{1qj}, c_{2qj}]\}$. We have

$$\min_{\Omega(a,b)} \left( \sum_{c\in[d^2]} z_c p_{t_1(c)} q_{t_2(c)} \right)^2 \geq d^4 \max_{ij} |z_{ij}|^2/4 \geq \|z\|_2^2/4.$$

Defining

$$C(\Omega) = \min_{a\in[d],b\in[d]} \int_{\Omega(a,b)} f_{pq}(p,q)dpdq,$$

$$S(\Omega, W) = \min_{a\in[d],b\in[d]} \min_{\Omega(a,b)} \left\{ \frac{1}{4}\sigma(p^\top Wq/2)(1 - \sigma(p^\top Wq/2)) \right\},$$

we have $S(\Omega, W) > 0$. Since with probability $\mathbb{P}(y_{\tau,\text{query}} = 1) = 1/2, q = \mu_1 + v$, with probability $\mathbb{P}(y_{\tau,\text{query}} = 0) = 1/2$, $q = \mu_0 + v$, where $v \sim \mathsf{N}(0, \Lambda)$ and $p = \mu + hu + g$, where $g \sim \mathsf{N}(0, 4\Lambda/N), v \sim \mathsf{N}(0, \Lambda), \mu_0 \sim \mathsf{N}(0, I_d)$, the covariance matrices of $p, q$ are positive definite and we have $f_{pq}(p, q) > 0$ for all $p, q \in \mathbb{R}^d$. Moreover, $\Omega(a, b)$ are non-zero measures on $\mathbb{R}^{d\times d}$. Thus, we have $C(\Omega) > 0$. Then, for any $z \neq 0$, we have

$$z^\top \nabla^2 L(W)z \geq \int_{\Omega(a,b)} \frac{1}{4}\sigma(p^\top Wq/2)(1 - \sigma(p^\top Wq/2)) \left( \sum_l z_l p_{t_1(l)} q_{t_2(l)} \right)^2 f_{pq}(p,q)dpdq$$

$$\geq C(\Omega)S(\Omega, W)\|z\|_2^2/4$$

$$> 0.$$

Thus, we have $\nabla^2 L(W) \succ 0$. $L(W)$ is strictly convex.

Moreover, for any compact set $R_W$ of $\mathbb{R}^{d\times d}$, for any $W \in R_W$, we have

$$S(\Omega) = \min_{W\in R_W} \min_{a\in[d],b\in[d]} \min_{\Omega(a,b)} \left\{ \frac{1}{4}\sigma(p^\top Wq/2)(1 - \sigma(p^\top Wq/2)) \right\} > 0.$$

Then, for any $W \in R_W$, for any $z \neq 0$, we have

$$z^\top \nabla^2 L(W)z \geq \int_{\Omega(a,b)} \frac{1}{4}\sigma(p^\top Wq/2)(1 - \sigma(p^\top Wq/2)) \left( \sum_l z_l p_{t_1(l)} q_{t_2(l)} \right)^2 f_{pq}(p,q)dpdq$$

$$\geq \frac{1}{4}C(\Omega)S(\Omega)\|z\|_2^2.$$

Thus, when $W \in R_W$, where $R_W$ is a compact set, we have $\nabla^2 L(W) \succ C(\Omega)S(\Omega)I_d/4$ and the loss function $L(W)$ is $\gamma-$strongly convex, where $\gamma = C(\Omega)S(\Omega)/4$.

Because our loss function is strictly convex in $\mathbb{R}^{d\times d}$, it has at most one global minimizer in $\mathbb{R}^{d\times d}$. Next, we prove all level sets of our loss function are compact, i.e. $V_\alpha = \{W \in \mathbb{R}^{d\times d} \mid L(W) \leq \alpha\}$ is compact for all $\alpha$. We prove it by contradiction. Suppose $V_\alpha$ is not compact for some $\alpha$. Since our loss function is continuous and convex, $V_\alpha$ is an unbounded convex set. Since the dimension of $V_\alpha$ is $d^2$, consider a point $W^\alpha \in V_\alpha$, there must exists a $W^k \neq 0_{d\times d}$ such that $\{W^\alpha + tW^k \mid t = [0, \infty)\} \in V_\alpha$. For this $W^k \neq 0_{d\times d}$, there must exist a set of constants $0 < c_{3pi} < c_{4pi}, 0 < c_{3qj} < c_{4qj}$ such that for any $c_{pi} \in [c_{3pi}, c_{4pi}], c_{qj} \in [c_{3qj}, c_{4qj}]$, we have

$$\left| \sum_{ij} c_{pi} c_{qj} W_{ij}^k \right| \neq 0.$$

Thus, we have

$$\lim_{t \to \infty} |\sum_{ij} c_{pi} c_{qj} (W_{ij}^\alpha + t W_{ij}^k)| = \infty.$$

We define $\Omega_0 = \{p = \sum_i c_{pi} \mathbf{e}_i, q = \sum_j c_{qj} \mathbf{e}_j, c_{pi} \in [c_{3pi}, c_{4pi}], c_{qj} \in [c_{3qj}, c_{4qj}], \|\mu\|_2^2 \le \sum_i c_{4pi}^2 + c_{4qj}^2\}$. Then, defining

$$C(\Omega_0) = \int_{\Omega_0} f_{pq}(p, q) dp dq,$$
$$S(\Omega_0) = \min_{\Omega_0} \left\{ \min\{\sigma(\mu^\top \Lambda^{-1} q), (1 - \sigma(\mu^\top \Lambda^{-1} q))\} \right\},$$

we have $S(\Omega_0) > 0$. Since $\Omega_0$ are non-zero measures for $p, q$, we have $C(\Omega_0) > 0$. Then, we have

$$\lim_{t \to \infty} L(W^\alpha + t W^k)$$
$$= \lim_{t \to \infty} \mathbb{E}[-\sigma(\mu^\top \Lambda^{-1} q) \log(\sigma(p^\top (W^\alpha + t W^k) q / 2)) - (1 - \sigma(\mu^\top \Lambda^{-1} q)) \log(1 - \sigma(p^\top (W^\alpha + t W^k) q / 2))]$$
$$\ge \lim_{t \to \infty} \int_{\Omega_0} [-\sigma(\mu^\top \Lambda^{-1} q) \log(\sigma(\sum_{ij} c_{pi} c_{qj} (W_{ij}^\alpha + t W_{ij}^k) / 2))] f_{pq}(p, q) dp dq$$
$$+ \lim_{t \to \infty} \int_{\Omega_0} [-(1 - \sigma(\mu^\top \Lambda^{-1} q)) \log(1 - \sigma(\sum_{ij} c_{pi} c_{qj} (W_{ij}^\alpha + t W_{ij}^k) / 2))] f_{pq}(p, q) dp dq$$
$$\ge C(\Omega_0) S(\Omega_0) \cdot \min_{\Omega_0} \left\{ \lim_{t \to \infty} [-\log(\sigma(\sum_{ij} c_{pi} c_{qj} (W_{ij}^\alpha + t W_{ij}^k) / 2))] \right\}$$
$$+ C(\Omega_0) S(\Omega_0) \cdot \min_{\Omega_0} \left\{ \lim_{t \to \infty} [-\log(1 - \sigma(\sum_{ij} c_{pi} c_{qj} (W_{ij}^\alpha + t W_{ij}^k) / 2))] \right\}$$
$$= \infty.$$

This contradicts the assumption $L(W^\alpha + t W^k) \le \alpha$. Thus, all level sets of the loss function $L(W)$ are compact, which means there exists a global minimizer for $L(W)$. Together with the fact that $L(W)$ is strictly convex, $L(W)$ has one unique global minimizer on $\mathbb{R}^{d \times d}$.

Similarly, we can prove the same conclusions for $\widetilde{L}(W)$. $\qquad\square$

**Lemma D.3.** *Denoting the global minimizer of the loss function* (7) *as* $W^*$, *we have* $W^* = 2(\Lambda^{-1} + G)$, *where* $\|G\|_{\max} = O(N^{-1/2})$.

*Proof.* Let $a = \mu^\top \Lambda^{-1} q$, $s = \mu^\top W q / 2$, $r = (hu + g)^\top W q / 2$. Performing the Taylor expansion on (7), we have

$$L(W) = \mathbb{E}\left[-\sigma(a) \log(\sigma(s + r)) - (1 - \sigma(a)) \log(1 - \sigma(s + r))\right]$$
$$= \mathbb{E}\left[-\sigma(a) \log(\sigma(s)) - (1 - \sigma(a)) \log(1 - \sigma(s))\right]$$
$$- \mathbb{E}\left[(\sigma(a)(1 - \sigma(s)) - (1 - \sigma(a))\sigma(s)) r\right]$$
$$+ \mathbb{E}\left[\sigma(\xi(s, r))(1 - \sigma(\xi(s, r))) r^2 / 2\right]$$
$$= \widetilde{L}(W) - \mathbb{E}\left[(\sigma(a)(1 - \sigma(s)) - (1 - \sigma(a))\sigma(s)) r\right]$$
$$+ \mathbb{E}\left[\sigma(\xi(s, r))(1 - \sigma(\xi(s, r))) r^2 / 2\right],$$

where $\xi(s, r)$ are real numbers between $s$ and $s + r$. According to Lemma D.1, we have $\mathbb{E}[r] = \mathbb{E}\left[(hu + g)^\top W q / 2\right] = 0$. Thus, we have

$$\mathbb{E}\left[(\sigma(a)(1 - \sigma(s)) - (1 - \sigma(a))\sigma(s)) r\right] = \mathbb{E}_{\mu, u, q}\left[(\sigma(a)(1 - \sigma(s)) - (1 - \sigma(a))\sigma(s)) \mathbb{E}_{g, h}[r]\right] = 0.$$

Moreover, we have

$$
\begin{aligned}
&\mathbb{E}\left[\sigma(\xi(s,r))(1-\sigma(\xi(s,r))r^2/2\right] \\
\leq &\mathbb{E}\left[r^2\right] \\
= &\mathbb{E}[h^2 u^\top W q u^\top W q + g^\top W q g^\top W q] \\
\overset{(a)}{=} &\mathbb{E}[u^\top W q u^\top W q/(4N) + 4(\Lambda W q)^\top W q/N] \\
\leq &C_l \|W\|_{\max}^2/N,
\end{aligned}
$$

where $(a)$ is due to Lemma D.1, $g^\top W q g^\top W q = \sum_{i,j,k,l\in[d]} g_i W_{ij} q_j g_k W_{kl} q_l = \sum_{i,j,k,l\in[d]} g_i g_k W_{kl} q_l W_{ij} q_j = (gg^\top W q)^\top W q$ and $\mathbb{E}[gg^\top] = 4\Lambda/N$. $C_l$ is a constant independent of $N$ and $W$. Thus, we have

$$
\left|\widetilde{L}(W) - L(W)\right| \leq C_l \|W\|_{\max}^2/N.
$$

This shows that $L(W)$ point wisely converges to $\widetilde{L}(W)$.

According to Lemma D.2, $\widetilde{L}(W)$ has one unique global minimizer. Consider the equation:

$$
\nabla\widetilde{L}(W) = \mathbb{E}[\sigma(\mu^\top W q/2) - \sigma(\mu^\top \Lambda^{-1} q)] = 0.
$$

We can easily find that $\nabla\widetilde{L}(2\Lambda^{-1}) = 0$ and $W = 2\Lambda^{-1}$ is the global minimizer of $\widetilde{L}(W)$.

Considering a compact set $R_W = \{W \mid \|W - 2\Lambda^{-1}\|_F \leq \rho_W\}$, we have $\|W\|_{\max} \leq C_W$ for $W \in R_W$. Here $\rho_W, C_W$ are some positive finite constants. Then, we have

$$
\left|\widetilde{L}(W) - L(W)\right| \leq C_l'/N, \ W \in R_W,
$$

where $C_l' = C_l C_W^2$ is a constant independent of $N$ and $W$. This shows that, for $W \in R_W$, our loss function $L(W)$ uniformly converge to $\widetilde{L}(W)$.

Denote $W^*$ as the global minimizer of the loss function $L(W)$ with prompt length $N$. Then, we show that, when $N$ is sufficiently large, $W^* \in R_W$. We first denote $\partial R_W = \{W \mid \|W - 2\Lambda^{-1}\|_F = \rho_W\}$ and $\Delta = \min_{W\in\partial R_W} \widetilde{L}(W) - \widetilde{L}(2\Lambda^{-1}) > 0$. Then, for $N \geq 4C_l'/\Delta$, and for any $W \in R_W$, we have

$$
\left|\widetilde{L}(W) - L(W)\right| \leq \Delta/4,
$$

This means

$$
\begin{aligned}
&\min_{W\in\partial R_W} L(W) - \min_{W\in R_W} L(W) \\
\geq &\min_{W\in\partial R_W} L(W) - L(2\Lambda^{-1}) \\
\geq &\min_{W\in\partial R_W} \widetilde{L}(W) - \widetilde{L}(2\Lambda^{-1}) - \Delta/2 \\
\geq &\Delta/2 > 0.
\end{aligned}
$$

Since $L(W)$ is strictly convex, we have $W^* = \operatorname{argmin}_W L(W) \in R_W$.

Then, we have

$$
\begin{aligned}
|\widetilde{L}(W^*) - L(W^*)| &\leq C_l'/N \\
|\widetilde{L}(2\Lambda^{-1}) - L(2\Lambda^{-1})| &\leq C_l'/N
\end{aligned}
$$

$$
\widetilde{L}(W^*) \leq L(W^*) + C_l'/N \leq L(2\Lambda^{-1}) + C_l'/N \leq \widetilde{L}(2\Lambda^{-1}) + 2C_l'/N.
$$

According to Lemma D.2, for $W \in R_W$, we have $\nabla^2 \widetilde{L}(W) \succ \gamma I_d$, where $\gamma$ is a positive constant independent of $N$. Thus, $\widetilde{L}(W)$ is $\gamma$-strongly convex in $R_W$. According to Lemma C.1, we have

$$\|W^* - 2\Lambda^{-1}\|_F^2 \leq \frac{2}{\gamma}(\widetilde{L}(W^*) - \widetilde{L}(2\Lambda^{-1})) \leq \frac{4C_l'}{\gamma N}.$$

Thus, when $N \to \infty$, we have $W^* \to 2\Lambda^{-1}$. Denoting $W^* = 2(\Lambda^{-1} + G)$, we have $\|G\|_{\max} = O(1/\sqrt{N})$. $\qquad \square$

**Lemma D.4.** *The global minimizer of the loss function* (7) *is* $W^* = 2(\Lambda^{-1} + G)$*, where*

$$\begin{aligned}
\|G\|_{\max} \leq &\frac{1}{N}\|S^{-1}(\mathbb{E}[\sigma'(a)(4qq^\top + uu^\top\Lambda^{-1}qq^\top/4) \\
&+ \sigma''(a)((u^\top\Lambda^{-1}q)^2\mu q^\top/8 + 2q^\top\Lambda^{-1}q\mu q^\top)])\|_{\max} + o(1/N),
\end{aligned}$$

$a = \mu^\top\Lambda^{-1}q$, $S = 4\nabla^2\widetilde{L}(2\Lambda^{-1})$.

*Proof.* According to Lemma D.2, the loss function $L(W)$ has a unique global minimizer $W^*$. We have

$$\nabla L(W^*) = \mathbb{E}\left[(\sigma(p^\top W^* q/2) - \sigma(\mu^\top\Lambda^{-1}q))pq^\top\right] = 0. \tag{27}$$

Let $W^* = 2(\Lambda^{-1} + G)$, $a = \mu^\top\Lambda^{-1}q$, $b = (\mu + hu + g)^\top Gq + (hu + g)^\top\Lambda^{-1}q$. We have

$$\begin{aligned}
&p^\top W^* q/2 \\
=&(\mu + hu + g)^\top(\Lambda^{-1} + G)q \\
=&(\mu + hu + g)^\top Gq + (hu + g)^\top\Lambda^{-1}q + \mu^\top\Lambda^{-1}q = a + b.
\end{aligned}$$

The Taylor expansion of $\sigma(a + b)$ at point $a$ with an Lagrange form of remainder is

$$\sigma(a + b)pq^\top = \sigma(a)pq^\top + \sigma'(a)bpq^\top + \frac{\sigma''(a)}{2}b^2pq^\top + \frac{\sigma'''(\xi(a,b))}{3!}b^3pq^\top,$$

where $\xi(a, b)$ are real numbers between $a$ and $a + b$. Thus, our equation (27) become

$$\mathbb{E}_{\mu,u,g,h,q}\left[\sigma'(a)bpq^\top + \frac{\sigma''(a)}{2}b^2pq^\top + \frac{\sigma'''(\xi(a,b))}{3!}b^3pq^\top\right] = 0. \tag{28}$$

Note that $\mathbb{E}[\sigma'(a)bpq^\top] = \mathbb{E}_{\mu,u,q}\left[\sigma'(a)\mathbb{E}_{g,h}\left[bpq^\top\right]\right]$. For $\mathbb{E}_{g,h}[bpq^\top]$, according to Lemma D.1 and $g \sim \mathsf{N}(0, 4\Lambda/N)$, we have

$$\begin{aligned}
&\mathbb{E}_{g,h}[bpq^\top] \\
=&\mathbb{E}[\mu^\top Gq\mu q^\top + g^\top\Lambda^{-1}qgq^\top + g^\top Gqgq^\top + h^2u^\top Gquq^\top + h^2u^\top\Lambda^{-1}quq^\top] \\
=&\mu\mu^\top Gqq^\top + 4qq^\top/N + 4\Lambda Gqq^\top/N + uu^\top Gqq^\top/(4N) + uu^\top\Lambda^{-1}qq^\top/(4N). \tag{29}
\end{aligned}$$

Then, we have

$$\|\mathbb{E}_{\mu,u,q}[\sigma'(a)(4\Lambda Gqq^\top/N + uu^\top Gqq^\top/(4N))]\|_{\max} \leq c_1\|G\|_{\max}/N,$$

where $c_1 = \max_{ij}|\mathbb{E}\left[\sum_{kl} 4\sigma'(a)\left(\Lambda_{ik}q_lq_j\right) + \sum_{kl}\sigma'(a)\left(u_iu_kq_lq_j/4\right)\right]|$ is a constant independent of $N$. According to Lemma D.3, $\|G\|_{\max} = O(1/\sqrt{N}) = o(1)$, we have

$$\|\mathbb{E}_{\mu,u,q}[\sigma'(a)(4\Lambda Gqq^\top/N + uu^\top Gqq^\top/(4N))]\|_{\max} = o(1/N), \tag{30}$$

Similarly for $\mathbb{E}[\sigma''(a)b^2 pq^\top/2]$, we have

$$
\begin{aligned}
&\mathbb{E}_{g,h}[b^2 pq^\top] \\
=&\underbrace{\mathbb{E}[\mu^\top Gq\mu^\top Gq\mu q^\top + h^2 u^\top Gqu^\top Gq\mu q^\top + g^\top Gqg^\top Gq\mu q^\top + 2h^2 u^\top Gq\mu^\top Gquq^\top + 2g^\top Gq\mu^\top Gqgq^\top]}_{(i)} \\
&+\underbrace{\mathbb{E}[2h^2 u^\top Gqu^\top \Lambda^{-1}q\mu q^\top + 2g^\top Gqg^\top \Lambda^{-1}q\mu q^\top + 2h^2\mu^\top Gqu^\top \Lambda^{-1}quq^\top + 2\mu^\top Gqg^\top \Lambda^{-1}qgq^\top]}_{(ii)} \\
&+\underbrace{\mathbb{E}[h^2 u^\top \Lambda^{-1}qu^\top \Lambda^{-1}q\mu q^\top + g^\top \Lambda^{-1}qg^\top \Lambda^{-1}q\mu q^\top]}_{(iii)}.
\end{aligned}
$$

For each term in $(i)$, it contains two $G$. Thus, their max norms are at most smaller than $O(\|G\|_{\max}^2)$. For each term in $(ii)$, it contains one $G$ and $h^2$ or contains one $G$ and two $g$. According to $\mathbb{E}[h^2] = 1/(4N)$ in Lemma D.1, the max norm of terms with one $G$ and $h^2$ are smaller than $O(\|G\|_{\max}/N)$. Defining $\bar{g} = N^{1/2}\Lambda^{-1/2}g/2$, we have $\bar{g} \sim \mathsf{N}(0, I_d)$ and $g = 2N^{-1/2}\Lambda^{1/2}\bar{g}$. Thus, converting two $g$ to $\bar{g}$, we have a coefficient of $N^{-1}$. Therefore, the max norms of terms with one $G$ and two $g$ are also smaller than $O(\|G\|_{\max}/N)$. Therefore, for terms $(i), (ii)$, we have

$$\|\mathbb{E}[\sigma''(a)(i)/2]\|_{\max} \le O(\|G\|_{\max}^2) = o(\|G\|_{\max}), \tag{31}$$
$$\|\mathbb{E}[\sigma''(a)(ii)/2]\|_{\max} \le O(\|G\|_{\max}/N) = o(1/N). \tag{32}$$

For term $(iii)$, according to Lemma D.1 and $g \sim \mathsf{N}(0, 4\Lambda/N)$, we have

$$
\begin{aligned}
&\|\mathbb{E}[\sigma''(a)(iii)/2]\|_{\max} \\
=&\|\mathbb{E}\left[\sigma''(a)(h^2 u^\top \Lambda^{-1}qu^\top \Lambda^{-1}q\mu q^\top + g^\top \Lambda^{-1}qg^\top \Lambda^{-1}q\mu q^\top)/2\right]\|_{\max} \tag{33} \\
=&\frac{1}{N}\|\mathbb{E}\left[\sigma''(a)((u^\top \Lambda^{-1}q)^2\mu q^\top/8 + 2q^\top \Lambda^{-1}q\mu q^\top)\right]\|_{\max}. \tag{34}
\end{aligned}
$$

For $\mathbb{E}[\sigma'''(\xi(a,b))b^3 pq^\top/3!]$, we have

$$
\begin{aligned}
&\|\mathbb{E}[\sigma'''(\xi(a,b))b^3 pq^\top/3!]\|_{\max} \\
\le&\max_{z\in\mathbb{R}}|\sigma'''(z)|/3! \cdot \max_{ij}\mathbb{E}\left[|b^3 p_i q_j|\right] \\
\le&O(1)\cdot\max_{ij}\mathbb{E}\bigg[\underbrace{\sum_{\phi_1,\phi_2,\phi_3\in\{\mu,hu,g\}}\left|\phi_1^\top Gq\phi_2^\top Gq\phi_3^\top Gqp_i q_j\right|}_{(*)} \\
&+\underbrace{\sum_{\phi_1,\phi_2\in\{\mu,hu,g\},\phi_3\in\{hu,g\}}\left|\phi_1^\top Gq\phi_2^\top Gq\phi_3^\top \Lambda^{-1}qp_i q_j\right|}_{(*)} \\
&+\underbrace{\sum_{\phi_1\in\{\mu,hu,g\},\phi_2,\phi_3\in\{hu,g\}}\left|\phi_1^\top Gq\phi_2^\top \Lambda^{-1}q\phi_3^\top \Lambda^{-1}qp_i q_j\right|}_{(**)} \\
&+\underbrace{\sum_{\phi_1,\phi_2,\phi_3\in\{hu,g\}}\left|\phi_1^\top \Lambda^{-1}q\phi_2^\top \Lambda^{-1}q\phi_3^\top \Lambda^{-1}qp_i q_j\right|}_{(***)}\bigg].
\end{aligned}
$$

For terms in $(*)$ containing two or three $G$, these terms' expected absolute values are at most smaller than $O(\|G\|_{\max}^2)$. For terms in $(**)$ containing one $G$, these terms must contain $n_1$ number of $h$ and $n_2$ number of elements of $g$, where $n_1 + n_2 = 2, 3, 4, n_1, n_2 \in \mathbb{N}$. According to Lemma D.1, we know that for $n_1 = 1, 2, 3, 4$, $\mathbb{E}|h^{n_1}| \le O(N^{-n_1/2})$. Defining $\bar{g} = N^{1/2}\Lambda^{-1/2}g/2$, we have $\bar{g} \sim \mathsf{N}(0, I_d)$ and $g = 2N^{-1/2}\Lambda^{1/2}\bar{g}$. Converting $g$ to $\bar{g}$, we have a coefficient of $N^{-n_2/2}$. Thus, for terms in $(**)$, these terms' expected absolute values are at most smaller than $O(\|G\|_{\max}N^{-(n_1+n_2)/2}) \le$

$O(\|G\|_{\max}N^{-1})$. For terms in $(\ast\ast\ast)$ without $G$, these terms must contain $n_1$ number of $h$ and $n_2$ number of elements of $g$, we have $n_1 + n_2 = 3, 4, n_1, n_2 \in \mathbb{N}$. Similarly, these term's expected absolute values are at most smaller than $O(N^{-(n_1+n_2)/2}) \le O(N^{-3/2})$. Therefore, we have

$$
\begin{aligned}
&\|\mathbb{E}[\sigma'''(\xi(a,b))b^3 pq^\top/3!]\|_{\max} \\
&\le \max_{ij} \mathbb{E}\left[|b^3 p_i q_j|\right] \cdot \max_z |\sigma'''(z)|/3! \\
&= O(\|G\|_{\max}^2) + O(\|G\|_{\max}/N) + O(1/N^{-3/2}) \\
&= o(\|G\|_{\max}) + o(1/N).
\end{aligned}
\tag{35}
$$

Moreover, we have

$$
\left\{\mathbb{E}_{\mu,u,q}[\sigma'(a)\mu\mu^\top Gqq^\top]\right\}_{ij} = \sum_{kl} s_{ijkl}G_{kl},
\tag{36}
$$

where $s_{ijkl} = \mathbb{E}\sigma'(a)\mu_i \mu_k q_l q_j$. We vectorize $G$ as $\text{Vec}(G)_i = G_{t_1(i),t_2(i)}$. Define $S \in \mathbb{R}^{d^2 \times d^2}$, where $S_{ij} = s_{t_1(i),t_2(i),t_1(j),t_2(j)} = \mathbb{E}\sigma'(a)\mu_{t_1(i)}q_{t_2(i)}\mu_{t_1(j)}q_{t_2(j)}$. Then (36) can be expressed as

$$
\left\{\mathbb{E}_{\mu,v}[\sigma'(a)\mu\mu^\top Gqq^\top]\right\} = SG.
\tag{37}
$$

Note that $S = 4\nabla^2 \widetilde{L}(2\Lambda^{-1})$. According to Lemma D.2, $S$ is positive definite. Thus, combining (28), (29), (30), (31), (32), (34), (35), (37), we have

$$
\begin{aligned}
&\|G\|_{\max} \\
&\le \frac{1}{N}\|S^{-1}\left(\mathbb{E}[\sigma'(a)(4qq^\top + uu^\top\Lambda^{-1}qq^\top/4) + \sigma''(a)((u^\top\Lambda^{-1}q)^2\mu q^\top/8 + 2q^\top\Lambda^{-1}q\mu q^\top)]\right)\|_{\max} \\
&\quad + o(1/N).
\end{aligned}
$$

$\square$

**Lemma D.5.** *The loss function* (7) *is* $l$*-smooth, where* $l \le \frac{1}{4}\sum_{i \in [d^2]} \mathbb{E}[(p_{t_1(i)}q_{t_2(i)})^2]$.

*Proof.* The Hessian matrix of the loss function is

$$
(\nabla^2 L(W))_{ij} = \frac{1}{4}\mathbb{E}[\sigma(p^\top Wq/2)(1 - \sigma(p^\top Wq/2))p_{t_1(i)}q_{t_2(i)}p_{t_1(j)}q_{t_2(j)}].
$$

Considering $z \in \mathbb{R}^{d^2}$ such that $z \ne 0$, we have

$$
\begin{aligned}
z^\top \nabla^2 L(W)z &= \mathbb{E}\left[\frac{1}{4}\sigma(p^\top Wq/2)(1 - \sigma(p^\top Wq/2))\sum_{ab} z_a z_b p_{t_1(a)}q_{t_2(a)}p_{t_1(b)}q_{t_2(b)}\right] \\
&= \mathbb{E}\left[\frac{1}{4}\sigma(p^\top Wq/2)(1 - \sigma(p^\top Wq/2))\left(\sum_{a \in [d^2]} z_a p_{t_1(a)}q_{t_2(a)}\right)^2\right] \\
&\le \mathbb{E}\left[\frac{1}{4}\left(\sum_{a \in [d^2]} z_a p_{t_1(a)}q_{t_2(a)}\right)^2\right] \\
&\overset{(a)}{\le} \frac{1}{4}\|z\|_2^2 \sum_{i \in [d^2]} \mathbb{E}[(p_{t_1(i)}q_{t_2(i)})^2]
\end{aligned}
$$

where $(a)$ is due to the Cauchy–Schwarz inequality. Thus, $\nabla^2 L(W) \preceq l I_d$ and $L(W)$ is $l$-smooth, where $l$ is a constant smaller than $\frac{1}{4}\sum_{i \in [d^2]} \mathbb{E}[(p_{t_1(i)}q_{t_2(i)})^2]$. $\square$

## D.4. Proof of Theorem 3.3

*Proof.* According to Lemma D.4, the global minimizer of $L(W)$ is $W^* = 2(\Lambda^{-1} + G)$, where

$$\|G\|_{\max} \leq \frac{1}{N}\|S^{-1}(\mathbb{E}[\sigma'(a)(4qq^\top + \frac{1}{4}uu^\top\Lambda^{-1}qq^\top)$$
$$+ \sigma''(a)(\frac{1}{8}(u^\top\Lambda^{-1}q)^2\mu q^\top + 2q^\top\Lambda^{-1}q\mu q^\top)])\|_{\max} + o(1/N). \tag{38}$$

Define $R_W = \{W \in \mathbb{R}^{d\times d} \mid \|W - W^*\|_F \leq \|W^0 - W^*\|_F\}$. $R_W$ is a compact set. Then, according to Lemma D.2, for $W \in R_W$, we have $\nabla^2 L(W) \succeq \alpha I_d$. Here $\alpha > 0$ is a positive constant number. Thus, $L(W)$ is $\alpha$-strongly convex in $R_W$. Moreover, according to Lemma D.5, $L(W)$ is $l$-smooth. Then according to Lemma C.2, applying gradient descent with $\eta = 1/l$, for any $t \geq 1$, we have

$$\|W^t - W^*\|_F^2 \leq \exp(-t/\kappa) \cdot \|W^0 - W^*\|_F^2,$$

where $\kappa = l/\alpha$. $\qquad\square$

# E. In-context inference of binary classification

## E.1. Notations

In this section, we use the following notations. We denote $\mu = \mu_1 - \mu_0$, $u = 2(\mu_1 + \mu_0)$, $q = x_{\text{query}}$. Define $p = \frac{2}{M}\sum_{i=1}^M y_i x_i$. Since with probability $\mathbb{P}(y_i = 1) = 1/2$, $x_i = \mu_1 + v_i$, with probability $\mathbb{P}(y_i = 0) = 1/2$, $x_i = \mu_0 + v_i$, where $v_i \sim \mathsf{N}(0, \Lambda)$, we have $p = 2M_1\mu_1/M - 2M_0\mu_0/M + g$, where $g = \frac{2}{M}\sum_{i=1}^M v_i$, $g \sim \mathsf{N}(0, 4\Lambda/M)$, $M_1 \sim \mathsf{Bin}(M, 1/2)$. Defining $h = M_1/N - 1/2$, $u = 2(\mu_1 + \mu_0)$, we have $M_0/N = 1/2 - h$ and

$$p = \mu + hu + g. \tag{39}$$

## E.2. Proof of Theorem 3.6

*Proof.* The output of the trained transformer is

$$\widehat{y}_{\text{out}} = \sigma\left(\left(\frac{2}{M}\sum_{i=1}^M y_i x_i^\top\right)(\Lambda^{-1} + \widehat{G})x_{\text{query}}\right) = \sigma(p^\top(\Lambda^{-1} + \widehat{G})q). \tag{40}$$

The probability of $y_{\text{query}} = 1$ given $x_{\text{query}}$ is

$$\mathbb{P}(y_{\text{query}} = 1|x_{\text{query}}) = \sigma((\mu_1 - \mu_0)^\top\Lambda^{-1}x_{\text{query}}) = \sigma(\mu^\top\Lambda^{-1}q).$$

Defining $a = \mu^\top\Lambda^{-1}q$, $b = (\mu + hu + g)^\top\widehat{G}q + (hu + g)^\top\Lambda^{-1}q$, we have

$$p^\top(\Lambda^{-1} + \widehat{G})q$$
$$= (\mu + hu + g)^\top(\Lambda^{-1} + \widehat{G})q$$
$$= (\mu + hu + g)^\top\widehat{G}q + (hu + g)^\top\Lambda^{-1}q + \mu^\top\Lambda^{-1}q = a + b,$$

and

$$\mathbb{E}\left[\sigma(p^\top(\Lambda^{-1} + \widehat{G})q)\right] = \mathbb{E}\left[\sigma(a + b)\right] = \mathbb{E}[\sigma(a) + \sigma'(a)b + \sigma''(\xi(a,b))b^2/2],$$

where $\xi$ are real numbers between $a$ and $a + b$. Thus, we have

$$\mathbb{E}[|\sigma(a + b) - \sigma(a)|]$$
$$\leq \mathbb{E}[|\sigma'(a)b + \sigma''(\xi(a,b))b^2/2|]$$
$$\leq \sigma'(a)\mathbb{E}[|b|] + \mathbb{E}[b^2]$$

We first consider the term $\sigma'(a)\mathbb{E}[|b|]$. Defining $\bar{g} = \Lambda^{-1/2}M^{1/2}g/2$, we have

$$
\begin{aligned}
&\sigma'(a)\mathbb{E}[|b|] \\
&\leq \sigma'(a)\left[|\mu^\top \widehat{G}q| + \mathbb{E}[|hu^\top \widehat{G}q|] + \mathbb{E}[|g^\top \widehat{G}q|] + \mathbb{E}[|hu^\top \Lambda^{-1}q|] + \mathbb{E}[|g^\top \Lambda^{-1}q|]\right] \\
&\overset{(a)}{\leq} \sigma'(a)\left[|\mu^\top \widehat{G}q| + \frac{1}{2M^{1/2}}|u^\top \widehat{G}q| + \frac{2}{M^{1/2}}\mathbb{E}[|\bar{g}^\top \Lambda^{1/2}\widehat{G}q|] + \frac{1}{2M^{1/2}}|u^\top \Lambda^{-1}q| + \frac{2}{M^{1/2}}\mathbb{E}[|\bar{g}^\top \Lambda^{-1/2}q|]\right] \\
&\overset{(b)}{\leq} \sigma'(a)\left[\|\widehat{G}\|_{\max}\sum_{i,j\in[d]}|\mu_i q_j| + \frac{1}{M^{1/2}}\left(\frac{1}{2}|u^\top \Lambda^{-1}q| + \frac{2\sqrt{2}}{\sqrt{\pi}}\sum_{i,j\in[d]}|\Lambda_{ij}^{-1/2}q_j|\right)\right] + o\left(\frac{1}{N} + \frac{1}{\sqrt{M}}\right),
\end{aligned}
$$

where $(a)$ is due to $\mathbb{E}[|h|] \leq 1/(2M^{1/2})$ in Lemma D.1. $(b)$ is because that $\bar{g}_i \sim \mathsf{N}(0,1)$ and $\mathbb{E}[|\bar{g}_i|] = \sqrt{2}/\sqrt{\pi}$, for $i \in [d]$. For $\mathbb{E}[b^2]$, we have

$$
\mathbb{E}[b^2] \leq \mathbb{E}\left[[(\mu + hu + g)^\top \widehat{G}q]^2\right] + \mathbb{E}\left[[(hu + g)^\top \Lambda^{-1}q]^2\right] + 2\mathbb{E}\left[(\mu + hu + g)^\top \widehat{G}q(hu + g)^\top \Lambda^{-1}q\right].
$$

Notice that terms in $\mathbb{E}\left[[(\mu + hu + g)^\top \widehat{G}q]^2\right]$ contain two $\widehat{G}$. Thus, they are at most smaller than $O(\|\widehat{G}\|_{\max}^2) = O(1/N^2)$. Terms in $\mathbb{E}\left[[(hu + g)^\top \Lambda^{-1}q]^2\right]/2$ contain two $h$, or two $g$, or one $h$ and one $g$. According to Lemma D.1, we have $\mathbb{E}[|h|] = O(1/\sqrt{M})$, $\mathbb{E}[h^2] = 1/(4M)$. Moreover, $g = 2M^{-1/2}\Lambda^{1/2}\bar{g}$. Converting one $g$ to $\bar{g}$, we have a coefficient of $M^{-1/2}$. Thus, terms in $E\left[[(hu + g)^\top \Lambda^{-1}q]^2\right]/2$ contain two $h$, or two $g$, or one $h$ and one $g$ are $O(1/M)$. Terms in $E\left[(\mu + hu + g)^\top \widehat{G}q(hu + g)^\top \Lambda^{-1}q\right]$ contain at least one $\widehat{G}$ and one $h$ or one $\widehat{G}$ and one $g$. Thus, they are at most smaller than $O(\|\widehat{G}\|_{\max}/\sqrt{M}) = O(1/(N\sqrt{M}))$. Therefore, we have $E[b^2]]/2 = O(1/N^2 + 1/M + 1/(N\sqrt{M})) = o(1/N + 1/\sqrt{M})$. Finally, we have

$$
\begin{aligned}
&\mathbb{E}[\Delta(y_{\mathsf{query}}, \widehat{y}_{\mathsf{query}})] = \mathbb{E}[|\widehat{y}_{\mathsf{out}} - \mathbb{P}(y_{\mathsf{query}} = 1|x_{\mathsf{query}})|] = \mathbb{E}[|\sigma(a + b) - \sigma(a)|] \leq \sigma'(a)\mathbb{E}[|b|] + \mathbb{E}[b^2] \\
&\leq \sigma'(a)\left[\|\widehat{G}\|_{\max}\sum_{i,j\in[d]}|\mu_i q_j| + \frac{1}{M^{1/2}}\left(\frac{1}{2}|u^\top \Lambda^{-1}q| + \frac{2\sqrt{2}}{\sqrt{\pi}}\sum_{i,j\in[d]}|\Lambda_{ij}^{-1/2}q_j|\right)\right] + o\left(\frac{1}{N} + \frac{1}{\sqrt{M}}\right).
\end{aligned}
$$

$\square$

# F. Training procedure for in-context multi-class classification

In this section, we present the proof of Theorem 4.3.

### F.1. Proof sketch

First, we prove in Lemma F.3 that the expected loss function $L(W)$ (16) is strictly convex w.r.t. $W$ and is strongly convex in a compact set of $\mathbb{R}^{d\times d}$. Moreover, we prove $L(W)$ has one unique global minimizer $W^*$. Then, in Lemma F.4, by analyzing the Taylor expansion of $L(W)$, we prove that as $N \to \infty$, our loss function $L(W)$ point wisely converges to $\widetilde{L}(W)$ (defined in (44)), and the global minimizer $W^*$ converge to $2\Lambda^{-1}$. Thus, we denote $W^* = 2(\Lambda^{-1} + G)$, and prove $\|G\|_{\max} = O(N^{-1/4})$. Next, in Lemma F.5, by further analyzing the Taylor expansion of the equation $\nabla L(W^*) = 0$ at the point $2\Lambda^{-1}$, we establish a tighter bound $\|G\|_{\max} = O(cN^{-1})$. In Lemma F.6, we prove that our loss function is $l$-smooth and provide an upper bound for $l$. Thus, in a compact set $R_W$, our loss function is $\alpha$-strongly convex and $l$-smooth. Finally, leveraging the standard results from the convex optimization, we prove Theorem 4.3 in subsection F.3.

In this section, we use the following notations.

## F.2. Notations

Recall the expected loss function (16) is

$$L(W) = -\mathbb{E}\left[\sum_{k=1}^{c}(y_{\tau,\text{query}})_k \log((\widehat{y}_{\tau,\text{out}})_k)\right],\tag{41}$$

where

$$(\widehat{y}_{\tau,\text{out}})_k = \text{softmax}\left(\frac{1}{c}\left(\frac{c}{N}\sum_{i=1}^{N}y_{\tau,i}x_{\tau,i}^{\top}\right)Wx_{\tau,\text{query}}\right)_k$$

is the output of the transformer, and the label of the data follows the distribution

$$\mathbb{P}\left(y_{\tau,\text{query}} = \mathbf{e}_k|x_{\tau,\text{query}}\right) = \text{softmax}(\mu_{\tau}^{\top}\Lambda^{-1}x_{\tau,\text{query}}))_k.$$

In this section, we introduce the following notations to analyze (16). We denote $\mu_k = \mu_{\tau,k}$, $\mu = (\mu_1, \mu_2, \ldots, \mu_k) \in \mathbb{R}^{d \times c}$ and $q = x_{\tau,\text{query}}$. Then with probability $\mathbb{P}\left(y_{\tau,\text{query}} = \mathbf{e}_k\right) = 1/c$, $q = \mu_k + v$, where $v \sim \mathsf{N}(0, \Lambda)$. We define $p_k = \frac{c}{N}\sum_{i=1}^{N}(y_{\tau,i})_k x_{\tau,i} \in \mathbb{R}^d$ and $P = (p_1, p_2, \ldots, p_c) \in \mathbb{R}^{d \times c}$. We have $P^{\top} = \frac{c}{N}\sum_{i=1}^{N}y_i x_{\tau,i}^{\top} \in \mathbb{R}^{c \times d}$. Since with probability $\mathbb{P}\left(y_{\tau,i} = \mathbf{e}_k\right) = 1/c$ we have $x_{\tau,i} = \mu_k + v_i$, where $v_i \sim \mathsf{N}(0, \Lambda)$, we known $p_k = \frac{c}{N}\sum_{i=1}^{N}(y_{\tau,i})_k x_{\tau,i} = cN_k\mu_k/N + g_k$, where $g_k = \frac{c}{N}\sum_{i \in \{i|y_{\tau,i} = \mathbf{e}_k\}} v_i$, $g_k \sim \mathsf{N}(0, c^2 N_k \Lambda/N^2)$ and $(N_1, N_2, \ldots, N_c) \sim \mathsf{Multin}(n, 1/c)$. Defining $h_k = N_k/N - 1/c$, we have $N_k/N = 1/c + h_k$ and $p_k = \mu_k + ch_k\mu_k + g_k$. Defining $\bar{g}_k = \Lambda^{-1/2}g_k$, we have $\bar{g}_k \sim \mathsf{N}(0, c^2 N_k I_d/N^2)$. Defining $\mu_h = (h_1\mu_1, h_2\mu_2, \ldots, h_k\mu_k) \in \mathbb{R}^{d \times c}$ and $g = (g_1, g_2, \ldots, g_k) \in \mathbb{R}^{d \times c}$, we have $P = \mu + c\mu_h + g$.

Then, the expected loss function (16) can be expressed as

$$L(W) = \mathbb{E}\left[\sum_{k=1}^{c} -\text{softmax}(\mu^{\top}\Lambda^{-1}q)_k \log(\text{softmax}(P^{\top}Wq/c)_k)\right].\tag{42}$$

The gradient of the loss function (16) can be expressed as

$$\nabla L(W) = \mathbb{E}\left[\sum_{k=1}^{c}\left[(\text{softmax}(P^{\top}Wq/c)_k - \text{softmax}(\mu^{\top}\Lambda^{-1}q)_k)p_k q^{\top}/c\right]\right].\tag{43}$$

Moreover, we define a function $\widetilde{L}(W)$ as

$$\widetilde{L}(W) = \mathbb{E}[\sum_{k=1}^{c} -\text{softmax}(\mu^{\top}\Lambda^{-1}q)_k \log(\text{softmax}(\mu^{\top}Wq/c)_k)].\tag{44}$$

In Lemma F.4, we show that as $N \to \infty$, $L(W)$ will point wisely converge to $\widetilde{L}(W)$.

**Lemma F.1.** *Suppose* $(N_1, N_2, \ldots, N_c) \sim \mathsf{Multin}(N, 1/c)$. *Defining* $h_k = N_k/N - 1/c$, *we have*

$$\mathbb{E}[h_k] = 0$$

$$\mathbb{E}[h_k^2] = \frac{1}{N}\left(\frac{1}{c} - \frac{1}{c^2}\right)$$

$$\mathbb{E}[h_i h_j] = -\frac{1}{Nc^2}, i \neq j$$

$$\mathbb{E}[\prod_{k=1}^{c} h_k^{n_k}] = O\left(N^{-2}\right), \sum_k n_k \geq 3$$

$$\mathbb{E}[|h_j|] \leq N^{-1/2}c^{-1/2}(1 - 1/c)^{1/2}$$

$$\mathbb{E}[|h_i h_j|] = O(N^{-1})$$

$$\mathbb{E}[|h_i h_j h_k|] = O\left(N^{-3/2}\right)$$

$$\mathbb{E}[|h_i h_j h_k h_l|] = O\left(N^{-2}\right),$$

*where* $i, j, k, l \in [c]$.

*Proof.* Since $(N_1, N_2, \ldots, N_c) \sim \mathsf{Multin}(N, 1/c)$, the moment-generating function of $(N_1, N_2, \ldots, N_c)$ is

$$M_N(t) = \left( \frac{1}{c} \sum_{i=1}^{c} \exp(t_i) \right)^N$$

We can compute the moment-generating function of $h = (h_1, h_2, \ldots, h_c)$ as follows:

$$
\begin{aligned}
M_h(t) &= \exp\left( -\sum_{i=1}^{c} t_i/c \right) M_N(t/N) = \left( \frac{1}{c} \sum_{i=1}^{c} \exp\left( \frac{1}{N} \left( t_i - \sum_{j=1}^{c} t_j/c \right) \right) \right)^N \\
&= \left[ 1 + \frac{1}{Nc} \left( \sum_{i=1}^{c} t_i - c \sum_{j=1}^{c} t_j/c \right) + \frac{1}{2N^2 c} \left( \sum_{i=1}^{c} \left( t_i - \sum_{j=1}^{c} t_j/c \right) \right)^2 \right. \\
&\quad \left. + \sum_{k=3}^{\infty} \frac{1}{k! N^k c} \left( \sum_{i=1}^{c} \left( t_i - \sum_{j=1}^{c} t_j/c \right) \right)^k \right]^N \\
&= \left[ 1 + \sum_{i=1}^{c} \frac{1}{2N} (1/c - 1/c^2) t_i^2 - \sum_{i \ne j \in [c]} \frac{1}{2Nc^2} t_i t_j + \sum_{k=3}^{\infty} \frac{1}{k! N^k c} \left( \sum_{i=1}^{c} \left( t_i - \sum_{j=1}^{c} t_j/c \right) \right)^k \right]
\end{aligned}
$$

Observing the coefficients of $h$, we have

$$\mathbb{E}[h_k] = 0$$

$$\mathbb{E}[h_k^2] = \frac{1}{N} \left( \frac{1}{c} - \frac{1}{c^2} \right)$$

$$\mathbb{E}[h_i h_j] = -\frac{1}{Nc^2}, i \ne j$$

$$\mathbb{E}[\prod_{k=1}^{c} h_k^{n_k}] = O\left( N^{-2} \right), \sum_k n_k \ge 3,$$

where $i, j, k \in [c]$.

Iteratively applying the Hölder's inequality, we have

$$
\begin{aligned}
\mathbb{E}[|h_j|] &\le \left( \mathbb{E}[h_j^2] \right)^{1/2} = N^{-1/2} c^{-1/2} (1 - 1/c)^{1/2} \\
\mathbb{E}[|h_i h_j|] &\le \left( \mathbb{E}[h_i^2 h_j^2] \right)^{1/2} = O(N^{-1}) \\
\mathbb{E}[|h_i|^3] &\le \mathbb{E}[|h_i|^4]^{3/4} = (N^{-3/2}) \\
\mathbb{E}[|h_i h_j h_k|] &\le \mathbb{E}[|h_i|^3]^{1/3} \mathbb{E}[|h_j|^3]^{1/3} \mathbb{E}[|h_k|^3]^{1/3} = O\left( N^{-3/2} \right) \\
\mathbb{E}[|h_i h_j h_k h_l|] &\le \mathbb{E}[|h_i|^4]^{1/4} \mathbb{E}[|h_j|^4]^{1/4} \mathbb{E}[|h_k|^4]^{1/4} \mathbb{E}[|h_l|^4]^{1/4} = O\left( N^{-2} \right)
\end{aligned}
$$

where $i, j, k, l \in [c]$. $\qquad \square$

**Lemma F.2.** *Suppose* $g_k \sim \mathsf{N}(0, c^2 N_k \Lambda / N^2)$ *and* $(N_1, N_2, \ldots, N_c) \sim \mathsf{Multin}(N, 1/c)$, *define* $\bar{g}_k = \Lambda^{-1/2} g_k$ *and* $N_k/N = 1/c + h_k$, *we have*

$$
\begin{aligned}
\mathbb{E}[(\bar{g}_k)_i] &= 0 \\
\mathbb{E}[(\bar{g}_k)_i (\bar{g}_l)_j] &= \delta_{kl} \delta_{ij} c/N \\
\mathbb{E}[(\bar{g}_{k_1})_{i_1} (\bar{g}_{k_2})_{i_2} (\bar{g}_{k_3})_{i_3}] &= 0 \\
\mathbb{E}[(\bar{g}_k)_i^4] &= \mathbb{E}[3c^2/N^2 (1 + ch_k)^2] = O(N^{-2}) \\
\mathbb{E}[h_m (\bar{g}_k)_i (\bar{g}_l)_j] &= \mathbb{E}[c^2 \delta_{kl} \delta_{ij} h_m h_k / N] = O(N^{-2}) \\
\mathbb{E}[h_m h_l (\bar{g}_k)_i] &= 0
\end{aligned}
$$

*where $i, j, i_1, i_2, i_3 \in [d], k, l, m, k_1, k_2, k_3 \in [c]$.*

*For any $n_{1k}$, $n_{2ki}$ satisfying $\sum_{k \in [c]} n_{1k} + \sum_{k \in [c], i \in [d]} n_{2ki} = 1, 2, 3$, we have*

$$\mathbb{E}[\prod_{k \in [c], i \in [d]} h_k^{n_{1k}} (\bar{g}_k)_i^{n_{2ki}}] = O(N^{-1})$$

*Moreover, we have*

$$\mathbb{E}[|(\bar{g}_k)_i|] \leq \mathbb{E}[(\bar{g}_k)_i^2]^{1/2} = N^{-1/2} c^{1/2}$$
$$\mathbb{E}[|(\bar{g}_k)_i|^3] \leq \mathbb{E}[(\bar{g}_k)_i^4]^{3/4} = O(N^{-3/2})$$

*where $i \in [d], k \in [c]$.*

*For any $n_{1k}$, $n_{2ki}$ satisfying $\sum_{k \in [c]} n_{1k} + \sum_{k \in [c], i \in [d]} n_{2ki} = n$, $n = 1, 2, 3, 4$, we have*

$$\mathbb{E}[\prod_{k \in [c], i \in [d]} |h_k^{n_{1k}} (\bar{g}_k)_i^{n_{2ki}}|] = O(N^{-n/2})$$

*Proof.* Since $g_k \sim \mathsf{N}(0, c^2 N_k \Lambda / N^2)$ and $\bar{g}_k \sim \mathsf{N}(0, c^2 N_k I_d / N^2) = \mathsf{N}(0, (c/N + c^2 h_k/N) I_d)$, we have

$$\mathbb{E}[(\bar{g}_k)_i] = 0$$
$$\mathbb{E}[(\bar{g}_k)_i (\bar{g}_l)_j] = \delta_{kl} \delta_{ij} c/N$$
$$\mathbb{E}[(\bar{g}_{k_1})_{i_i} (\bar{g}_{k_2})_{i_2} (\bar{g}_{k_3})_{i_3}] = 0$$
$$\mathbb{E}[(\bar{g}_k)_i^4] = \mathbb{E}[3c^2/N^2 (1 + ch_k)^2] = O(N^{-2})$$
$$\mathbb{E}[h_m (\bar{g}_k)_i (\bar{g}_l)_j] = \mathbb{E}[c^2 \delta_{kl} \delta_{ij} h_m h_k / N] = O(N^{-2})$$
$$\mathbb{E}[h_m h_l (\bar{g}_k)_i] = 0$$

where $i, j, i_1, i_2, i_3 \in [d], k, l, m, k_1, k_2, k_3 \in [c]$. Thus, with the results from Lemma F.1, for any $n_{1k}$, $n_{2ki}$ satisfying $\sum_{k \in [c]} n_{1k} + \sum_{k \in [c], i \in [d]} n_{2ki} = 1, 2, 3$, we have

$$\mathbb{E}[\prod_{k \in [c], i \in [d]} h_k^{n_{1k}} (\bar{g}_k)_i^{n_{2ki}}] = O(N^{-1})$$

Moreover, according to the Jensen's inequality, we have

$$\mathbb{E}[|(\bar{g}_k)_i|] \leq \mathbb{E}[(\bar{g}_k)_i^2]^{1/2} = N^{-1/2} c^{1/2}$$
$$\mathbb{E}[|(\bar{g}_k)_i|^3] \leq \mathbb{E}[(\bar{g}_k)_i^4]^{3/4} = O(N^{-3/2})$$

where $i \in [d], k \in [c]$. Thus, with the results from Lemma F.1, for any $n_{1k}$, $n_{2ki}$ satisfying $\sum_{k \in [c]} n_{1k} + \sum_{k \in [c], i \in [d]} n_{2ki} = n$, $n = 1, 2, 3, 4$, we have

$$\mathbb{E}[\prod_{k \in [c], i \in [d]} |h_k^{n_{1k}} (\bar{g}_k)_i^{n_{2ki}}|] \leq \prod_{k \in [c], i \in [d]} \mathbb{E}[|h_k^n|]^{n_{1k}/n} \mathbb{E}[|(\bar{g}_k)_i^n|]^{n_{2ki}/n} = O(N^{-n/2}).$$

$\square$

**Lemma F.3.** *For the loss function $L(W)$ (16), we have $\nabla^2 L(W) \succ 0$. For any compact set $R_W$, when $W \in R_W$, we have $\nabla^2 L(W) \succ \gamma I_d$ for some $\gamma > 0$. Additionally, $L(W)$ has one unique global minimizer on $\mathbb{R}^{d \times d}$.*

*For $\widetilde{L}(W)$ defined in (44), we also have $\nabla^2 \widetilde{L}(W) \succ 0$. For any compact set $R_W$, when $W \in R_W$, we have $\nabla^2 \widehat{L}(W) \succ \gamma I_d$ for some $\gamma > 0$. Additionally, $\widetilde{L}(W)$ has one unique global minimizer on $\mathbb{R}^{d \times d}$.*

*Proof.* We vectorize $W$ as $\text{Vec}(W) \in \mathbb{R}^{d^2}$, where $\text{Vec}(W)_i = W_{t_1(i),t_2(i)}$, $t_1(x) = \lfloor (x-1)/d \rfloor + 1, t_2(x) = ((x-1) \bmod d) + 1$. Then, we have

$$(\nabla L(W))_i = \mathbb{E}\left[\sum_{k=1}^{c}\left[(\text{softmax}(P^\top W q/c)_k - \text{softmax}(\mu^\top \Lambda^{-1} q)_k)(p_k)_{t_1(i)} q_{t_2(i)}/c\right]\right] \tag{45}$$

Note that

$$\text{softmax}(P^\top W q/c)_k = \sigma(a_k)$$
$$\nabla\text{softmax}(P^\top W q/c)_k = \sigma(a_k)(1 - \sigma(a_k))\nabla a_k,$$

where $a_k = -\log(\sum_{l=1,\dots,c,l\neq k} \exp((p_l - p_k)W q/c))$. For $\nabla a_k$, we have

$$\nabla a_k = \frac{\sum_{l=1,\dots,c,l\neq k} \exp\left((p_l - p_k)^\top W q/c\right)(p_k - p_l)q^\top/c}{\sum_{l=1,\dots,c,l\neq k} \exp\left((p_l - p_k)^\top W q/c\right)}$$
$$= \frac{\sum_{l=1,\dots,c,l\neq k} \exp\left(p_l^\top W q/c\right)(p_k - p_l)q^\top/c}{\sum_{l=1,\dots,c,l\neq k} \exp\left(p_l^\top W q/c\right)}.$$

Then we have

$$\nabla\text{softmax}(P^\top W q/c)_k = \text{softmax}(P^\top W q/c)_k \frac{\sum_{l=1,\dots,c,l\neq k} \exp\left(p_l^\top W q\right)(p_k - p_l)q^\top/c}{\sum_{n=1,\dots,c} \exp\left(p_n^\top W q/c\right)}$$
$$= \sum_{l=1,\dots,c,l\neq k} \text{softmax}(P^\top W q/c)_k \text{softmax}(P^\top W q/c)_l (p_k - p_l)q^\top/c$$

and

$$(\nabla\text{softmax}(P^\top W q/c)_k)_j = \sum_{l=1,\dots,c,l\neq k} \text{softmax}(P^\top W q/c)_k \text{softmax}(P^\top W q/c)_l (p_k - p_l)_{t_1(j)} q_{t_2(j)}/c.$$

We can express the Hessian matrix of the loss function with the following form:

$$(\nabla^2 L(W))_{ij} = \mathbb{E}\left[\sum_{k=1}^{c}\sum_{l=1,\dots,c,l\neq k} \text{softmax}(P^\top W q/c)_k \text{softmax}(P^\top W q/c)_l (p_k)_{t_1(i)} q_{t_2(i)}(p_k - p_l)_{t_1(j)} q_{t_2(j)}/c^2\right]$$
$$= \mathbb{E}\left[\sum_{k=2}^{c}\sum_{l=1}^{k-1} \text{softmax}(P^\top W q/c)_k \text{softmax}(P^\top W q/c)_l (p_k - p_l)_{t_1(i)} q_{t_2(i)}(p_k - p_l)_{t_1(j)} q_{t_2(j)}/c^2\right].$$

Considering $z \in \mathbb{R}^{d^2}$ such that $z \neq 0$, we have

$$z^\top \nabla^2 L(W) z = \mathbb{E}\left[\frac{1}{c^2}\sum_{k=2}^{c}\sum_{l=1}^{k-1} \text{softmax}(P^\top W q/c)_k \text{softmax}(P^\top W q/c)_l \sum_{ab} z_a z_b (p_k - p_l)_{t_1(a)} q_{t_2(a)}(p_k - p_l)_{t_1(b)} q_{t_2(b)}\right]$$
$$= \mathbb{E}\left[\frac{1}{c^2}\sum_{k=2}^{c}\sum_{l=1}^{k-1} \text{softmax}(P^\top W q/c)_k \text{softmax}(P^\top W q/c)_l \left(\sum_{a\in[d^2]} z_a (p_k - p_l)_{t_1(a)} q_{t_2(a)}\right)^2\right]$$

Since for any $P, q, k, l$, $\text{softmax}(P^\top W q/c)_k \text{softmax}(P^\top W q/c)_l > 0$, we have $z^\top \nabla^2 L(W) z \geq 0$. Thus, $\nabla^2 L(W) \succeq 0$ and $L(W)$ is convex.

Defining $\tilde{p} = p_1 - p_2$, we have

$$z^\top \nabla^2 L(W) z$$

$$\geq \mathbb{E}\left[ \frac{1}{c^2} \text{softmax}(P^\top Wq/c)_1 \text{softmax}(P^\top Wq/c)_2 \left( \sum_{a \in [d^2]} z_a (p_1 - p_2)_{t_1(a)} q_{t_2(a)} \right)^2 \right]$$

$$= \int \frac{1}{c^2} \text{softmax}(P^\top Wq/c)_1 \text{softmax}(P^\top Wq/c)_2 \left( \sum_{a \in [d^2]} z_a \tilde{p}_{t_1(a)} q_{t_2(a)} \right)^2 f_{Pq}(P, q) dP dq$$

where $f_{Pq}(P, q)$ are the PDF function of $P, q$. For any $z \neq 0$, we denote $z_{ij} = z_{((i-1)d+j)}$, suppose $a, b \in \arg\max_{i,j} |z_{ij}|$, we consider a set of constants $\{c_{1pi}, c_{2pi}\}, \{c_{1qi}, c_{2qi}\}, i, j \in [d]$, where $c_{1pa} = d, c_{2pa} = d + 1, c_{1qb} = d, c_{2qb} = d + 1$, and $c_{1pi} = 1/16, c_{2pi} = 1/8, i \neq a, c_{1qj} = 1/16, c_{2qj} = 1/8, j \neq b$. Then, for any $c_{pi} \in [c_{1pi}, c_{2pi}], c_{qj} \in [c_{1qj}, c_{2qj}]$, we have

$$\left| \sum_{i,j \in [d]} z_{ij} c_{pi} c_{qj} \right| \geq \left[ d^2 - 2(d+1)(d-1)/8 - (d-1)^2/64 \right] \max_{ij} |z_{ij}| \geq d^2 \max_{ij} |z_{ij}|/2.$$

Then, we define region $\Omega(a, b) = \{\tilde{p} = \sum_i c_{pi} \mathbf{e}_i, q = \sum_j c_{qj} \mathbf{e}_j, c_{pi} \in [c_{1pi}, c_{2pi}], c_{qj} \in [c_{1qj}, c_{2qj}], \|P\|_F^2 \leq c^2(\sum_i c_{2pi}^2 + c_{2qj}^2)\}$. We have

$$\min_{\Omega(a,b)} \left( \sum_{l \in [d^2]} z_l \tilde{p}_{t_1(l)} q_{t_2(l)} \right)^2 \geq d^4 \max_{ij} |z_{ij}|^2/4 \geq \|z\|_2^2/4.$$

Defining

$$C(\Omega) = \min_{a \in [d], b \in [d]} \int_{\Omega(a,b)} f_{Pq}(P, q) dP dq,$$

$$S(\Omega, W) = \min_{a \in [d], b \in [d]} \min_{\Omega(a,b)} \left\{ \frac{1}{c^2} \text{softmax}(P^\top Wq/c)_1 \text{softmax}(P^\top Wq/c)_2 \right\},$$

we have $S(\Omega, W) > 0$. Since we have $f_{Pq}(P, q) > 0$ for all $P, q$ and $\Omega(a, b)$ are non-zero measures for $P, q$. Thus, we have $C(\Omega) > 0$. Then, for any $z \neq 0$, we have

$$z^\top \nabla^2 L(W) z$$

$$\geq \int_{\Omega(a,b)} \frac{1}{c^2} \text{softmax}(P^\top Wq/c)_1 \text{softmax}(P^\top Wq/c)_2 \left( \sum_{l \in [d^2]} z_l \tilde{p}_{t_1(l)} q_{t_2(l)} \right)^2 f_{Pq}(P, q) dP dq$$

$$\geq C(\Omega) S(\Omega, W) \|z\|_2^2/4 > 0$$

Thus, we have $\nabla^2 L(W) \succ 0$. $L(W)$ is strictly convex.

Moreover, for any compact set $R_W$ of $\mathbb{R}^{d \times d}$, for any $W \in R_W$, we have

$$S(\Omega) = \min_{W \in R_W} \min_{a \in [d], b \in [d]} \min_{\Omega(a,b)} \left\{ \frac{1}{c^2} \text{softmax}(P^\top Wq/c)_1 \text{softmax}(P^\top Wq/c)_2 \right\} > 0.$$

Then, for any $W \in R_W$, for any $z \neq 0$, we have

$$z^\top \nabla^2 L(W) z$$

$$\geq \int_{\Omega(a,b)} \frac{1}{c^2} \text{softmax}(P^\top Wq/c)_1 \text{softmax}(P^\top Wq/c)_2 \left( \sum_{l \in [d^2]} z_l \tilde{p}_{t_1(l)} q_{t_2(l)} \right)^2 f_{Pq}(P, q) dP dq$$

$$\geq C(\Omega) S(\Omega) \|z\|_2^2/4.$$

Thus, when $W \in R_W$, $R_W$ is a compact set, we have $\nabla^2 L(W) \succ C(\Omega)S(\Omega)I_d/4$, our loss function is $\gamma-$strongly convex, where $\gamma = C(\Omega)S(\Omega)/4$.

Because our loss function is strictly convex in $\mathbb{R}^{d \times d}$, it has at most one global minimizer in $\mathbb{R}^{d \times d}$. Next, we prove all level sets of our loss function are compact, i.e. $V_\alpha = \{W \in \mathbb{R}^{d \times d} \mid L(W) \leq \alpha\}$ is compact for all $\alpha$. We prove it by contradiction. Suppose $V_\alpha$ is not compact for some $\alpha$. Since our loss function is continuous and convex, $V_\alpha$ is an unbounded convex set. Since the dimension of $V_\alpha$ is $d^2$, consider a point $W^\alpha \in V_\alpha$, there must exists a $W^k \neq 0_{d \times d}$ that $\{W^\alpha + tW^k \mid t = [0, \infty)\} \in V_\alpha$. For this $W^k \neq 0_{d \times d}$, there must exist a set of constants $0 < c_{3pi} < c_{4pi}, 0 < c_{3qj} < c_{4qj}$ such that for any $c_{pi} \in [c_{3pi}, c_{4pi}], c_{qj} \in [c_{3qj}, c_{4qj}]$, we have

$$|\sum_{ij} c_{pi}c_{qj}W_{ij}^k| \neq 0.$$

Thus, we have

$$\lim_{t \to \infty} |\sum_{ij} c_{pi}c_{qj}(W_{ij}^\alpha + tW_{ij}^k)| = \infty.$$

We define $\Omega_0 = \{\tilde{p} = \sum_i c_{pi}\mathbf{e}_i, q = \sum_j c_{qj}\mathbf{e}_j, c_{pi} \in [c_{3pi}, c_{4pi}], c_{qj} \in [c_{3qj}, c_{4qj}], \|P\|_F^2 \leq c^2(\sum_i c_{4pi}^2 + c_{4qj}^2), \|\mu\|_F^2 \leq c^2(\sum_i c_{4pi}^2 + c_{4qj}^2)\}$. Then, defining

$$C(\Omega_0) = \int_{\Omega_0} f_{Pq}(P, q)dPdq,$$
$$S(\Omega_0) = \min_{\Omega_0} \{\min\{\text{softmax}(\mu^\top Wq/c)_1, \text{softmax}(\mu^\top Wq/c)_2\}\}$$

we have $S(\Omega_0) > 0$. Since $\Omega_0$ are non-zero measures for $P, q$, we have $C(\Omega_0) > 0$. Then, we have

$$\lim_{t \to \infty} L(W^\alpha + tW^k)$$
$$= \lim_{t \to \infty} \mathbb{E}[\sum_{l=1}^{c} -\text{softmax}(\mu^\top \Lambda^{-1}q)_l \log(\text{softmax}(P^\top(W^\alpha + tW^k)q/c)_l)]$$
$$\geq \lim_{t \to \infty} \int_{\Omega_0} [-\text{softmax}(\mu^\top \Lambda^{-1}q)_1 \log(\text{softmax}(P^\top(W^\alpha + tW^k)q/c)_1)]f_{Pq}(P, q)dPdq$$
$$+ \lim_{t \to \infty} \int_{\Omega_0} [-\text{softmax}(\mu^\top \Lambda^{-1}q)_2 \log(\text{softmax}(P^\top(W^\alpha + tW^k)q/c)_2)]f_{Pq}(P, q)dPdq$$
$$\geq \lim_{t \to \infty} \int_{\Omega_0} [-\text{softmax}(\mu^\top \Lambda^{-1}q)_1 \log(\sigma(\tilde{p}^\top(W^\alpha + tW^k)q/c))]f_{Pq}(P, q)dPdq$$
$$+ \lim_{t \to \infty} \int_{\Omega_0} [-\text{softmax}(\mu^\top \Lambda^{-1}q)_2 \log(\sigma(-\tilde{p}^\top(W^\alpha + tW^k)q/c))]f_{Pq}(P, q)dPdq$$
$$\geq C(\Omega_0)S(\Omega_0) \cdot \min_{\Omega_0}\left\{\lim_{t \to \infty}[-\log(\sigma(\sum_{ij} c_{pi}c_{qj}(W_{ij}^\alpha + tW_{ij}^k)/c))]\right\}$$
$$+ C(\Omega_0)S(\Omega_0) \cdot \min_{\Omega_0}\left\{\lim_{t \to \infty}[-\log(\sigma(-\sum_{ij} c_{pi}c_{qj}(W_{ij}^\alpha + tW_{ij}^k)/c))]\right\}$$
$$= \infty$$

This contradicts the assumption $L(W^\alpha + tW^k) \leq \alpha$. Thus, all level sets of the loss function $L(W)$ are compact, which means there exists a a global minimizer for $L(W)$. Together with the fact that $L(W)$ is strictly convex, $L(W)$ has one unique a global minimizer on $\mathbb{R}^{d \times d}$.

Similarly, we can prove the same conclusions for $\widetilde{L}(W)$. □

**Lemma F.4.** *Denoting the global minimizer of our loss function* (16) *as* $W^*$, *we have* $W^* = c(\Lambda^{-1} + G)$, *where* $\|G\|_{\max} = O(N^{-1/4})$.

*Proof.* Let $a = \mu^\top \Lambda^{-1} q, s = \mu^\top W q/c, r = (\mu_h + g)^\top W q/c, a_k = \mu_k^\top \Lambda^{-1} q, s_k = \mu_k^\top W q/c, r_k = (ch_k \mu_k + g_k)^\top W q/c$. Performing the Taylor expansion on (16), we have

$$L(W) = \mathbb{E}\left[ \sum_{k=1}^c -\zeta_k(a) \log(\zeta_k(s + r)) \right]$$

$$= \mathbb{E}\left[ \sum_{k=1}^c -\zeta_k(a) \log(\zeta_k(s)) - \sum_{k,l=1}^c \zeta_k(a) R_{kl}(s,r) r_l \right]$$

$$= \widetilde{L}(W) - \mathbb{E}\left[ \sum_{k,l=1}^c \zeta_k(a) R_{kl}(s,r) r_l \right]$$

where $|R_{kl}(s,r)| \leq \sup_y |\frac{\partial \log(\zeta_k(y))}{\partial y_l}| \sup_y |\frac{1}{\zeta_k(y)} \frac{\partial \zeta_k(y)}{\partial y_l}| = \sup_y |\delta_{kl} - \zeta_l(y)| \leq 1$. Thus, we have

$$\left| \widetilde{L}(W) - L(W) \right|$$

$$\leq c \sum_{l=1}^c \mathbb{E}\left[ |r_l| \right]$$

$$\leq \sum_{l=1}^c c\mathbb{E}\left[ |h_l \mu_l^\top W q| \right] + \mathbb{E}\left[ |g_l^\top W q| \right]$$

$$\leq O(1) \|W\|_{\max} \mathbb{E}[|h_l|] + O(1) \|W\|_{\max} \mathbb{E}[|(\bar{g}_l)_i|]$$

$$\leq C_l \|W\|_{\max} N^{-1/2}$$

where the last inequality is due to Lemma F.1, F.2. $C_l$ is a constant independent of $N$ and $W$. This shows that $L(W)$ point wisely converge to $\widetilde{L}(W)$.

According to Lemma D.2, $\widetilde{L}(W)$ has one unique global minimizer. Considering the equation:

$$\nabla \widetilde{L}(W) = \mathbb{E}[\sum_{k=1}^c -\text{softmax}(\mu^\top \Lambda^{-1} q)_k \log(\text{softmax}(\mu^\top W q/c)_k)] = 0$$

We can easily find that $\nabla \widetilde{L}(c\Lambda^{-1}) = 0$ and $W = c\Lambda^{-1}$ is the global minimizer of $\widetilde{L}(W)$.

Considering a compact set $R_W = \{W \mid \|W - 2\Lambda^{-1}\|_F \leq \rho_W\}$, we have $\|W\|_{\max} \leq C_W$ for any $W \in R_W$. Here $\rho_W, C_W$ are some positive finite constants. Then, we have

$$\left| \widetilde{L}(W) - L(W) \right| \leq C_l' N^{-1/2}, \ W \in R_W$$

where $C_l' = C_l C_W$ is a constant independent of $N$ and $W$. This shows that, for any $W \in R_W$, $L(W)$ uniformly converge to $\widetilde{L}(W)$.

Denote $W^*$ as the global minimizer of $L(W)$ with prompt length $N$. Then, we show that, when $N$ is sufficiently large, $W^* \in R_W$. We first denote $\partial R_W = \{W \mid \|W - c\Lambda^{-1}\|_F = \rho_W\}$, $\Delta = \min_{W \in \partial R_W} \widetilde{L}(W) - \widetilde{L}(c\Lambda^{-1}) > 0$. Then, for $N \geq (4C_l'/\Delta)^2$, and for any $W \in R_W$, we have

$$\left| \widetilde{L}(W) - L(W) \right| \leq \Delta/4$$

$$\min_{W \in \partial R_W} L(W) - \min_{W \in R_W} L(W) \geq \min_{W \in \partial R_W} L(W) - L(c\Lambda^{-1}) \geq \Delta/2 > 0$$

Since $L(W)$ is strictly convex, we have $W^* = \text{argmin}_W L(W) \in R_W$.

Then, we have

$$|\widetilde{L}(W^*) - L(W^*)| \leq C_l'/N$$
$$|\widetilde{L}(c\Lambda^{-1}) - L(c\Lambda^{-1})| \leq C_l'/N$$

$$\widetilde{L}(W^*) \leq L(W^*) + C_l'/N \leq L(c\Lambda^{-1}) + C_l'/N \leq \widetilde{L}(c\Lambda^{-1}) + 2C_l'N^{-1/2}$$

According to Lemma D.2, for $W \in R_W$, we have $\nabla^2 \widetilde{L}(W) \succ \gamma I_d$, where $\gamma$ is a positive constant independent of $N$. Thus, $\widetilde{L}(W)$ is $\gamma$-strongly convex in $R_W$. According to Lemma C.1, we have

$$\|W^* - c\Lambda^{-1}\|_F^2 \leq \frac{2}{\gamma}(\widetilde{L}(W^*) - \widetilde{L}(c\Lambda^{-1})) \leq \frac{4C_l'}{\gamma N^{1/2}}$$

Thus, when $N \to \infty$, we have $W^* \to c\Lambda^{-1}$. Denoting $W^* = c(\Lambda^{-1} + G)$, we have $\|G\|_{\max} = O(N^{-1/4})$. $\qquad\square$

**Lemma F.5.** *The global minimizer of the loss function* (16) *is* $W^* = c(\Lambda^{-1} + G)$. *We have*

$$\|G\|_{\max} \leq \frac{1}{N}\left\|S^{-1}\mathbb{E}\left[\sum_{k,l=1}^{c}\frac{\partial\zeta_k(a)}{\partial a_l}(c\delta_{kl} - 1)\mu_k\mu_l^\top\Lambda^{-1}qq^\top + \sum_{k=1}^{c}\frac{\partial\zeta_k(a)}{\partial a_k}cqq^\top\right.\right.$$
$$\left.\left. + \sum_{k,l,n=1}^{c}\frac{\partial^2\zeta_k(a)}{\partial a_l\partial a_n}(c\delta_{ln} - 1)\mu_l^\top\Lambda^{-1}q\mu_n^\top\Lambda^{-1}q\mu_kq^\top/2 + \sum_{k,l=1}^{c}\frac{\partial^2\zeta_k(a)}{\partial a_l^2}cq^\top\Lambda^{-1}q\mu_kq^\top/2\right]\right\|_{\max}$$
$$+ o(1/N),$$

*where* $a = \mu^\top\Lambda^{-1}q$, $a_k = \mu_k^\top\Lambda^{-1}q$, $S = c^2\nabla^2\widetilde{L}(c\Lambda^{-1})$. *Ignoring constants other than* $c, N$, *we have* $\|G\|_{\max} \leq O(c/N)$.

*Proof.* According to Lemma F.3, the loss function $L(W)$ has a unique global minimizer $W^*$. We have

$$\nabla L(W^*) = \mathbb{E}\left[\sum_{k=1}^{c}\left[(\zeta_k(P^\top W^*q/c) - \zeta_k(\mu^\top\Lambda^{-1}q))p_kq^\top/c\right]\right] = 0. \tag{46}$$

Let $W^* = c(\Lambda^{-1} + G)$, $a = \mu^\top\Lambda^{-1}q$, $a_k = \mu_k^\top\Lambda^{-1}q$, $b = (\mu + c\mu_h + g)^\top Gq + (c\mu_h + g)^\top\Lambda^{-1}q$, $b_k = (\mu_k + ch_k\mu_k + g_k)^\top Gq + (ch_k\mu_k + g_k)^\top\Lambda^{-1}q$. The Taylor expansion of $\zeta_k(a + b)$ at point $a$ is

$$\zeta_k(a + b) = \zeta_k(a) + \sum_{l=1}^{c}\frac{\partial\zeta_k(a)}{\partial a_l}b_l + \sum_{l,n=1}^{c}\frac{\partial^2\zeta_k(a)}{\partial a_l\partial a_n}b_lb_n/2! + \sum_{l,n,m=1}^{c}R_{klnm}(a, b)b_lb_nb_m/3!,$$

where $|R_{klnm}(a, b)| \leq \sup_x|\frac{\partial^3\zeta_k(x)}{\partial x_l\partial x_n\partial x_m}|$. Thus, our equation (46) become

$$\mathbb{E}\left[\sum_{k,l=1}^{c}\frac{\partial\zeta_k(a)}{\partial a_l}b_lp_kq^\top + \sum_{k,l,n=1}^{c}\frac{\partial^2\zeta(a)}{\partial a_l\partial a_n}b_lb_np_kq^\top/2! + \sum_{k,l,n,m=1}^{c}R_{klnm}(a, b)b_lb_nb_mp_kq^\top/3!\right] = 0. \tag{47}$$

For the first term $\sum_{k,l=1}^c \frac{\partial \zeta_k(a)}{\partial a_l} b_l p_k q^\top$, according to Lemma F.1, we have

$$
\mathbb{E}\left[\sum_{k,l=1}^c \frac{\partial \zeta_k(a)}{\partial a_l} b_l p_k q^\top\right]
$$

$$
=\mathbb{E}\left[\sum_{k,l=1}^c \frac{\partial \zeta_k(a)}{\partial a_l}\left[\mu_l^\top G q \mu_k q^\top + c^2 h_l h_k \mu_l^\top G q \mu_k q^\top + c^2 h_l h_k \mu_l^\top \Lambda^{-1} q \mu_k q^\top + g_l^\top \Lambda^{-1} q g_k q^\top + g_l^\top G q g_k q^\top\right]\right]
$$

$$
=\mathbb{E}\left[\sum_{k,l=1}^c \frac{\partial \zeta_k(a)}{\partial a_l}\left(\mu_k \mu_l^\top G q q^\top + (c\delta_{kl}-1)\mu_k \mu_l^\top G q q^\top/N + (c\delta_{kl}-1)\mu_k \mu_l^\top \Lambda^{-1} q q^\top/N\right)\right.
$$

$$
\left. + \sum_{k=1}^c \frac{\partial \zeta_k(a)}{\partial a_k}\left(c q q^\top/N + c\Lambda G q q^\top/N\right)\right]. \tag{48}
$$

According to Lemma F.4, $O(\|G\|_{\max}) = O(N^{-1/4}) = o(1)$, we have

$$
\left\|\mathbb{E}\left[\sum_{k,l=1}^c \frac{\partial \zeta_k(a)}{\partial a_l}(c\delta_{kl}-1)\mu_k \mu_l^\top G q q^\top/N + \sum_{k=1}^c \frac{\partial \zeta_k(a)}{\partial a_k} c\Lambda G q q^\top/N\right]\right\|_{\max}
$$

$$
\leq O(\|G\|_{\max}/N) = o(1/N) \tag{49}
$$

For the second term $\sum_{k,l,n=1}^c \frac{\partial^2 \zeta_k(a)}{\partial a_l \partial a_n} b_l b_n p_k q^\top/2!$, we have

$$
\mathbb{E}\left[\sum_{k,l,n=1}^c \frac{\partial^2 \zeta_k(a)}{\partial a_l \partial a_n} b_l b_n p_k q^\top/2!\right]
$$

$$
=\frac{1}{2}\mathbb{E}\left[\sum_{k,l,n=1}^c \frac{\partial^2 \zeta_k(a)}{\partial a_l \partial a_n}\left(\underbrace{\sum_{\phi_1 \in \{\mu_l, ch_l\mu_l, g_l\}, \phi_2 \in \{\mu_n, ch_n\mu_n, g_n\}} \phi_1^\top G q \phi_2^\top G q p_k q^\top}_{(i)}\right.\right.
$$

$$
+ \underbrace{\sum_{\phi_1 \in \{\mu_l, ch_l\mu_l, g_l\}, \phi_2 \in \{ch_n\mu_n, g_n\}} 2\phi_1^\top G q \phi_2^\top \Lambda^{-1} q p_k q^\top}_{(ii)}
$$

$$
\left.\left. + \underbrace{\sum_{\phi_1 \in \{ch_l\mu_l, g_l\}, \phi_2 \in \{ch_n\mu_n, g_n\}} \phi_1^\top \Lambda^{-1} q \phi_2^\top \Lambda^{-1} q p_k q^\top}_{(iii)}\right)\right].
$$

For terms $(i)$ having two $G$, their max norms are at most smaller than $O(\|G\|_{\max}^2)$. For terms $(ii)$ having one $G$, define $\bar{g}_l = \Lambda^{-1/2} g_l$, these terms must contain $n_{1j}$ number of $h_j$ and $n_{2ji}$ number of $(\bar{g}_j)_i$, we have $\sum_{j\in[c],i\in[d]} n_{1j} + n_{2ji} = n_t, n_t = 1,2,3$. According to Lemma F.2, we know that for $n_t = 1,2,3$,

$$
\mathbb{E}\left[\prod_{j\in[c],i\in[d]} h_j^{n_{1j}} (\bar{g}_j)_i^{n_{2ji}}\right] = O(N^{-1})
$$

Thus, the max norm of expectations of terms in (ii) are at most smaller than $O(\|G\|_{\max} N^{-1})$. Therefore, for terms $(i), (ii)$, we have

$$
\|\mathbb{E}[(i)]\|_{\max} \leq O(\|G\|_{\max}^2) = o(\|G\|_{\max}) \tag{50}
$$

$$
\|\mathbb{E}[(ii)]\|_{\max} \leq O(\|G\|_{\max}/N) = o(1/N) \tag{51}
$$

For terms $(iii)$ without $G$, we have

$$
\begin{aligned}
&\|\mathbb{E}[(iii)]\|_{\max}\\
=&\Bigg\|\mathbb{E}\Bigg[\sum_{k,l,n=1}^{c}\frac{\partial^2\zeta_k(a)}{\partial a_l\partial a_n}c^2 h_l h_n\mu_l^\top\Lambda^{-1}q\mu_n^\top\Lambda^{-1}q\mu_k q^\top/2+\sum_{k,l=1}^{c}\frac{\partial^2\zeta_k(a)}{\partial a_l^2}g_l^\top\Lambda^{-1}qg_l^\top\Lambda^{-1}q\mu_k q^\top/2\\
&+\sum_{k,l=1}^{c}\frac{\partial^2\zeta_k(a)}{\partial a_l\partial a_k}ch_l\mu_l^\top\Lambda^{-1}qg_k^\top\Lambda^{-1}qg_k q^\top+\sum_{k,l,n=1}^{c}\frac{\partial^2\zeta_k(a)}{\partial a_l\partial a_n}c^3 h_l h_n h_k\mu_l^\top\Lambda^{-1}q\mu_n^\top\Lambda^{-1}q\mu_k q^\top/2\Bigg]\Bigg\|_{\max}\\
\leq&\frac{1}{2N}\Bigg\|\mathbb{E}\Bigg[\sum_{k,l,n=1}^{c}\frac{\partial^2\zeta_k(a)}{\partial a_l\partial a_n}(c\delta_{ln}-1)\mu_l^\top\Lambda^{-1}q\mu_n^\top\Lambda^{-1}q\mu_k q^\top+\sum_{k,l=1}^{c}\frac{\partial^2\zeta_k(a)}{\partial a_l^2}cq^\top\Lambda^{-1}q\mu_k q^\top\Bigg]\Bigg\|_{\max}\\
&+O(1/N^2)
\end{aligned}
\tag{52}
$$

where the last inequity is due to Lemma F.1, F.2.

For the third term $\sum_{k,l,n,m=1}^{c}R_{klnm}(a,b)b_l b_n b_m p_k q^\top/3!$, we have

$$
\begin{aligned}
&\Bigg\|\mathbb{E}\Bigg[\sum_{k,l,n,m=1}^{c}R_{klnm}(a,b)b_l b_n b_m p_k q^\top/3!\Bigg]\Bigg\|_{\max}\\
\leq&O(1)\max_{l,m\in[d]}\mathbb{E}\Big[\sum_{k_1,k_2,k_3,k_4\in[c]}|b_{k_1}b_{k_2}b_{k_3}(p_{k_4})_l q_m|\Big]\\
\leq&O(1)\mathbb{E}\sum_{k_1,k_2,k_3,k_4\in[c]}\Bigg[\underbrace{\sum_{\phi_1\in\{\mu_{k_1},ch_{k_1}\mu_{k_1},g_{k_1}\},\phi_2\in\{\mu_{k_2},ch_{k_2}\mu_{k_2},g_{k_2}\},\phi_3\in\{\mu_{k_3},ch_{k_3}\mu_{k_3},g_{k_3}\}}\phi_1^\top Gq\phi_2^\top Gq\phi_3^\top Gq(p_{k_4})_l q_m}_{(*)}\\
&+\underbrace{\sum_{\phi_1\in\{\mu_{k_1},ch_{k_1}\mu_{k_1},g_{k_1}\},\phi_2\in\{\mu_{k_2},ch_{k_2}\mu_{k_2},g_{k_2}\},\phi_3\in\{ch_{k_3}\mu_{k_3},g_{k_3}\}}\phi_1^\top Gq\phi_2^\top Gq\phi_3^\top\Lambda^{-1}q(p_{k_4})_l q_m}_{(*)}\\
&+\underbrace{\sum_{\phi_1\in\{\mu_{k_1},ch_{k_1}\mu_{k_1},g_{k_1}\},\phi_2\in\{ch_{k_2}\mu_{k_2},g_{k_2}\},\phi_3\in\{ch_{k_3}\mu_{k_3},g_{k_3}\}}\phi_1^\top Gq\phi_2^\top\Lambda^{-1}q\phi_3^\top\Lambda^{-1}q(p_{k_4})_l q_m}_{(**)}\\
&+\underbrace{\sum_{\phi_1\in\{ch_{k_1}\mu_{k_1},g_{k_1}\},\phi_2\in\{ch_{k_2}\mu_{k_2},g_{k_2}\},\phi_3\in\{ch_{k_3}\mu_{k_3},g_{k_3}\}}\phi_1^\top\Lambda^{-1}q\phi_2^\top\Lambda^{-1}q\phi_3^\top\Lambda^{-1}q(p_{k_4})_l q_m}_{(***)}\Bigg].
\end{aligned}
$$

For terms in $(*)$ having two or three $G$, these terms' expected absolute values are at most smaller than $O(\|G\|_{\max}^2)$. For terms in $(**)$ having one $G$, these terms must contain $n_{1j}$ number of $h_j$ and $n_{2ji}$ number of $(\bar{g}_j)_i$, we have $\sum_{j\in[c],i\in[d]}n_{1j}+n_{2ji}=n_t,n_t=2,3,4$. According to Lemma F.2, for $n_t=2,3,4$, we have

$$
\mathbb{E}\Big[\prod_{j\in[c],i\in[d]}|h_k^{n_{1k}}(\bar{g}_j)_i^{n_{2ji}}|\Big]=O(N^{-n_t/2})=O(N^{-1})
$$

Thus, these term's expected absolute values are at most smaller than $O(\|G\|_{\max}N^{-1})$. For terms in $(***)$ without $G$, these terms must contain $n_{1j}$ number of $h_j$ and $n_{2ji}$ number of $(\bar{g}_j)_i$, we have $\sum_{j\in[c],i\in[d]}n_{1j}+n_{2ji}=n_t,n_t=3,4$. According to Lemma F.2, for $n_t=3,4$, we have

$$
\mathbb{E}\Big[\prod_{j\in[c],i\in[d]}|h_k^{n_{1k}}(\bar{g}_j)_i^{n_{2ji}}|\Big]=O(N^{-n_t/2})=O(N^{-3/2})
$$

Thus, these term's expected absolute values are at most smaller than $O(N^{-3/2})$. Therefore, we have

$$\left\| \mathbb{E}\left[ \sum_{k,l,n,m=1}^{c} R_{klnm}(a,b)b_l b_n b_m p_k q^\top / 3! \right] \right\|_{\max}$$
$$\leq O(1) \max_{l,m\in[d]} \mathbb{E}[ \sum_{k_1,k_2,k_3,k_4\in[c]} |b_{k_1} b_{k_2} b_{k_3}(p_{k_4})_l q_m|]$$
$$\leq O(\|G\|_{max}^2) + O(\|G\|_{\max} N^{-1}) + O(N^{-3/2})$$
$$\leq o(\|G\|_{\max}) + o(1/N). \tag{53}$$

Moreover, we have

$$\left\{ \mathbb{E}\left[ \sum_{k,l=1}^{c} \frac{\partial \zeta_k(a)}{\partial a_l} \mu_k \mu_l^\top G q q^\top \right] \right\}_{ij}$$
$$= \left\{ \mathbb{E}\left[ \sum_{k=1}^{c} \zeta_k(a)(1-\zeta_k(a))\mu_k \mu_k^\top G q q^\top - \sum_{k,l=1,k\neq l}^{c} \zeta_k(a)\zeta_l(a)\mu_k \mu_l^\top G q q^\top \right] \right\}_{ij}$$
$$= \left\{ \mathbb{E}\left[ \sum_{k,l=1,k\neq l}^{c} \zeta_k(a)\zeta_l(a)\mu_k(\mu_k - \mu_l)^\top G q q^\top \right] \right\}_{ij}$$
$$= \left\{ \mathbb{E}\left[ \sum_{k=2}^{c}\sum_{l=1}^{k-1} \zeta_k(a)\zeta_l(a)(\mu_k - \mu_l)(\mu_k - \mu_l)^\top G q q^\top \right] \right\}_{ij}$$
$$= \sum_{n,m=1}^{d} s_{ijnm} G_{nm}, \tag{54}$$

where $s_{ijnm} = \mathbb{E}\left[ \sum_{k=2}^{c}\sum_{l=1}^{k-1} \zeta_k(a)\zeta_l(a)(\mu_k - \mu_l)_i(\mu_k - \mu_l)_n q_m q_j \right]$. We vectorize $G$ as $\text{Vec}(G)_i = G_{t_1(i),t_2(i)}$. Define $S \in \mathbb{R}^{d^2 \times d^2}$, where $S_{ij} = s_{t_1(i),t_2(i),t_1(j),t_2(j)} = \mathbb{E}\left[ \sum_{k=2}^{c}\sum_{l=1}^{k-1} \zeta_k(a)\zeta_l(a)(\mu_k - \mu_l)_{t_1(i)} q_{t_2(i)}(\mu_k - \mu_l)_{t_1(j)} q_{t_2(j)} \right]$, (54) can be expressed as

$$\mathbb{E}\left[ \sum_{k,l=1}^{c} \frac{\partial \zeta_k(a)}{\partial a_l} \mu_k \mu_l^\top G q q^\top \right] = SG. \tag{55}$$

Note that $S = c^2 \nabla^2 \widetilde{L}(c\Lambda^{-1})$. According to Lemma F.3, $S$ is positive definite. Thus, combining (47), (48), (49), (50), (51), (52), (53), (55), we have

$$\|G\|_{\max}$$
$$\leq \frac{1}{N} \left\| S^{-1} \mathbb{E}\left[ \sum_{k,l=1}^{c} \frac{\partial \zeta_k(a)}{\partial a_l}(c\delta_{kl} - 1)\mu_k \mu_l^\top \Lambda^{-1} q q^\top + \sum_{k=1}^{c} \frac{\partial \zeta_k(a)}{\partial a_k} c q q^\top \right.\right.$$
$$\left.\left. + \sum_{k,l,n=1}^{c} \frac{\partial^2 \zeta_k(a)}{\partial a_l \partial a_n}(c\delta_{ln} - 1)\mu_l^\top \Lambda^{-1} q \mu_n^\top \Lambda^{-1} q \mu_k q^\top / 2 + \sum_{k,l=1}^{c} \frac{\partial^2 \zeta_k(a)}{\partial a_l^2} c q^\top \Lambda^{-1} q \mu_k q^\top / 2 \right] \right\|_{\max}$$
$$+ o(1/N).$$

Ignoring constants other than $c, N$, we have $\|G\|_{\max} \leq O(c/N)$. $\qquad \square$

**Lemma F.6.** *The loss function* (7) *is $l$-smooth, where $l \leq \frac{1}{c^2} \sum_{k=2}^{c} \sum_{l=1}^{k-1} \sum_{i\in[d^2]} \mathbb{E}[((p_k - p_l)_{t_1(i)} q_{t_2(i)})^2]$.*

*Proof.* The Hessian matrix of the loss function is

$$
(\nabla^2 L(W))_{ij} = \mathbb{E}\left[\sum_{k=2}^{c}\sum_{l=1}^{k-1} \text{softmax}(P^\top W q/c)_k \text{softmax}(P^\top W q/c)_l (p_k - p_l)_{t_1(i)} q_{t_2(i)} (p_k - p_l)_{t_1(j)} q_{t_2(j)}/c^2\right].
$$

Considering $z \in \mathbb{R}^{d^2}$ such that $z \neq 0$, we have

$$
z^\top \nabla^2 L(W) z
$$

$$
= \mathbb{E}\left[\frac{1}{c^2}\sum_{k=2}^{c}\sum_{l=1}^{k-1} \text{softmax}(P^\top W q/c)_k \text{softmax}(P^\top W q/c)_l \left(\sum_{a\in[d^2]} z_a (p_k - p_l)_{t_1(a)} q_{t_2(a)}\right)^2\right]
$$

$$
\overset{(a)}{\leq} \frac{1}{c^2}\|z\|_2^2 \sum_{k=2}^{c}\sum_{l=1}^{k-1}\sum_{i\in[d^2]} \mathbb{E}[((p_k - p_l)_{t_1(i)} q_{t_2(i)})^2]
$$

where $(a)$ is due to the Cauchy–Schwarz inequality. Thus, $\nabla^2 L(W) \preceq l I_d$ and $L(W)$ is $l$-smooth, where $l$ is a constant smaller than $\frac{1}{c^2}\sum_{k=2}^{c}\sum_{l=1}^{k-1}\sum_{i\in[d^2]} \mathbb{E}[((p_k - p_l)_{t_1(i)} q_{t_2(i)})^2]$. $\square$

**Theorem F.7** (Formal statement of Theorem 4.3). *The following statements hold.*

*(1) Optimizing training loss $L(W)$ (16) with training prompt length $N$ via gradient descent $W^{t+1} = W^t - \eta\nabla L(W^t)$, we have for any $t$*

$$
\|W^t - W^*\|_F^2 \leq \exp(-t/\kappa)\|W^0 - W^*\|_F^2,
$$

*where $W^0$ is the initial parameter and $W^*$ is the global minimizer of $L(W)$, $\kappa = l/\alpha$. $\alpha, l$ are constants such that*

$$
0 < \alpha \leq \lambda_{\min}(\nabla^2 L(W)) \leq \lambda_{\max}(\nabla^2 L(W)) \leq l, \text{ for all } W \in R_W, \tag{56}
$$

*where $R_W = \{W \in \mathbb{R}^{d\times d} \mid \|W - W^*\|_F \leq \|W^0 - W^*\|_F\}$.*

*(2) Denoting $W^* = c(\Lambda^{-1} + G)$, we have*

$$
\|G\|_{\max} \leq \frac{1}{N}\left\|S^{-1}\mathbb{E}\left[\sum_{k,l=1}^{c} \frac{\partial\zeta_k(a)}{\partial a_l}(c\delta_{kl} - 1)\mu_k\mu_l^\top \Lambda^{-1} q q^\top + \sum_{k=1}^{c} \frac{\partial\zeta_k(a)}{\partial a_k} c q q^\top\right.\right.
$$

$$
\left.\left. + \frac{1}{2}\sum_{k,l,n=1}^{c} \frac{\partial^2\zeta_k(a)}{\partial a_l \partial a_n}(c\delta_{ln} - 1)\mu_l^\top \Lambda^{-1} q \mu_n^\top \Lambda^{-1} q \mu_k q^\top + \frac{1}{2}\sum_{k,l=1}^{c} \frac{\partial^2\zeta_k(a)}{\partial a_l^2} c q^\top \Lambda^{-1} q \mu_k q^\top\right]\right\|_{\max}
$$

$$
+ o(1/N)
$$

$$
= O(c/N)
$$

*where $S = c^2\nabla^2\widetilde{L}(2\Lambda^{-1})$, $\widetilde{L}(2\Lambda^{-1}) = \lim_{N\to\infty} L(2\Lambda^{-1})$. The expectation is taken over $\mu_\tau \sim \mathcal{P}_\Omega^m(\Lambda)$, $x_{\tau,\text{query}} \sim \mathcal{P}_x^m(\mu_\tau, \Lambda)$.*

*(3) After $T \geq 2\kappa\log(N \cdot \|W^0 - W^*\|_F)$ gradient steps, denoting $\widehat{W}$ as the final model, we have*

$$
\widehat{W} = c(\Lambda^{-1} + \widehat{G}), \tag{57}
$$

*where $\|\widehat{G}\|_{\max} = O(c/N)$.*

### F.3. Proof of Theorem 4.3

*Proof.* According to Lemma F.5, the global minimizer of $L(W)$ is $W^* = c(\Lambda^{-1} + G)$, where

$$\|G\|_{\max}$$
$$\leq \frac{1}{N} \left\| S^{-1} \mathbb{E} \left[ \sum_{k,l=1}^{c} \frac{\partial \zeta_k(a)}{\partial a_l} (c\delta_{kl} - 1)\mu_k \mu_l^\top \Lambda^{-1} qq^\top + \sum_{k=1}^{c} \frac{\partial \zeta_k(a)}{\partial a_k} cqq^\top \right. \right.$$
$$\left. \left. + \sum_{k,l,n=1}^{c} \frac{\partial^2 \zeta_k(a)}{\partial a_l \partial a_n} (c\delta_{ln} - 1)\mu_l^\top \Lambda^{-1} q\mu_n^\top \Lambda^{-1} q\mu_k q^\top/2 + \sum_{k,l=1}^{c} \frac{\partial^2 \zeta_k(a)}{\partial a_l^2} cq^\top \Lambda^{-1} q\mu_k q^\top/2 \right] \right\|_{\max}$$
$$+ o(1/N).$$

Ignoring constants other than $c, N$, we have $\|G\|_{\max} \leq O(c/N)$.

Define $R_W = \{W \in \mathbb{R}^{d \times d} \mid \|W - W^*\|_F \leq \|W^0 - W^*\|_F\}$, and $R_W$ is a compact set. Then, according to Lemma F.3, for $W \in R_W$, we have $\nabla^2 L(W) \succeq \alpha I_d$. Here $\alpha > 0$ is a positive constant number. Thus, $L(W)$ is $\alpha$-strongly convex in $R_W$. Moreover, according to Lemma F.6, $L(W)$ is $l$-smooth. Then according to Lemma C.2, applying gradient descent with $\eta = 1/l$, for any $t \geq 1$, we have

$$\|W^t - W^*\|_F^2 \leq \exp(-t/\kappa) \cdot \|W^0 - W^*\|_F^2,$$

where $\kappa = l/\alpha$.

After $T \geq 2\kappa \log(N \cdot \|W^0 - W^*\|_F)$ gradient steps, we have $\widehat{W} = W^T = c(\Lambda^{-1} + G + H^T/c) = 2(\Lambda^{-1} + \widehat{G})$, where $\widehat{G} = G + H^T/c$, $\|H^T\|_{\max} \leq \exp(-T/\kappa) \cdot \|W^0 - W^*\|_F^2 \leq 1/N$. Thus, $\|\widehat{G}\|_{\max} \leq \|G\|_{\max} + \|H^T\|_{\max} = O(c/N)$. $\square$

## G. In-context inference of multi-class classification

### G.1. Notations

In this section, we use the following notations. We denote $\mu = (\mu_1, \mu_2, \ldots, \mu_c)$, $q = x_{\mathsf{query}}$. Define $p_k = \frac{c}{M} \sum_{i=1}^{M} (y_i)_k x_i$, and define $P = (p_1, p_2, \ldots, p_c) \in \mathbb{R}^{d \times c}$. We have $P^\top = \frac{c}{M} \sum_{i=1}^{M} y_i x_{\tau,i}^\top \in \mathbb{R}^{c \times d}$. Since with probability $\mathbb{P}(y_{\tau,i} = \mathbf{e}_k) = 1/c$, $x_{\tau,i} = \mu_k + v_i$, where $v_i \sim \mathsf{N}(0, \Lambda)$, we have $p_k = \frac{c}{M} \sum_{i=1}^{M} (y_{\tau,i})_k x_{\tau,i} = cM_k \mu_k/M + g_k$, where $g_k = \frac{c}{M} \sum_{i \in \{i|y_{\tau,i} = \mathbf{e}_k\}} v_i$, $g_k \sim \mathsf{N}(0, c^2 M_k \Lambda/M^2)$ and $(M_1, M_2, \ldots, M_c) \sim \mathsf{Multin}(M, 1/c)$. Defining $h_k = M_k/M - 1/c$, we have $M_k/M = 1/c + h_k$ and $p_k = \mu_k + ch_k\mu_k + g_k$.

**Theorem G.1** (Formal statement of Theorem 4.5). *Let $\widehat{y}_{\mathsf{query}}$ be the prediction of the trained transformer with parameters $\widehat{W}$ in (19) and $P_{\mathsf{test}}$ satisfying Assumption 4.4, and let $y_{\mathsf{query}} \sim \mathcal{P}_{y|x_{\mathsf{query}}}^m(\mu, \Lambda)$. Then, for the inference error defined in (3), we have*

$$\mathbb{E}[\Delta(y_{\mathsf{query}}, \widehat{y}_{\mathsf{query}})]$$
$$\leq \max_{k \in [c]} \left\{ \sum_{l=1}^{c} \frac{\partial \zeta_k(a)}{\partial a_l} \left[ \|\widehat{G}\|_{\max} \sum_{i,j \in [d]} |(\mu_l)_i q_j| + \frac{1}{M^{1/2}} \left( \sqrt{c(1 - 1/c)} |\mu_l^\top \Lambda^{-1} q| + \sqrt{c} \sum_{i,j \in [d]} |\Lambda_{ij}^{-1/2} q_j| \right) \right] \right\}$$
$$+ o\left( \frac{1}{N} + \frac{1}{\sqrt{M}} \right),$$

*where $a = \mu^\top \Lambda^{-1} q$, $a_k = \mu_k^\top \Lambda^{-1} q$. The expectation is taken over $\{x_i, y_i\}_{i=1}^{M} \overset{\text{i.i.d.}}{\sim} \mathcal{P}^m(\mu, \Lambda)$.*

### G.2. Proof of Theorem 4.5

*Proof.* The output of the trained transformer is

$$\widehat{y}_{\mathsf{out}} = \mathsf{softmax}\left( \left( \frac{c}{M} \sum_{i=1}^{M} y_i x_i^\top \right) (\Lambda^{-1} + \widehat{G}) x_{\mathsf{query}} \right) = \mathsf{softmax}(P^\top (\Lambda^{-1} + \widehat{G})q) \tag{58}$$

The probability of $y_{\mathsf{query}} = \mathbf{e}_k$ given $x_{\mathsf{query}}$ is

$$\mathbb{P}\left(y_{\mathsf{query}} = \mathbf{e}_k | x_{\mathsf{query}}\right) = \mathrm{softmax}(\mu^\top \Lambda^{-1} x_{\mathsf{query}})_k = \mathrm{softmax}(\mu^\top \Lambda^{-1} q)_k$$

Defining $a = \mu^\top \Lambda^{-1} q$, $b = (\mu + \mu_h + g)^\top \widehat{G} q + (\mu_h + g)^\top \Lambda^{-1} q$, $a_k = \mu_k^\top \Lambda^{-1} q$, $b_k = (\mu_k + ch_k\mu_k + g_k)^\top \widehat{G} q + (ch_k\mu_k + g_k)^\top \Lambda^{-1} q$, we have

$$\mathbb{E}\left[\mathrm{softmax}(P^\top(\Lambda^{-1} + \widehat{G})q)_k\right] = \mathbb{E}\left[\zeta_k(a + b)\right] = \mathbb{E}[\zeta_k(a) + \sum_{l=1}^{c} \frac{\partial \zeta_k(a)}{\partial a_l} b_l + \sum_{l,n=1}^{c} R_{kln}(a, b) b_l b_n / 2]$$

where $|R_{kln}(a, b)| \leq \sup_x |\frac{\partial^2 \zeta_k(x)}{\partial x_l \partial x_n}|$. Thus, we have

$$\mathbb{E}[|\zeta_k(a + b) - \zeta_k(a)|] \leq \mathbb{E}\left[\sum_{l=1}^{c} \left|\frac{\partial \zeta_k(a)}{\partial a_l} b_l\right|\right] + \mathbb{E}\left[\left|\sum_{l,n=1}^{c} R_{kln}(a, b) b_l b_n / 2\right|\right].$$

We first consider the term $\mathbb{E}\left[\sum_{l=1}^{c} \left|\frac{\partial \zeta_k(a)}{\partial a_l} b_l\right|\right]$. Defining $\bar{g}_l = \Lambda^{-1/2} g_l$, we have

$$\mathbb{E}\left[\sum_{l=1}^{c} \left|\frac{\partial \zeta_k(a)}{\partial a_l} b_l\right|\right]$$

$$\leq \sum_{l=1}^{c} \frac{\partial \zeta_k(a)}{\partial a_l} \left(|\mu_l^\top \widehat{G} q| + \mathbb{E}[|ch_l\mu_l^\top \widehat{G} q|] + \mathbb{E}[|g_l^\top \widehat{G} q|] + \mathbb{E}[|ch_l\mu_l^\top \Lambda^{-1} q|] + \mathbb{E}[|g_l^\top \Lambda^{-1} q|]\right)$$

$$\overset{(a)}{\leq} \sum_{l=1}^{c} \frac{\partial \zeta_k(a)}{\partial a_l} \left(|\mu_l^\top \widehat{G} q| + \frac{\sqrt{c(1 - 1/c)}}{M^{1/2}}|\mu_l^\top \widehat{G} q| + \mathbb{E}[|\bar{g}_l^\top \Lambda^{1/2} \widehat{G} q|] + \frac{\sqrt{c(1 - 1/c)}}{M^{1/2}}|\mu_l^\top \Lambda^{-1} q| + \mathbb{E}[|\bar{g}_l^\top \Lambda^{-1/2} q|]\right)$$

$$\overset{(b)}{\leq} \sum_{l=1}^{c} \frac{\partial \zeta_k(a)}{\partial a_l} \left[\|\widehat{G}\|_{\max} \sum_{i,j\in[d]} |(\mu_l)_i q_j| + \frac{1}{M^{1/2}} \left(\sqrt{c(1 - 1/c)}|\mu_l^\top \Lambda^{-1} q| + \sqrt{c} \sum_{i,j\in[d]} |\Lambda_{ij}^{-1/2} q_j|\right)\right]$$

$$+ o\left(\frac{1}{N} + \frac{1}{\sqrt{M}}\right),$$

where $(a)$ is due to Lemma F.1 that $\mathbb{E}[|h|] \leq M^{-1/2} c^{-1/2}(1 - 1/c)^{1/2}$. $(b)$ is because that $\bar{g}_l \sim \mathsf{N}(0, c^2 M_l I_d / M^2)$, $\mathbb{E}[|(\bar{g}_l)_i|] \leq \mathbb{E}[(\bar{g}_l)_i^2]^{1/2} = (c/M)^{1/2}$, for $l \in [c], i \in [d]$.

For $\mathbb{E}\left[\left|\sum_{l,n=1}^{c} R_{kln}(a, b) b_l b_n / 2\right|\right]$, we have

$$\mathbb{E}\left[\left|\sum_{l,n=1}^{c} R_{kln}(a, b) b_l b_n / 2\right|\right] = O(1)\mathbb{E}\left[\sum_{l,n=1}^{c} \left(\underbrace{\sum_{\phi_1 \in \{\mu_l, ch_l\mu_l, g_l\}, \phi_2 \in \{\mu_n, ch_n\mu_n, g_n\}} \left|\phi_1^\top \widehat{G} q \phi_2^\top \widehat{G} q\right|}_{(i)}\right.\right.$$

$$\left.\left. + \underbrace{\sum_{\phi_1 \in \{\mu_l, ch_l\mu_l, g_l\}, \phi_2 \in \{ch_n\mu_n, g_n\}} \left|2\phi_1^\top \widehat{G} q \phi_2^\top \Lambda^{-1} q\right|}_{(ii)} + \underbrace{\sum_{\phi_1 \in \{ch_l\mu_l, g_l\}, \phi_2 \in \{ch_n\mu_n, g_n\}} \left|\phi_1^\top \Lambda^{-1} q \phi_2^\top \Lambda^{-1} q\right|}_{(iii)}\right)\right].$$

For terms $(i)$ having two $\widehat{G}$, they are at most smaller than $O(\|\widehat{G}\|_{\max}^2) = O(1/N^2)$. For terms $(ii)$ having one $G$, these terms must contain $n_{1j}$ number of $h_j$ and $n_{2ji}$ number of $(\bar{g}_j)_i$, we have $\sum_{j\in[c], i\in[d]} n_{1j} + n_{2ji} = n_t, n_t = 1, 2$. According to Lemma F.2, we know that for $n_t = 1, 2$,

$$\mathbb{E}[\prod_{j\in[c], i\in[d]} |h_j^{n_{1j}} (\bar{g}_j)_i^{n_{2ji}}|] = O(M^{-1/2}).$$

Thus, terms in (ii) are at most smaller than $O(\|G\|_{\max}M^{-1/2}) = O(1/(N\sqrt{M}))$. For terms $(iii)$ without $G$, these terms must contain $n_{1j}$ number of $h_j$ and $n_{2ji}$ number of $(\bar{g}_j)_i$, we have $\sum_{j\in[c],i\in[d]} n_{1j} + n_{2ji} = n_t, n_t = 2$. According to Lemma F.2, for $n_t = 2$, we have

$$\mathbb{E}[\prod_{j\in[c],i\in[d]} |h_k^{n_{1k}}(\bar{g}_j)_i^{n_{2ji}}|] = O(M^{-n_t/2}) = O(M^{-1}).$$

Thus, these term are $O(M^{-1})$. Therefore, we have $\mathbb{E}\left[\left|\sum_{l,n=1}^c R_{kln}(a,b)b_lb_n/2\right|\right] = O(1/N^2 + 1/M + 1/(N\sqrt{M})) = o(1/N + 1/\sqrt{M})$.

Finally, we have

$$\mathbb{E}[\Delta(y_{\mathsf{query}}, \widehat{y}_{\mathsf{query}})] = \max_k\{\mathbb{E}[|\mathrm{softmax}(a+b)_k - \mathrm{softmax}(a)_k|]\}$$

$$\leq \max_{k\in[c]} \left\{ \sum_{l=1}^c \frac{\partial \zeta_k(a)}{\partial a_l} \left[ \|\widehat{G}\|_{\max} \sum_{i,j\in[d]} |(\mu_l)_i q_j| + \frac{1}{M^{1/2}} \left( \sqrt{c(1-1/c)}|\mu_l^\top \Lambda^{-1} q| + \sqrt{c} \sum_{i,j\in[d]} |\Lambda_{ij}^{-1/2} q_j| \right) \right] \right\}$$
$$+ o\left( \frac{1}{N} + \frac{1}{\sqrt{M}} \right).$$

$\square$

*Remark* G.2. We note that Theorem 4.5 requires Assumption 4.4 to hold. For example, we need the covariance $\Lambda$ in training and testing to be the same. A similar consistency requirement of the covariance $\Lambda$ in training and testing had also been observed for in-context linear regression in Zhang et al. (2023a) and for in-context binary classification in the previous section 3.2.

Here, we discuss the consequences when Assumption 4.4 does not hold. For example, suppose the labels of our data in test prompts are not balanced $\mathbb{P}(y = \mathbf{e}_k) = p_k$, $\mu$ do not have the same $\Lambda^{-1}$ weighted norm $\mu_k^\top \Lambda^{-1} \mu_k \triangleq \Psi_k$, and the covariance matrix of test data is $\Gamma \neq \Lambda$, then as $N, M \to \infty$, we have

$$\frac{c}{M} \sum_{i=1}^M y_i x_i^\top \to c(p_1\mu_1, p_2\mu_2, \ldots, p_c\mu_c)^\top,$$

and

$$\mathbb{P}(\widehat{y}_{\mathsf{query}} = 1) \to \mathrm{softmax}(c(p_1\mu_1, p_2\mu_2, \ldots, p_c\mu_c)^\top \Lambda^{-1} x_{\mathsf{query}}).$$

Denote $\Psi = (\Psi_1, \ldots, \Psi_c)^\top$, $\Phi = (\log(p_1), \ldots, \log(p_c))^\top$ and $z = \mu^\top \Gamma^{-1} x_{\mathsf{query}} - \Psi/2 + \Phi$. Then distribution of the ground truth label is

$$\mathbb{P}(y_{\mathsf{query}} = \mathbf{e}_k) = \mathrm{softmax}(z)_k.$$

Define $\hat{z} = c(p_1\mu_1, p_2\mu_2, \ldots, p_c\mu_c)^\top \Lambda^{-1} x_{\mathsf{query}}$. Then, unless $\hat{z} = z$ or $\|\mathrm{softmax}(\hat{z}) - \mathrm{softmax}(z)\|_2$ is sufficiently small, the transformer cannot correctly perform the in-context multi-class classification.

## H. Additional Experiments

In this section, we provide additional experimental results and the detailed experimental settings.

### H.1. Single-layer transformers

We train single-layer transformers for in-context classification of Gaussian mixtures with different numbers of classes $c$, different lengths of training prompts $N$, and test them with different test prompt lengths $M$. The results are reported in Figure 4. We can see from Figure 4 (a,b) that the inference errors decrease as $N$ and $M$ increase, and they increase as $c$ increases. In Figure 4 (c,d), we first fix the training prompt length (test prompt length) to a large number 2000, and then vary the test prompt length (training prompt length) from 20 to 2000. The results show that, as $M$ and $N$ become sufficiently large, the inference error, which is an approximation of $\mathbb{E}[\Delta(y_{\mathsf{query}}, \widehat{y}_{\mathsf{query}})]$ (see Appendix H.2 for detailed definitions), decreases to near-zero. This indicates that the prediction of the trained transformer approaches the Bayes-optimal classifier. All these experimental results corroborate our theoretical claims.

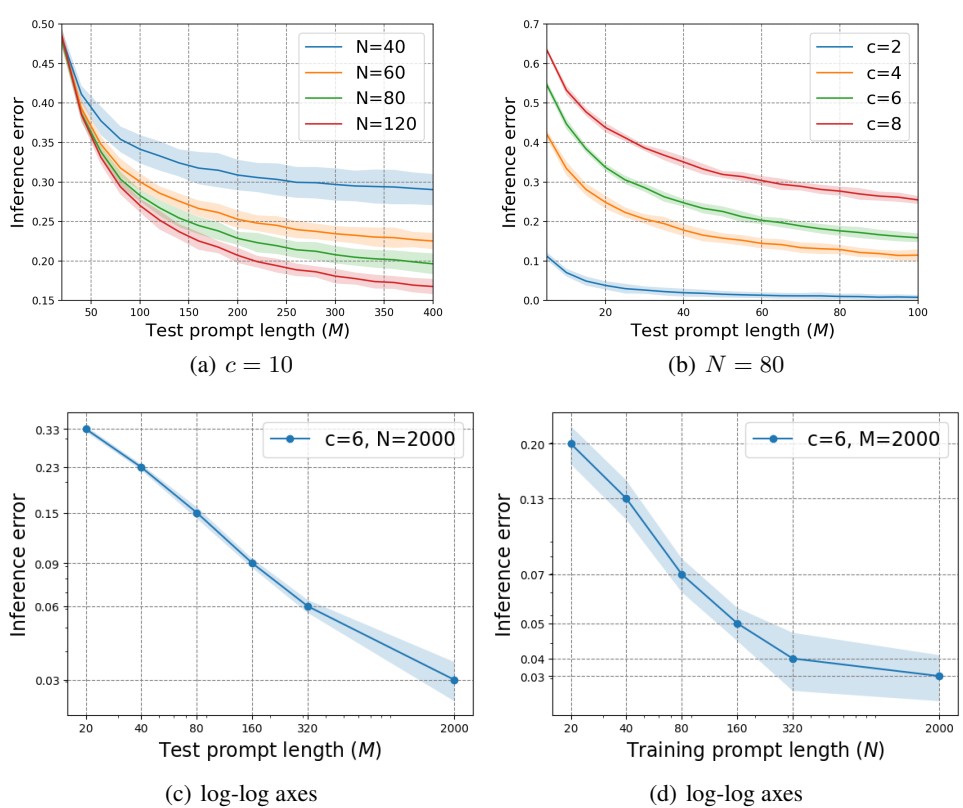

**(a) $c = 10$**

**(b) $N = 80$**

**(c) log-log axes**

**(d) log-log axes**

*Figure 4.* Inference errors of single-layer transformers. (a): Models trained on different training prompt lengths $N$ on classification tasks involving $c = 10$ classes. (b): Models trained on different classification tasks involving $c$ classes with a fixed training prompt length $N = 80$. (c): Relationship between the inference error and the test prompt length $M$ in log-log axes. Training prompt length $N = 2000$ and number of classes $c = 6$. (d): Relationship between the inference error and the training prompt length $N$ in log-log axes. Test prompt length $M = 2000$ and number of classes $c = 6$.

## H.2. Experiment Details

For all tasks, we set $d = 20$ and we randomly generate a covariance matrix $\Lambda = \text{diag}(\lambda_1, \ldots, \lambda_d)$, where $\lambda_i = |\hat{\lambda}_i|$ and $\hat{\lambda}_i \overset{\text{i.i.d.}}{\sim} \mathsf{N}(3, 1)$. For each training dataset with different training prompt lengths $N$, and different class numbers $c$, we randomly generate $B$ training samples. Training prompts $P_\tau, \tau \in [B]$ and their corresponding labels $y_{\tau,\text{query}}$ are generated according to Assumption 4.2. Moreover, we also generate testing datasets. For example, for each testing dataset, we first randomly generate 20 pairs of $(\mu_j, x_{j,\text{query}}, y_{j,\text{prob}}), j \in [20]$, where $(\mu_j) \overset{\text{i.i.d.}}{\sim} \mathcal{P}_\Omega^m(\Lambda)$, $x_{j,\text{query}} \sim \mathcal{P}_x^m(\mu_j, \Lambda)$. $y_{j,\text{prob}} = \text{softmax}(\mu_j^\top \Lambda^{-1} x_{j,\text{query}})$ are the corresponding probability distributions of the ground truth label $y_{j,\text{query}}$. For each $j$, we generate 100 testing prompts $P_{jk} = (x_{jk,1}, y_{jk,1}, \ldots, x_{jk,M}, y_{jk,M}, x_{j,\text{query}})$, where $(x_{jk,i}, y_{jk,i}) \overset{\text{i.i.d.}}{\sim} \mathcal{P}^m(\mu_j, \Lambda), j \in [20], k \in [100], i \in [M]$. We denote a model's output for testing prompts $P_{jk}$ as $\hat{y}_{jk}$. We calculate its inference error with $\frac{1}{20 \times 100} \sum_{j \in [20], k \in [100]} \max_{l \in [c]} \left|(\hat{y}_{jk})_l - (y_{j,\text{prob}})_l\right|$, which serves an approximation of the expected total variation distance we defined in (3).

For the '3-layer' model, we used the x-transformers library and defined it as an encoder-only transformer with 64 embedding sizes, 3 layers, 2 heads and without positional encoding.

For experiments in Figure 1, we set the size of the training dataset to $B = 100,000$ and set the batch size to 50. We train the '1-layer' using Adam with learning rate 0.0005 for 10 epochs, and train the '3-layer' using Adam with learning rate 0.0001 for 5 epochs. Each experiment is repeated 3 times with different random seeds. For experiments in Figure 2, we also set the size of the training dataset to $B = 100,000$ and set the batch size to 50. We train the '1-layer' using Adam with learning rate 0.001 for 5 epochs, and train the '3-layer' using Adam with learning rate 0.0001 for 5 epochs. In 'same norm' and 'same covariance' settings, pre-training data are sampled according to Assumption 4.2 with a fixed $\Lambda$ that $\Lambda = \text{diag}(\lambda_1, \ldots, \lambda_d)$, where $\lambda_i = |\hat{\lambda}_i|$ and $\hat{\lambda}_i \overset{\text{i.i.d.}}{\sim} \mathsf{N}(3, 1)$. In 'different norms' setting, for each $\tau \in [B]$, with probability $\mathbb{P}(k = j) = 1/10, \mu_{\tau,i} \sim \mathsf{N}(k, I_d), j = 0, 1, \ldots, 9$, then each Gaussian component is sampled according to $\mathsf{N}(\mu_{\tau,i}, \Lambda)$. In (different covariances) setting, we randomly generate $v_1, v_2, v_3 \in \mathbb{R}^d$ that half of their elements are 0.1 and the other half elements are 100. Then, we define $\Lambda_i = \text{diag}(v_i), i = 1, 2, 3$ and generate pre-training data according to Assumption 4.2 with $\Lambda, \Lambda_1, \Lambda_2, \Lambda_3$. Each experiment is repeated 3 times with different random seeds. For experiments in Figure 3, the structure of the transformer with full parameters '1-layer, full' is defined as

$$F(E; W^V, W^{KQ}) = E + W^V E \cdot \frac{E^\top W^{KQ} E}{\rho}, \tag{59}$$

where $W^V, W^{KQ} \in \mathbb{R}^{(d+c) \times (d+c)}$ are the parameters for optimization. For all three transformer models, we set the size of the training dataset to $B = 400,000$ and set the batch size to 50. We train the '1-layer, sparse' and '1-layer, full' using Adam with learning rate 0.001 for 5 epochs, and train the 3-layer transformer model with softmax attention using Adam with learning rate 0.0001 for 5 epochs. Each experiment is repeated 3 times with different random seeds. For experiments in Figure 4, we train the single-layer transformers with the sparse-form parameters and structures defined in Section 4. We set the size of the training dataset to $B = 10,000$ and set the batch size to 50. We train the transformers using SGD with learning rate $\{0.1, 0.5, 1\}$ for 10 epochs, and get the best model on each training dataset. Then, we test these trained models on different testing datasets. Each experiment is repeated 10 times with different random seeds.

