# OpenReview forum: "On the Training Convergence of Transformers for In-Context Classification of Gaussian Mixtures"
_ICML.cc/2025/Conference — ICML 2025 poster_

### Official Review · Reviewer_M6vU · 2025-03-12

**Overall Recommendation:** 3

**Summary:**

This work studies the convergence and training dynamics of transformers for in-context classification tasks of Gaussian mixture data. The results show that a single-layer transformer trained by gradient descent converges to the global optimal at a linear rate. A quantification of how the training and testing prompt lengths affect the inference is provided. The analysis can also be extended to multi-classification problems.

--------------------------------
## Update after rebuttal

I appreciate the authors' clarification. I prefer to keep my current rating.

**Claims And Evidence:**

Yes, the analysis is solid to me.

**Essential References Not Discussed:**

N/A

**Experimental Designs Or Analyses:**

N/A. This paper is mainly theoretical.

**Methods And Evaluation Criteria:**

N/A. This paper is mainly theoretical.

**Other Comments Or Suggestions:**

N/A

**Other Strengths And Weaknesses:**

I would like to mention some weaknesses here.

1. Theoretical novelty. The technical contribution beyond [Zhang et al., 2023a] is not clear to me. You consider the classification problem with a different loss function. But why is it challenging enough compared with [Zhang et al., 2023a]. It is better to include related discussions in the paper.

2. The practical insight of this work is not clear. For example, how can the theoretical analysis be used to improve ICL in practice? Are the theoretical conclusions aligned with any existing empirical finding? Or can the theory be used to explain any theoretical finding?

3. The writing needs some improvement. It is better to include some remark after each Theorem to provide an intuitive explanation. This can help readers understand the key point of each result.

Zhang et al., 2023a. Trained transformers learn linear models in-context.

**Questions For Authors:**

Can you prove that $\alpha>0$ in Eqn 18? Otherwise, the convergence analysis is not that strong.

**Relation To Broader Scientific Literature:**

N/A

**Theoretical Claims:**

The theoretical analysis looks rigorous, and the conclusions make sense.

---

> ### Author Rebuttal · Authors · 2025-04-01
>
> >The technical contribution beyond [Zhang et al., 2023a] is not clear to me. You consider the classification problem with a different loss function. But why is it challenging enough compared with [Zhang et al., 2023a]. It is better to include related discussions in the paper.
>
> Compared to [1], we use the cross-entropy loss for the classification problem. Thus, a critical distinction to [1] is that our model does not have a closed-form expression of the global minimizer, while [1] does. This distinction adds a significant challenge and leads to a different technical approach compared to [1]. By analyzing the Taylor expansion at the stationary point, we establish the convergence properties of $W^*$. Moreover, [1] considered optimizing the transformer using gradient flow. In contrast, our work proves the convergence of optimizing the transformer with the more practical gradient descent method. We will add more related discussions in our revised paper.
>
> >The practical insight of this work is not clear. For example, how can the theoretical analysis be used to improve ICL in practice? Are the theoretical conclusions aligned with any existing empirical finding? Or can the theory be used to explain any theoretical finding?
>
> Yes, our theoretical conclusions align with existing empirical findings. For example, in Figure 1, we conducted experiments of single-layer and multi-layer transformers for in-context classification of Gaussian mixtures. We can notice that both transformer models' ICL inference errors decrease as training prompt length ($N$) and test prompt length ($M$) increase, and increase as the number of Gaussian mixtures ($c$) increases. These behaviors are consistent with our theoretical claims. Moreover, the results also indicate that some of our insights obtained from studying this simplified model may hold for transformers with more complex structures, and studying this simplified model can help us have a better understanding of the ICL abilities of transformers. We hope this paper can provide valuable insights into the theoretical understanding of the ICL mechanisms of transformers. These insights may be helpful for potential architectural design and building safe and explainable AI systems.
>
> >The writing needs some improvement. It is better to include some remark after each Theorem to provide an intuitive explanation. This can help readers understand the key point of each result.
>
> Thanks for your suggestion. We will add more marks and intuitive explanations in the revised paper.
>
> **References**:
>
> [1] Ruiqi Zhang, Spencer Frei, and Peter L Bartlett. Trained transformers learn linear models in-context. arXiv preprint arXiv:2306.09927, 2023a.
>
> [8] Yu Huang, Yuan Cheng, and Yingbin Liang. In-context convergence of transformers. arXiv preprint arXiv:2310.05249, 2023.
>
> [9] Hongkang Li, Meng Wang, Songtao Lu, Xiaodong Cui, and Pin-Yu Chen. Training nonlinear transformers for efficient in-context learning: A theoretical learning and generalization analysis. arXiv preprint arXiv:2402.15607, 2024.
>
> [10] Jingfeng Wu, Difan Zou, Zixiang Chen, Vladimir Braverman, Quanquan Gu, and Peter L Bartlett. How many pretraining tasks are needed for in-context learning of linear regression? arXiv preprint arXiv:2310.08391, 2023.

---

> > ### Comment · Reviewer_M6vU · 2025-04-02
> >
> > Thank the authors for the response. I prefer to keep my current rating. The theoretical analysis is solid. The contributions are not that exciting to lead to a higher rating. BTW, I left a question in "Questions For Authors", where I am actually asking whether a positive lower bound of $\alpha$ can be proved. If so, it can make the convergence analysis stronger.
> >
> > ---------------------------
> >
> > Thank you for the further clarification.

---

> > > ### Author Response · Authors · 2025-04-04
> > >
> > > Thanks for your acknowledgment for our contributions.
> > >
> > > >Can you prove that $\alpha>0$ in Eqn 18? Otherwise, the convergence analysis is not that strong.
> > >
> > > Yes. Actually, we have proved it in Lemma D.2 and Lemma F.3. For example, in Lemma D.2, Line 789-802, we proved that, for any compact set $R_W$ of $R^{d\times d}$, for any $W\in R_W$, we have $\nabla^2 L(W)\succ C(\Omega)S(\Omega)I_d/4$ with $C(\Omega)S(\Omega)>0$. The definitions of $C(\Omega)$ and $S(\Omega)$ can be found in Line 772 and 792. Thus, with the initial point $W_0$ and the $R_W$ we defined in Line 1091, we have $\nabla^2 L(W)\succeq \alpha I_d$ with $\alpha>0$. We will make this point more clear in the revised paper.

---

### Official Review · Reviewer_aULa · 2025-03-13

**Overall Recommendation:** 3

**Summary:**

This paper studies the in-context learning (ICL) capabilities of the transformer model. In particular, this paper shows that one layer of linear attention mechanism, after pre-training through gradient descent, can implement classification of Gaussian mixture data. The main results of this paper are the convergence guarantees and test-loss bounds of pre-training a (single-layer, sparse, linear) attention model for both binary and multi-class classification.

Compare to previous results such as [Bai et al. 2023], which only provided a construction, this paper presents rigorous guarantees on the GD pre-training convergence dynamics and additionally studies the case of multi-class classification.

**Claims And Evidence:**

I find the overall writing of this paper to be good and the theoretical claims to be clear. The study of ICL for classification tasks is under-explored in the literature and this paper is a valuable addition towards that direction.

However, I find that the paper is lacking in a few areas:
1. The multi-class case in Section 4 is certainly a worthwhile exercise, but the setting and results are largely direct extension of the binary case in Section 3. I find that there is very little benefit from essentially repeating Section 3. I suggest the author to condense this section, elaborate more on the technical difficulty of the multi-class case, and leave rest of the details to the Appendix.
2. I find Assumptions 3.1 and 3.5 to be slightly problematic. It is okay to assume homo-scedasticity (same $\Lambda$ for both classes), but the assumption on the means having the same $\Lambda^{-1}$-weighted norms is hard to justify. I guess this assumption is made for the convenience of the proof. If so, the authors should be upfront about this and explain why the problem is intractable without this assumption.
3. The discussion of the prediction (around eq (12)) is rather lacking. It seems that the derivation is about to connect the pre-trained attention model to linear/quadratic discriminant analysis (LDA/QDA) but then abruptly stops. Actually, I think it is very helpful to describe what kind of learning algorithm is implemented by the pre-trained transformer. This type of argument is in fact a major selling point of [von Oswald, 2022] and really helped with its popularity. In fact, if I am not wrong, setting $W = 2\Lambda^{-1}$ implements the LDA decision criterion?
4. What happens when you stack multiple layers? We know this helps in the regression setting. Do you have reasons/hypothesis on why more layers does not seem to help much for the classification task?

And some minor points:
1. remarks E.1 and G.2 should be included as part of the main body.
2. The well-conditioned property (9) is proven in the paper, but currently it sounds like an assumption. Please clarify this.

**Essential References Not Discussed:**

I think the coverage of the recent works on ICL is sufficient, but I think adding a few references on the classical methods for Gaussian mixtures, e.g. linear discriminant analysis, would be very helpful.

**Experimental Designs Or Analyses:**

I know this is a theory paper, so the few proof-of-concept experiments in Section 5 are fine. However, I find that the discussion around Figure 3 to be extremely lacking.
1. How many classes are there?
2. For the "inference error" are you referring to the TV distance in equation 3? I don't think this notion is directly applicable to k-nearest neighbor or SVM.
3. I find this comparison to be unfair since the transformer models have been pre-trained on a lot (what is the exact number?) of data, whereas I assume the classical models are directly applied to the test prompt.
4. Why is LDA not part of the classical baseline? This is the perfect setting for LDA.

Also, you should include your source code even if they are short.

**Methods And Evaluation Criteria:**

n/a

**Other Comments Or Suggestions:**

see above

**Other Strengths And Weaknesses:**

see above

**Questions For Authors:**

see above

**Relation To Broader Scientific Literature:**

see above

**Theoretical Claims:**

The theoretical claims and analysis look to be the natural extension of the techniques of [Zhang et. al. 2023], and the authors did a decent job at differentiating their works from the existing results. While I am not a fan of the sparsity assumption (5), almost everyone in the area imposes this assumption. So I will not complain about this too much.

I noticed that in Section D.1, the description of D.3 does not match the actual lemma, and I cannot seem to find a lemma that fits the description. So I would like to see a more elaborate proof sketch that correctly connects different steps of the proof.

Overall, I think the ideas presented in the paper are interesting and valuable. However, I have some reservations regarding 1) the assumptions on the data, 2) connection to classical Gaussian mixture techniques such as LDA, 3) the gap in the proof sketch. I think this paper barely falls short of the standard for publication, but I am happy to upgrade my score if the authors can address my concerns.

---

> ### Author Rebuttal · Authors · 2025-04-01
>
> >I suggest the author to condense Section 4. Remarks E.1 and G.2 should be included as part of the main body.
>
> Thanks for your suggestions. We will modify them in the revised paper.
>
> >It is okay to assume homo-scedasticity (same $\Lambda$ for both classes), but the assumption on the means having the same $\Lambda^{-1}$-weighted norms is hard to justify. I guess this assumption is made for the convenience of the proof. If so, the authors should be upfront about this and explain why the problem is intractable without this assumption.
>
> Yes, this assumption is made for the convenience of the proof. Actually, in Remark E.1 and G.2, we explained why we need this assumption and showed that if this assumption is not satisfied, the single-layer transformer cannot correctly perform the in-context classification tasks. We also verified this in experiments (Section 5.1, Figure 2) that the single-layer transformer cannot perform well for varying norms.
>
> >It seems that the derivation is about to connect the pre-trained attention model to linear/quadratic discriminant analysis (LDA/QDA) but then abruptly stops. Actually, I think it is very helpful to describe what kind of learning algorithm is implemented by the pre-trained transformer. Setting $W = 2\Lambda^{-1}$ implements the LDA decision criterion?
>
> Yes, we are connecting the pre-trained attention model to LDA and setting $W = 2\Lambda^{-1}$ implements the LDA decision criterion. We will add more discussions about the connection between the pre-trained attention model and LDA in the revised paper. Thanks for your suggestion.
>
> >What happens when you stack multiple layers?
>
> In our experiments (Section 5.1, Figure 2), we showed that multi-layer transformers have better robustness for varying covariances and norms. We believe developing a better understanding of multi-layer and more complex transformers is an intriguing direction for future research.
>
> >The well-conditioned property (9)
>
> es, we prove the that there exist $\alpha>0$ and $l<\infty$ that satisfy $(9)$. In Lemma D.2, we prove that in a compact domain, the strong convexity parameter $\alpha$ is larger than 0. In Lemma D.5, we show that $l\leq  \frac{1}{4}\sum_{i\in[d^2]}E[(p_{t_1(i)}q_{t_2(i)})^2]$, where $p,q$ are some combinations and rotations of Gaussian random variables. Since Gaussian random variables have finite second moments. Thus, $l<\infty$. We will clarify this in the revised paper.
>
> >I noticed that in Section D.1, the description of D.3 does not match the actual lemma
>
> What we exactly did in Lemma D.3 matches what we described in Section D.1. We are analyzing the Taylor expansion of $L(W)$ in Lemma D.3. In line 892-894, we show that as $N\to\infty$, our loss function $L(W)$ point-wisely converges to $\widetilde{L}(W)$. In line 926, we also show that as $N\to\infty$, the global minimizer $W^*$ converges to $2\Lambda^{-1}$. The logic of the proof here is that we first show the loss function $L(W)$ converges point-wisely to $\widetilde{L}(W)$, which implies that the global minimizer $W^*$ converges to $2\Lambda^{-1}$. Then, given the property that the global minimizer $W^*$ converges to $2\Lambda^{-1}$, in Lemma D.4, we can derive a tighter convergence rate for $W^*$. We will clarify this in the revised paper.
>
> >Figure 3: How many classes are there?
>
> We mentioned in the paper that we compare the classification of three Gaussian mixtures.
>
> >For the "inference error" are you referring to the TV distance in equation 3? I don't think this notion is directly applicable to k-nearest neighbor or SVM.
>
> We used KNeighborsClassifier and SVC from sklearn, and they provided the prediction probability for given test data. We then use the prediction probability to calculate the TV distance. See the API of sklearn for details.
>
> >I find this comparison to be unfair since the transformer models have been pre-trained on a lot (what is the exact number?) of data
>
> The exact number of training data can be found in H.2. Experiment Details. It is somewhat unfair to compare Transformer models with classical models. However, the goal of our experiments is not to make a fair comparison between Transformers and classical methods, nor to prove that Transformers are superior. Instead, our main purpose is to demonstrate that trained Transformers have in-context learning capabilities and perform no worse than classical methods.
>
> >Why is LDA not part of the classical baseline? You should include your source code
>
> We upload our code and additional results in the following link. https://anonymous.4open.science/r/In-Context-Classification-of-Gaussian-Mixtures-2374
> We added LDA in LDA_Comp.png
>
> >Adding a few references on the classical methods for Gaussian mixtures, e.g. linear discriminant analysis, would be very helpful.
>
> Thanks for this comment. We will cite and discuss a few references about Gaussian mixtures and linear discriminant analysis in the revised paper.

---

> > ### Comment · Reviewer_aULa · 2025-04-02
> >
> > I thank the authors for their detailed responses.
> >
> > > Yes, this assumption is made for the convenience of the proof.
> >
> > I don't think this is a fatal weakness, but please be upfront about it in your revision.
> >
> > > We will add more discussions about the connection between the pre-trained attention model and LDA in the revised paper.
> >
> > **I want to see a precise statement for this before I can decide if I should change my rating.**
> >
> > > What we exactly did in Lemma D.3 matches what we described in Section D.1.
> >
> > After reading the Lemma more carefully, I think you are right. But please consider revise Appendix D.1 to make it easier to understand.
> >
> > > We used KNeighborsClassifier and SVC from sklearn, and they provided the prediction probability for given test data
> >
> > I see. If I recall correctly, the prediction probability of `SVC` is computed by doing K-fold validation, which I think is not a fair comparison to make. But the most important comparison is with LQA, so I will forgive this.

---

> > > ### Author Response · Authors · 2025-04-04
> > >
> > > >I don't think this is a fatal weakness, but please be upfront about it in your revision.
> > >
> > > Thanks for your suggestions. We will do it upfront in the revised paper.
> > >
> > > >I want to see a precise statement for this before I can decide if I should change my rating.
> > >
> > > Thanks for your suggestions. We would like to add the following discussion in the revised paper to better illustrate the connections between the trained transformer and LDA.
> > >
> > > Our pre-trained single-layer transformer can be viewed as approximately implementing Linear Discriminant Analysis (LDA). To see this, we consider the binary classification case as example. Suppose we are given $x_i,y_i, i=1,..., M$ and $x_q$, and we need to predict the label $y_q$ for $x_q$. LDA assumes that $x_i,y_i, i=1,..., M$ and $x_q, y_q$ are i.i.d. samples, with $P(y=1)=P(y=-1)$, and the conditional probability density functions $f(x|y=1)$ and $f(x| y=-1)$ are Gaussian with means $\mu_1, \mu_{-1}$ and same covariance $\Sigma$. Under these assumptions, it can be derived that the optimal decision criterion for $x_q$ is to predict $y_q=1$ if $(\mu_1-\mu_{-1})^\top \Sigma^{-1}x_q+\frac{1}{2}(\mu_{-1}^\top\Sigma^{-1}\mu_{-1}-\mu_1^\top\Sigma^{-1}\mu_1)>0$ and $y_q=-1$, otherwise. LDA can estimate $\hat{\mu}\_1$ as the average of $x_i$ with $y_i=1$, $\hat{\mu}\_{-1}$ as the average of $x_i$ with $y_i=-1$, and estimate the covariance $\hat{\Sigma}$ from the within-class variances. The decision criterion becomes $(\hat \mu_1-\hat \mu_{-1})^\top \hat \Sigma^{-1}x_q+\frac{1}{2}(\hat \mu_{-1}^\top \hat \Sigma^{-1}\hat \mu_{-1}-\hat \mu_1^\top\Sigma^{-1}\hat \mu_1)$. For the single-layer transformer, it can compute the in-context estimate $\hat \mu_1-\hat \mu_{-1}=\sum_{i=1,…,M}y_ix_i/M$, however, it is hard for the single-layer transformer to compute $\hat \Sigma$ and $\hat \mu_{-1}^\top \hat \Sigma^{-1}\hat \mu_{-1}$ in-context. Thus, in our paper, we make the following assumptions. We assume the pre-train data and test data have the same covariance $\Lambda$ so that the transformer can learn an approximation of $\Lambda$ during pre-training. Moreover, we assume the two class means $\mu_1$, $\mu_{-1}$ have the same $\Lambda^{-1}$-weighted norm so that $\mu_{-1}^\top\Sigma^{-1}\mu_{-1}-\mu_1^\top\Sigma^{-1}\mu_1=0$. Under these assumptions, the quadratic term cancels out, and the decision criterion simplifies to $(\sum_{i=1,…,M}y_ix_i/M)^\top \Lambda^{-1}x_q$, which is very close to Eqn (12) in our paper. When we use $\hat W$ to approximate $2\Lambda^{-1}$, this becomes exactly Eqn (12). Therefore, when $\hat W = 2\Lambda^{-1}$ and the in-context examples are balanced across classes, the transformer's decision criterion is the same as that of the LDA with exact knowledge of $\Lambda$. In our experiment (see LDA\_Comp.png), since the pre-trained transformer has already learned a relatively good approximation of $\Lambda^{-1}$, while LDA must estimate $\Lambda^{-1}$ in context, the trained transformer significantly outperforms LDA when the number of in-context examples is small. As the context length increases, LDA's performance approaches that of the trained transformer. Thus, our paper theoretically demonstrate that the trained transformer can approximately implement LDA, and our experiments (LDA\_Comp.png) corroborate the theoretical findings.
> > >
> > > >After reading the Lemma more carefully, I think you are right. But please consider revise Appendix D.1 to make it easier to understand.
> > >
> > > Thanks for your suggestions. We will revise Appendix D.1 to make it easier to understand in the revised paper.
> > >
> > > >I see. If I recall correctly, the prediction probability of SVC is computed by doing K-fold validation, which I think is not a fair comparison to make. But the most important comparison is with LQA, so I will forgive this.
> > >
> > > Thanks. Yes, it is not a fair comparison. We will modify it in the revised paper.

---

### Official Review · Reviewer_je9W · 2025-03-25

**Overall Recommendation:** 3

**Summary:**

This paper looks at ICL for classification using Gaussian mixtures (same covariance across classes but different means) by trained transformers. They show that under the condition that the training and test data come from the same covariate covariance distribution, using linear attention can provably work with a single layer. They also show experimental evidence that multiple layers can do better as well as work even when this assumption of the same covariance distribution is broken.

**Claims And Evidence:**

Yes, although I must confess that I did not have time to read the full proofs or check them carefully.

**Essential References Not Discussed:**

They made a good choice I feel.

**Experimental Designs Or Analyses:**

Yes, see above.

**Methods And Evaluation Criteria:**

Yes, sort of. There is a key missing comparison in my opinion: to just doing "least-squares for classification" --- what is sometimes called the LS-SVM --- where the shared covariate covariance from the training and test data is known to this LS-SVM as well (either directly by magic or from just doing a dumb natural algorithm involving averaging over the training data to extract it.)

Given the work in Zhang, Frei, and Bartlett (2023a), this feels like the critical question.

**Other Comments Or Suggestions:**

None. Just answer my questions.

**Other Strengths And Weaknesses:**

Fundamentally, I felt that the lack of serious comparison to just the linear-regression approach is a big weakness. Treating classification as linear regression isn't optimal, but it can work decently well especially in the kinds of settings here.

So, what parts of the analysis are picking up extra nuances of what single-layer linear-attention transformers can do beyond linear regression and what parts are just casting this problem into its shadow linear-regression form in disguise? I can't tell from the discussion. But this is what I really want to know.

**Questions For Authors:**

How does just doing "least-squares for classification" --- what is sometimes called the LS-SVM --- do for the experiment in Figure 3? First straight-up least-squares knowing nothing. Second, one that has learned the covariance $\Lambda$ by magic. Third, one that has learned the covariance $\Lambda$ using the same training data provided to the transformers?

If I think about how linear regression for classification works in a high-enough-dimensional space, when the means are large, it works quite decently. Anyone who studies mixture-models as toys for classification knows that the signal-to-noise ratio is important. Given that your plots have inference errors dropping to by 5% or lower, it suggests that the SNR is decent. So how do the results change if we vary the SNR?

**Relation To Broader Scientific Literature:**

The problem of understanding exactly why transformer layers work and what the limitations are is an important problem. The paper does a good job of surveying the literature.

**Theoretical Claims:**

Not really. Sorry.

---

> ### Author Rebuttal · Authors · 2025-04-01
>
> We conducted additional experiments and uploaded code in https://anonymous.4open.science/r/In-Context-Classification-of-Gaussian-Mixtures-2374
>
> >Fundamentally, I felt that the lack of serious comparison to just the linear-regression approach is a big weakness. Treating classification as linear regression isn't optimal, but it can work decently well especially in the kinds of settings here.
> So, what parts of the analysis are picking up extra nuances of what single-layer linear-attention transformers can do beyond linear regression and what parts are just casting this problem into its shadow linear-regression form in disguise? I can't tell from the discussion. But this is what I really want to know.
>
> In SNR_LR_Comp.png, we conducted experiments comparing the transformer with linear regression for classification tasks. We can notice that trained transformers, in general, have better performance than linear regression, especially when the prompt length is small. This is not that surprising since the trained transformers somewhat learned $\Lambda$ during training.
>
> >Given the work in Zhang, Frei, and Bartlett (2023a), this feels like the critical question.
>
> Compared to [1], we use the cross-entropy loss for the classification problem. Thus, a critical distinction to [1] is that our model does not have a closed-form expression of the global minimizer, while [1] does. This distinction adds a significant challenge and leads to a different technical approach compared to [1]. By analyzing the Taylor expansion at the stationary point, we establish the convergence properties of $W^*$. Moreover, [1] considered optimizing the transformer using gradient flow. In contrast, our work proves the convergence of optimizing the transformer with the more practical gradient descent method. We will add more related discussions in our revised paper.
>
>
> >First straight-up least-squares knowing nothing. Second, one that has learned the covariance $\Lambda$ by magic. Third, one that has learned the covariance $\Lambda$ using the same training data provided to the transformers?
>
> In SNR_LR_Comp.png, we compared the trained transformer with straight-up linear regression, knowing nothing. Actually, as the reviewer aULa pointed out, our pre-trained attention transformer approximately implements LDA. If a LDA model learned the covariance $\Lambda$ by magic or learned the covariance $\Lambda$ using the same training data provided to the transformers, this LDA model should have better or comparable performance as the trained transformer.
>
> >If I think about how linear regression for classification works in a high-enough-dimensional space, when the means are large, it works quite decently. Anyone who studies mixture-models as toys for classification knows that the signal-to-noise ratio is important. Given that your plots have inference errors dropping to by $5\%$ or lower, it suggests that the SNR is decent. So how do the results change if we vary the SNR?
>
> In SNR_LR_Comp.png, by changing the magnitude of the covariance matrix of testing data ($\Lambda, 4\Lambda, 16\Lambda$), we compared the accuracy of trained transformers and linear regression with different rates of SNR. Our results show that smaller SNR leads to worse accuracy for both the transformer model and the linear regression model.
>
> References can be found in our reply to reviewer M6vU.

---

### Official Review · Reviewer_yaQy · 2025-03-25

**Overall Recommendation:** 2

**Summary:**

In this paper, the authors provide a theoretical analysis of in-context learning of linear classification tasks on the Gaussian mixture models. By assuming a simplified linear self-attention structure and fixing some parameters during the whole training, the authors prove that linear attention can converge to the global minimum in linear convergence rate when minimizing the population loss with gradient descent. Additionally, the authors prove that the global minimum can achieve a small total variance between the output of the trained model and the true label when applied to a new testing prompt. Finally, the authors conduct simple numerical experiments, supporting their conclusions.

**Claims And Evidence:**

From my perspective, there exist some major concerns regarding the assumptions and conclusions of this paper. I will list them as follows and discuss the details in the next "Methods" and "Theory" sections.

1. By assuming an over-simplified attention structure, the optimization problem equation (7) is a convex optimization problem obviously, which is almost totally understood. Additionally, given the loss function (cross-entropy loss function) considered in equation (7) and equation (17), the optimization problem considered in this paper is essentially a logistic regression for the binary case and multi-class case respectively, which is highly understood. I do not feel that there exist any essential challenges in converting the conclusions from binary cases to multi-class cases. Therefore, I do not feel that the optimization problem considered in this paper is as highly non-linear (actually this is a generalized linear model) and challenging as they claimed.

2. Additionally, the author proposes utilizing total variance as the criteria to evaluate the test performance of the new prompt. However, the total variance would only imply the similarity between the distributions, instead of the random variables themselves. Consequently, the Theorem 3.6 implies nothing regarding the testing performance.

**Essential References Not Discussed:**

There are several theoretical studies on the optimization of one-layer transformers that consider training $W^v$ and $W^{KQ}$ simultaneously, even in the more challenging softmax attention setting [2, 3, 4, 5]. There are also results on general convergence guarantees of transformers such as [6]. In addition, [7] considers almost the same question as this paper. I suggest that the authors should compare their results with these works.

[2]. Jelassi, S., Sander, M. and Li, Y., 2022. Vision transformers provably learn spatial structure. NeurIPS

[3]. Li, H., Wang, M., Liu, S. and Chen, P.Y., A Theoretical Understanding of Shallow Vision Transformers: Learning, Generalization, and Sample Complexity. ICLR

[4]. Wang, Z., Wei, S., Hsu, D. and Lee, J.D., 2024, July. Transformers Provably Learn Sparse Token Selection While Fully-Connected Nets Cannot. ICML

[5]. Zhang, C., Meng, X. and Cao, Y., 2025. Transformer learns optimal variable selection in group-sparse classification. ICLR

[6]. Gao, C., Cao, Y., Li, Z., He, Y., Wang, M., Liu, H., Klusowski, J.M. and Fan, J., Global Convergence in Training Large-Scale Transformers.  NeurIPS

[7]. Frei, S. and Vardi, G., Trained Transformer Classifiers Generalize and Exhibit Benign Overfitting In-Context. ICLR

**Experimental Designs Or Analyses:**

I have checked the experimental results. There is no major incorrectness.

**Methods And Evaluation Criteria:**

1. As I mentioned in the previous section, the attention layer considered in this paper, adopts a simplified linear attention structure. Additionally, it constrains all the parameters in $W^V$ and $W^{KQ}$ as fixed except the left-top block in $W^{KQ}$. Therefore, all the nonlinearity of this model comes from the cross-entropy loss function. Such oversimplification has rendered this linear attention model equivalent to logistic regression, for which the conclusion that this represents a strongly convex optimization problem within each compact set is well-established. Additionally, the data model of this paper is specified as a Gaussian mixture, which is obviously non-linear separable when given the size of the training set, i.e. $B$ is infinitely large. Therefore, the loss function is coercive and must have a unique minimum given the strongly convex property. Therefore, the optimization problem considered in this paper, from my perspective, is well-studied and trivial to some extent. I have reviewed the proof details and found that most of the proof in this paper focuses on establishing the strong convexity of the cross-entropy loss, which is a well-known fact. Note that even in the original paper studying the in-context learning for training [1], the authors consider a more practical training strategy without fixing any parameters, which makes their loss non-convex and the theoretical analysis highly non-trivial. Besides, there exists multiple theoretical studies that consider training the $W^v$ and $W^{KQ}$ simultaneously, even in a more challenging softmax attention setting [2, 3, 4, 5].

2. Additionally, as I mentioned in the previous section, total variation cannot be used to evaluate test performance. To illustrate this, consider a simple example where the true label $y$ is a Rademacher random variable, and I choose $\hat y = -y$ as the prediction of this label $y$. According to the total variation formula in equation (3), we have $\Delta(y,
\hat y)=0$. However, the test accuracy for this prediction $\hat y$ is $0$. Consequently, the conclusion of Theorem 3.6 does not provide any meaningful insights regarding test performance.

[1]. Zhang, R., Frei, S. and Bartlett, P.L., 2024. Trained transformers learn linear models in-context. JMLR

[2]. Jelassi, S., Sander, M. and Li, Y., 2022. Vision transformers provably learn spatial structure. NeurIPS

[3]. Li, H., Wang, M., Liu, S. and Chen, P.Y., A Theoretical Understanding of Shallow Vision Transformers: Learning, Generalization, and Sample Complexity. ICLR

[4]. Wang, Z., Wei, S., Hsu, D. and Lee, J.D., 2024, July. Transformers Provably Learn Sparse Token Selection While Fully-Connected Nets Cannot. ICML

[5]. Zhang, C., Meng, X. and Cao, Y., 2025. Transformer learns optimal variable selection in group-sparse classification. ICLR

**Other Comments Or Suggestions:**

No.

**Other Strengths And Weaknesses:**

No.

**Questions For Authors:**

Is there a typo in line 719? Should it be $\frac{1}{4N^2}$ instead of $\frac{1}{4N}$?

**Relation To Broader Scientific Literature:**

As I said in the previous section, even compared with the original paper [1], the model considered in this paper is much over-simplified.

[1]. Zhang, R., Frei, S. and Bartlett, P.L., 2024. Trained transformers learn linear models in-context. JMLR

**Theoretical Claims:**

Besides the major concerns I proposed in the previous sections, I still have the following concerns for the proof details.

1. The authors obscure several important conditions in their presentation of theorems. What concerns me most is their decision to drop all terms and factors that are independent of $N$ during the training process. In most theoretical studies related to ICL or transformers, the embedding dimension $d$, is an extremely critical parameter. However, in this paper, the authors directly treat $d$, along with the norms of the mean vectors $\mu_1$ and $\mu_2$, and other factors related to the spectral distribution of the variance matrix $\Lambda$ at the scale of constant order, neglecting these factors entirely. In comparison, in [1], the authors retain all these terms in their conclusions of theorems, therefore the readers can clearly understand how large the sequence length $N$ is required to cancel the effect from the other terms and achieve a good performance.

2. Additionally, I also do not understand why the query token $q$ can appear on the RHS of the formula of Theorem 3.6 when you are taking the expectation over the query token pair.

3. I find that Assumptions 3.2 and 3.5, which state that the mean vectors $\mu_1$ and $\mu_2$ have the same $\Lambda^{-1}$ norm, are somewhat strong. Given these assumptions, I do not see a significant difference from directly assuming an isotropic Gaussian noise.

---

> ### Author Rebuttal · Authors · 2025-04-01
>
> >By assuming an over-simplified attention structure...
>
> Setting some parameters to fixed values and considering spare form parameters is commonly used in ICL theory papers [7,8,9,10], and we adopt a similar parameterization as in [7,8,10]. Even for this simplified structure, our analysis for the convergence of $W^*$ is nontrivial. For example, in order to get a tighter bound for $W^*$, in Lemma D3, we first show the loss function $L(W)$ converges point-wisely to $\widetilde{L}(W)$, which implies that the global minimizer $W^*$ converges to $2\Lambda^{-1}$. Then, given the property we prove in Lemma D3, we can derive a tighter convergence rate for $W^*$ in Lemma D4 than the bound we got in Lemma D3.
>
> >total variation cannot be used to evaluate test performance.
>
> We thank the reviewer for this comment, but we believe that total variation distance can be used to evaluate test performance in our setting. We will illustrate this point using a binary case example: Suppose for data $x_q$, the conditional distribution of the ground truth label $y$ is $P(y=1|x_q)=p$, $P(y=-1|x_q)=1-p$. Suppose the prediction of the model is $z$, which depends on $P=(x_1, y_1, ...,x_M, y_M, x_q)$. Suppose the conditional distribution of $z$ is $P(z=1|P)=q$, $P(z=-1|P)=1-q$. Since $(x_i, y_i)$ are i.i.d. sampled, for given $P=(x_1, y_1, x_2, y_2, ...,x_M, y_M, x_q)$, $z|P$ and $y|x_q$ are independent. Thus, the example given by the reviewer is not valid in our setting. If we consider the accuracy, we can calculate it as $Acc(y, z)=pq+(1-p)(1-q)=1+2pq-p-q$. We can notice that, if $p>1/2$, the best $q$ to maximize the accuracy should be $q=1$ and if $p<1/2$, the best $q$ to maximize the accuracy should be $q=0$. Thus, if the output of the model $\widehat y_{out} | P$  is the same as $P(y=0|x_q)$, we can output $z$ based on the value of $\widehat y_{out} | P$ to maximize the possible accuracy. The smaller the difference between $\widehat y_{out} | P$ and $P(y=0|x_q)$, the better the model performs. Actually, the total variation distance in our case measures this difference.
>
> >What concerns me most is their decision to drop all terms and factors that are independent of $N$ during the training process...
>
> We'd like to clarify that we did not drop all terms and factors that are independent of $N$ during the training process. For example, in Theorem 3.3 Eqn (10), we retained the coefficient of the $1/N$ term and showed that this coefficient is affected by the distribution of distance between two means $\mu=\mu_{\tau, 1}-\mu_{\tau, 0}$, the query, and the covariance matrix $\Lambda$. Intuitively, the classification problem becomes easier when the distance between the two class means $\mu$ increases and the variance $\Lambda$ decreases; conversely, it becomes harder when $\mu$ is small and $\Lambda$ is large. Eqn (10) reflects this intuition. For instance, when $\mu$ is large and $\Lambda$ is small -- corresponding to an easier classification problem -- the parameter $a$ becomes larger, and the derivative $\sigma'(a)$ decays exponentially with increasing $a$, thereby reducing the overall training error. On the other hand, when $\mu$ is small and $\Lambda$ is large -- corresponding to a harder classification problem -- the training error increases accordingly. This theoretical prediction aligns well with our intuition. We did not explicitly present results in terms of the dimensionality $d$, because intuitively, $d$ is not the primary factor determining the classification difficulty. Instead, it is the relationship between $\mu$ and $\Lambda$ that plays the central role. Our theory captures this key insight effectively.
>
> >I also do not understand why the query token $q$ can appear on the RHS of the formula of Theorem 3.6 when you are taking the expectation over the query token pair.
>
> We did not take the expectation over the query token $q$. You can check Theorem 3.6. Our expectation is take over $x_i, y_i; i=1,.., M$, which does include $q$.
>
> >I find that Assumptions 3.2 and 3.5, which state that the mean vectors $\mu_1$ and $\mu_2$ have the same $\Lambda^{-1}$ norm, are somewhat strong.
>
> See our second reply to reviewer aULa.
>
> >There are several theoretical studies on the optimization of one-layer transformers that consider training $W^v$ and $W^{KQ}$ simultaneously, even in the more challenging softmax attention setting [2, 3, 4, 5]. There are also results on general convergence guarantees of transformers such as [6]. In addition, [7] considers almost the same question as this paper. I suggest that the authors should compare their results with these works.
>
> Thank you very much for providing those related works. We will cite and discuss them in the revised paper.
>
> >Is there a typo in line 719? Should it be $\frac{1}{4N^2}$ instead of $\frac{1}{4N}$?
>
> No. This is not a typo. You can verify that the result is $\frac{1}{4N}$.
>
> References can be found in our reply to reviewer M6vU.

---

> > ### Comment · Reviewer_yaQy · 2025-04-02
> >
> > 1. The authors' rebuttal can not address my concerns regarding the technical contribution of the optimization problem considered in this paper.  As I mentioned very clearly, the over-simplified settings reduce the optimization of transformers to a logistic regression. Under this classic setting, the existence of a global minimum, the strong convexity, the smoothness, and linear convergence have been well-studied and demonstrated in the previous works. The author seems to be deliberately ignoring this comment in their rebuttal. Based on this discussion, the only theoretical contribution of the training optimization is to provide a characterization of the global minimum. However, we need to further note that there is a closed-form for this global minimum. The author only provides a range for this minimum. When $N$ is not large, this range is loose and can not provide informative guidance for the global minimum. On the other hand, when $N$ is large, LLN guarantee that $\frac{1}{N}\sum x_i y_i \to \frac{1}{2}(\mu_1 -\mu_0)$, which provide intuitive choice of the global minimum. I do not feel this is a challenging issue based on all previous discussions. As I mentioned, several existing works have addressed more impractical and challenging settings compared to those presented in this paper.
> >
> > 2. I completely fail to understand why the independence between$\hat y$ and $y$ is of such great importance. Even following the authors' requirement for independence between $\hat{y}$ and $y$, I can still propose counterexamples. Let $\hat{y}$ and $y$ be two independent Rademacher random variables; then $\Delta(\hat{y}, y) = 0$, but there still exists a half classification error. I believe I have clearly stated that, due to the existence of such counterexamples, the total variation itself is not suitable for measuring test performance, nor is it a common choice, as it only addresses the distance between distributions rather than the relationship between random variables. There are standard metrics for evaluating population generalization, such as test loss (utilized in [1]) and test error (utilized in [7]). I suggest that the authors consider these metrics as alternatives to total variation.
> >
> > 3. Additionally, if you do not take the expectation over $q$, how can you guarantee the RHS of your Theorem 3.6 is small, given that $q$ is generated from an unbounded distribution and can take on extremely large values? If such a guarantee can only hold with high probability, it must be explicitly calculated and illustrated how it affects your final results.
> >
> > 4. I respectfully disagree with the authors' claim that they did not drop any factor. If not, why does there appear a term $o(\frac{1}{n})$? I wonder what the original numerator and denominator of this term are and why this term could be represented as $o(\frac{1}{n})$ if you never use $N$ to cancel other factors? Indeed, I have checked the proof details, and the authors directly treat a term as $o(\frac{1}{n})$ if the denominator contains a factor $n$ with power larger than 1, regardless of the numerator. As I clearly mentioned, I do not feel it's reasonable to directly drop these terms, as the authors do not claim any requirement or assumptions for the scale of $N$ compared to other parameters. I also do not recognize the authors' claim that "$d$  is not the primary factor determining the classification difficulty". There is no doubt that the dimension $d$ is of great importance for studying the modern over-parameterized and high-dimensional regimes. I sincerely suggest the authors compare their presentations regarding theorems with those of [1].
> >
> > 5. I know that your conclusion fails without such an assumption, and this is the issue. If your conclusion has to be established under such an assumption, what is the essential difference with the settings considering isotropic noise?
> >
> > In summary, I'm not satisfied with the current manuscript or the rebuttal from the authors. All my previous concerns remain.

---

> > > ### Author Response · Authors · 2025-04-04
> > >
> > > 1. Since the reviewer claims there is a closed-form solution, we would greatly appreciate it if the reviewer could provide it. To the best of our knowledge, we are aware of an $l_2$-max margin solution in [7] for a setting similar to ours, which however is not a closed-form solution. Moreover, to ensure the max-margin solution in [7] is well-behaved enough for their theoretical analysis, they made additional assumptions, such as requiring a sufficiently large signal-to-noise ratio. In contrast, our work does not rely on such assumptions. Additionally, compared to [7], we consider a more general multi-class setting. We believe these distinctions highlight the independent contributions of our work.
> > >
> > > 2. If for a given $x_q$, the ground truth label $y|x_q$ is a Rademacher random variable, i.e. $P(y=1|x_q)=0.5$, $P(y=-1|x_q)=0.5$, as the reviewer claimed, in the setting of our paper, this only happens when there are two Gaussian classes with $\mu_1$, $\mu_{-1}$ and $x_q$ lies exactly on the line where $(\mu_1-\mu_{-1})^\top\Lambda^{-1} x_q=0$ (for example, perpendicular to $(\mu_1-\mu_{-1})$ when $\Lambda=I$). In this situation, even the best classifier will inevitably have a 50% classification error, and this error is **intrinsic**. As a result, we do not consider this to be a problem. Total variation distance ($\Delta(y, \hat y)$) can serve as the loss to measure the difference between two independent random variables. Moreover, if we consider the cross-entropy loss between $\hat{y}$ and $y$ ($CEL(\hat{y}; y)$), a bounded TV distance also means a bounded $CEL(\hat{y}; y)$. One can easily prove that the minimum of cross-entropy loss $CEL(\hat{y}; y)$ is achieved when $\hat{y}$ has the same distribution as $y$, which means $\Delta(y, \hat y)=0$. Moreover, denoting the $\delta=\min_{i} P(\hat y=i)$, one can easily prove that $CEL(\hat{y}; y)\leq CEL(y; y)+\Delta(y, \hat y)/\delta$. We will clarify it in the revised paper.
> > >
> > >
> > > 3. We showed how $q$ affects the inference error in Theorem 3.6. Actually, in most cases, **extremely large $q$ makes the classification problem extremely easy**. For example, let us consider the binary case and let $\Lambda=I$ without loss of generality. We can see that the main coefficient of the inference error in Theorem 3.6 is $\sigma'(\mu^\top\Lambda^{-1}q)=\sigma'(\mu^\top q)$. Thus, if $\mu^\top q = 0$, $q$ lies exactly on the line perpendicular to $\mu$. In this case, there is an equal 50% chance of belonging to class 1 or -1. However, in most cases, when $\mu^\top q \neq 0$, an extremely large $q$ makes $\sigma'(\mu^\top q)$ extremely small, because $\sigma'(x)$ decays exponentially as $|x|$ increases. This, in turn, causes the inference error in Theorem 3.6 to become extremely small as well (in Line 1134, you can see that the coefficient of the $o(1/N + 1/\sqrt{M})$ term is related to $\sigma''(\mu^\top q)$, which also becomes extremely small). This also matches the real-world behavior: as long as $q$ does not lie on the line perpendicular to $\mu$, an extremely large $q$ makes the classification problem extremely easy. This example demonstrates the consistency between our theory and reality.
> > >
> > >
> > > 4. Actually, we kept the coefficient of $1/N$ and dropped the coefficients of $o(1/N)$.
> > >
> > > >I do not feel it's reasonable to directly drop these terms, as the authors do not claim any requirement or assumptions for the scale of $N$ compared to other parameters.
> > >
> > > When we use the notion $o(1/N)$, the notion itself contains the assumption that $N$ is sufficiently large. Thanks for the reviewer's suggestion, we will clarify this in the revised paper.
> > >
> > > >There is no doubt that the dimension $d$ is of great importance for studying the modern over-parameterized and high-dimensional regimes.
> > >
> > > As we have clarified in the original rebuttal, **$d$ is not the direct and primary factor determining the classification difficulty**. Factors like the SNR (signal-to-noise ratio, here we can consider it as the relationship between the $\mu_1-\mu_{-1}$ and $\Lambda$) play a more critical role. For example, one can easily construct Gaussian mixtures with a large $d$ but low SNR, and compare with another one with a small $d$ but high SNR. In these cases, the latter can be significantly more difficult to classify than the former. The impact of $d$ in our setting is more on the memory and computation costs of the model. However, as our main focus is on the inference error, we have correspondingly focused on how relevant factors like $\mu, \Lambda, q, N, M$ affect the inference error.
> > >
> > > 5. We do not think our setting is equivalent to isotropic noise. We couldn’t quite understand this comment. Could you clarify?

---

### Decision · Program_Chairs · 2025-05-01

**Decision:**

Accept (poster)

**Comment:**

This paper studies in-context learning of linear classification on Gaussian mixture data. The authors show that a single-layer transformer with linear attention trained via gradient descent converges to an optimal model, and demonstrate that when the lengths of training and testing prompts are sufficiently large, the trained transformer can perform near-optimal prediction.
Several reviewers have shared the concern that the results are not very surprising given the simplifications considered in the paper: the training is essentially linear logistic regression. However, after several rounds of discussions, the reviewers agree that the theoretical studies and conclusions are sound. Overall, the paper is on the borderline, and the rating slightly leans towards acceptance.